# Global impact of somatic structural variation on the cancer proteome

Fengju Chen[1,8], Yiqun Zhang [1,8], Darshan S. Chandrashekar[2,3,4], Sooryanarayana Varambally[2,3,5] & Chad J. Creighton [1,6,7] ✉

Both proteome and transcriptome data can help assess the relevance of non-coding somatic mutations in cancer. Here, we combine mass spectrometry-based proteomics data with whole genome sequencing data across 1307 human tumors spanning various tissues to determine the extent somatic structural variant (SV) breakpoint patterns impact protein expression of nearby genes. We find that about 25% of the hundreds of genes with SV-associated cis-regulatory alterations at the mRNA level are similarly associated at the protein level. SVs associated with enhancer hijacking, retrotransposon translocation, altered DNA methylation, or fusion transcripts are implicated in protein over-expression. SVs combined with altered protein levels considerably extend the numbers of patients with tumors somatically altered for critical pathways. We catalog both SV breakpoint patterns involving patient survival and genes with nearby SV breakpoints associated with increased cell dependency in cancer cell lines. Pan-cancer proteogenomics identifies targetable non-coding alterations, by virtue of the associated deregulated genes.

At a global level, gene transcription is normally tightly regulated in cells. In cancer, however, this regulation is lost, resulting in widespread aberrant over-expression or under-expression of genes, including oncogenes and tumor suppressor genes, respectively[1]. Transcript alterations—including deregulated expression and gene fusions—often result from somatic changes in cancer genomes[2]. Somatic structural variation in cancer, stemming from genomic rearrangements, is one major driver of altered transcription through Copy Number Alterations (CNAs), gene fusions, and altered cis-regulation[2–8]. Altered gene cis-regulation, as mediated by structural variants (SVs) with genomic breakpoints falling in proximity to genes, may involve diverse mechanisms, not limited to enhancer hijacking, disruption of Topologically Associating Domains (TADs), and altered DNA methylation[5,7,9]. Whole Genome Sequencing (WGS) enables the detection of somatic SVs impacting both coding and non-coding

regions of the cancer genome[10]. Recently, the Pan-Cancer Analysis of Whole Genomes (PCAWG) consortium comprehensively synthesized and collated WGS data on cancer genomes from some 2658 patients[11], The Cancer Genome Atlas (TCGA) consortium and International Cancer Genome Consortium (ICGC) having generated these data. Some 1220 of the 2658 patients had tumors with gene transcription data by RNA-sequencing, which allowed for systematic cataloging of genes recurrently altered in expression by SVs, through gene fusion or altered cis-regulation, across cancers of diverse lineages[2–5].

In annotating the true functional impact of specific somatic SV events in cancer, which could have implications for personalized medicine[12], both protein and mRNA expression should ideally be considered. Gene expression can be regulated both at the transcriptional and post-transcriptional levels[1]. For most genes, transcript levels across cancers only partially predict the corresponding protein

[1]Dan L. Duncan Comprehensive Cancer Center Division of Biostatistics, Baylor College of Medicine, Houston, TX, USA. [2]Molecular and Cellular Pathology, Department of Pathology, University of Alabama at Birmingham, Birmingham, AL 35233, USA. [3]O'Neal Comprehensive Cancer Center, University of Alabama at Birmingham, Birmingham, AL 35233, USA. [4]Genomic Diagnostics and Bioinformatics, Department of Pathology, University of Alabama at Birmingham, Birmingham, AL 35233, USA. [5]The Informatics Institute, University of Alabama at Birmingham, Birmingham, AL 35233, USA. [6]Human Genome Sequencing Center, Baylor College of Medicine, Houston, TX 77030, USA. [7]Department of Medicine, Baylor College of Medicine, Houston, TX, USA. [8]These authors contributed equally: Fengju Chen, Yiqun Zhang. ✉e-mail: creighto@bcm.edu

levels[13–16]. Recent technological advancements in mass spectrometry (MS)-based proteomics technologies have allowed for profiling the expression of tens of thousands of protein features across hundreds of human tumor specimens[17]. Major scientific endeavors such as the Clinical Proteomic Tumor Analysis Consortium (CPTAC) have generated MS-based proteomic profiling data combined with corresponding multi-omics data on multiple cancer types defined by histology and tissue-of-origin[14–16]. WGS combined with MS-based proteomics data currently in the public domain would cumulatively involve over 1300 tumors, comparable to the set of tumors previously analyzed by the PCAWG consortium for WGS combined with mRNA expression[4], which numbers would be needed to study sparse somatic alteration events such as SVs[3].

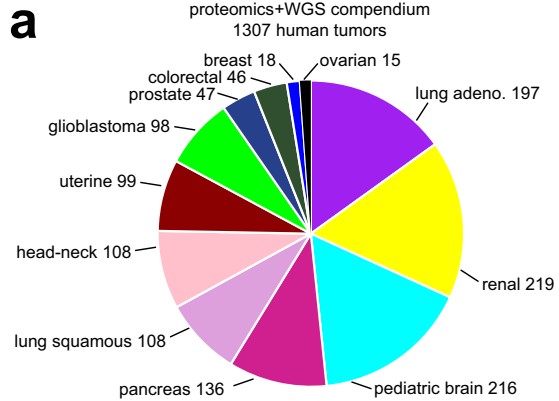

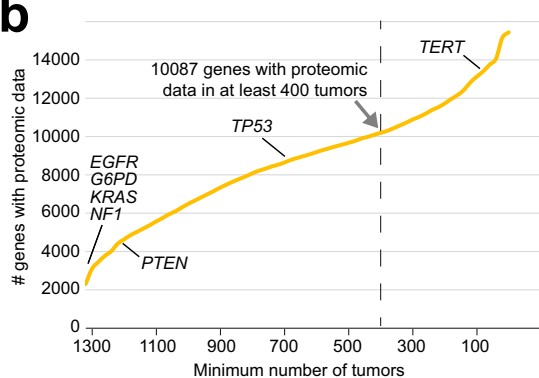

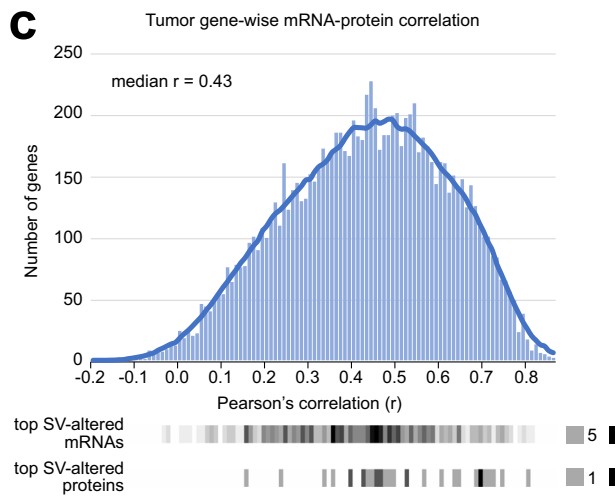

**Fig. 1 | Combined proteomic and WGS datasets and associated gene features used in this study. a** Combined proteomic and WGS data involved 1307 tumors from 1290 patients, broken down by the 12 major cancer types represented. **b** From the 1307 tumors, the numbers of tumors for which a given number of genes have proteomic data detected by mass spectrometry. Of the 15,439 genes represented in the combined proteomics and WGS compendium, 10,087 had protein values for at least 400 tumors. The 10,087 genes were the focus of the SV-expression associations of Fig. 2. **c** Histogram of gene-wise Pearson correlations of mRNA and protein expression across 1294 of the 1307 tumors from part **a**, based on the 10,087 genes with protein data for at least 400 tumors. Genes with SV-associated altered expression at either the mRNA or protein levels (using FDR < 10%, 442 genes for mRNA and 32 genes for protein) are indicated along the bottom. The densities of the respective top SV-associated gene sets with respect to the mRNA-protein correlations across tumors are denoted by the degree of shading (e.g., for mRNA: black denotes 15 or more genes with the given Pearson's r-values). See Supplementary Dataset 1 and Supplementary Fig. 1 to obtain a breakdown of tumors by data platform availability and patient cohort.

In this present study, we combine MS-based proteomics data with WGS and other multi-omics data across different cancer types to determine to what extent somatic SV patterns impacting genes at the mRNA level are reflected at the protein level. We systematically catalog gene-level associations with altered expression—by either mRNA or protein—in conjunction with nearby SV breakpoints. We focus on associations not attributable to CNA but to altered cis-regulation, gene fusions, or gene disruption. We find that a substantial percentage of SV-altered mRNAs were not similarly altered at the protein level. However, numerous cancer-relevant genes of interest are reflected at both protein and mRNA levels. SVs also involve DNA methylation alterations in a fraction of associated protein expression changes. We find somatic SVs and associated altered protein levels to considerably extend the numbers of patient tumors somatically altered for critical cancer-associated and targetable pathways. In addition, we can associate a significant fraction of genes with SV-protein associations with patterns of worse patient survival or with sensitivity to knockout in cancer cell lines.

## Results

### Compendium of SVs and protein expression

We assembled a compendium dataset of combined WGS and gene expression data on 1426 human tumors (representing 1409 patients) from multiple public sources (Supplementary Dataset 1 and Supplementary Fig. 1a, b)[4–6,14–16,18–30]—involving CPTAC, TCGA, ICGC, Children's Brain Tumor Network (CBTN), and Applied Proteogenomic OrganizationaL Learning and Outcomes (APOLLO). Of the 1426 tumors, 1307 (representing 1290 patients) had protein expression by mass spectrometry-based proteomics data, 1413 had mRNA expression by RNA-seq or microarray (including 1294 tumors with corresponding proteomic data), and 988 had DNA methylation array data. The combined WGS-proteomics dataset involved 12 cancer types (Fig. 1a). For the proteomic and transcriptomic compendium datasets, we normalized expression values within each cancer type, whereby neither tissue-dominant differences nor inter-laboratory batch effects would drive the downstream analyses (Supplementary Fig. 1c–g)[13,31–34]. The total proteomics compendium dataset consisted of 15,439 genes with proteins measured in at least one tumor, including 10,087 genes with proteins represented in at least 400 tumors profiled (Fig. 1b). In contrast, other genes of potential interest, such as TERT[4,8], had protein levels undetected in most tumors studied due to limitations of the mass spectrometry platform. The phospho-protein compendium consisted of 199,284 phospho-protein features involving 11,671 genes, 8835 phospho-proteins represented in 400 tumors profiled. As anticipated[13], protein expression of genes broadly correlated with the corresponding mRNA levels across tumors (Fig. 1c), though not with a high degree of correlation for most genes (median Pearson's r-value of

0.43). Due to the discordance between protein and mRNA, many genes previously associated with SV-altered expression at the mRNA level[4–6,19,35] may not be statistically significant at the protein level. We deposited the proteomic and phospho-proteomic data into the UAL-CAN cancer data analysis portal (https://ualcan.path.uab.edu)[36,37].

## Gene-level SV-associated protein alterations

By integrating proteomic and transcriptomic data with WGS-derived SV data[38], hundreds of genes showed significantly altered gene expression—at either the protein or mRNA levels—in relation to nearby somatic SV breakpoints (relative to tumors without breakpoints), that would not be explainable due to any associated CNA. For associating SV breakpoints with altered protein expression, we focused here on the set of 10,087 genes in our compendium with protein expression detected for at least 400 tumors (Fig. 1b). SV breakpoints associated with altered expression include breakpoints located downstream or upstream of genes or occurring in the gene body (Fig. 2a and Supplementary Datasets 2 and 3). Significant genes after CNA correction (see Methods and Supplementary Fig. 2a, b) represent those for which any SV-associated CNA[6] by itself would not explain away the SV-associated altered expression patterns observed. Consistent with previous observations[4–6,19,35,38], many more genes in our 1426-tumor compendium dataset showed positive correlations with SV breakpoints (i.e., higher expression associated with nearby breakpoint) than negative correlations, the former including known oncogenes and the latter including tumor suppressor genes. At the mRNA level, top SV-associated altered genes in our WGS-mRNA cohort of 1413 tumors overlapped highly with the top genes as previously identified using TCGA and PCAWG cohorts[5] (Supplementary Fig. 2c). We could also define SV-associated protein and mRNA alterations according to cancer type as defined by tissue of origin, including genes that were not statistically significant in the pan-cancer analyses (Supplementary Fig. 2d, e).

For either mRNA or protein levels, a set of 1200 recurrently altered genes by SV breakpoints combined with expression differences, with False Discovery Rate (FDR)[39] of <10%, were identified across the set of region windows examined relative to genes: 100 kb upstream, 100 kb downstream, within the gene (including gene fusion events), and 1 Mb upstream or downstream (including long-range effects by enhancer hijacking events, etc.). Only a fraction of the genes significant for a given region at the mRNA level were significant at the protein level (Fig. 2a–c). When comparing gene-level SV associations common between mRNA and protein, we utilized relaxed statistical criteria to lower false negatives, consistent with previous studies[40]. In addition, we generated two sets of gene-level SV-mRNA associations (Supplementary Dataset 3 and Supplementary Fig. 2a, b), using both the complete mRNA compendium with all available expression data and a "filtered" mRNA dataset with any values not represented in the protein dataset filtered out. With the filtered mRNA dataset, any disparate results observed between protein and mRNA should have less to do with the diminished power for some genes due to lack of detection by proteomics. Out of 10,087 genes, 657 had a significant SV association for the 1 Mb region ($p < 0.01$ with cancer type and CNA correction) at the mRNA level using the unfiltered dataset, of which just 170 (26%) had a corresponding association at the protein level ($p < 0.05$), this overlap being highly statistically significant (chance expected overlap of 31 genes, one-sided Fisher's exact test $p < 1E–60$ for over-expressed genes, $p < 1E–13$ for under-expressed genes). When using the filtered mRNA dataset, 574 genes had SV associations at the mRNA level ($p < 0.01$), of which 180 (31%) were concordant by protein analysis ($p < 0.05$, Fig. 2b, c and Supplementary Dataset 3). Alternatively, using a more stringent statistical cutoff for protein, at $p < 0.01$, with a less stringent cutoff for mRNA, at $p < 0.05$ for the filtered dataset, a relatively higher percentage—57%—of the 245 statistically significant proteins were also significant by mRNA. Any observed

discordances between protein and mRNA can stem in part from biology (e.g., post-transcriptional regulation) and statistical false negatives.

For a set of 201 genes with significant SV-associations for both protein and mRNA ($p < 0.05$ for both, $p < 0.01$ for either, using filtered mRNA dataset), these genes were enriched for specific gene categories, including 'ATP hydrolysis activity' and 'progesterone metabolic process' (Fig. 2b and Supplementary Dataset 2, the latter involving aldo-keto reductase family 1 genes). In contrast, the 319 genes with significant SV-association for mRNA but not protein (mRNA $p < 0.01$, protein $p > 0.05$) were enriched for other gene categories, including 'ribonucleoprotein complex' and 'mRNA splicing – via spliceosome' (Fig. 2b and Supplementary Dataset 3). The 201 concordant genes had higher mRNA-protein correlations on average across the 1307 tumors than the 319 discordant genes (average Pearson's r of 0.49 versus 0.36, $p < 1E–19$ by t-test). The 201 concordant genes, but not the 301 discordant genes, were statistically enriched ($p = 0.03$, one-sided Fisher's exact test) for genes found elsewhere[41] to harbor somatic SNV hotspots, involving eight genes: *CDK4*, *EGFR*, *ERBB2*, *KRAS*, *PLCB3*, *S100A3*, *TACC3*, and *COBL*. The 201 genes and the 319 genes did not significantly differ in terms of intrachromosomal SV size (6.4 versus 6.2 Mb, respectively, $p = 0.44$, t-test on logged values) or of SV class representation (translocation, deletion, amplification, inversion, insertion). Well-known oncogenes and tumor suppressor genes had protein expression impacted by SV breakpoints in a sizable number of tumors—on the order of 1 to 3% with available data—that did not harbor amplifications in the given gene in the case of oncogenes (Fig. 2c). Genes of interest, previously associated with SV-altered expression at the mRNA level, found here also to show a corresponding association include AKR1C family genes[3,42] and *IGF2*[5] (Fig. 2d). We also identified genes with altered phospho-protein expression associated with nearby SV breakpoints, which genes overlapped significantly with the genes with SV-associated altered total protein expression (Supplementary Fig. 3 and Supplementary Dataset 2). All other mRNA-related results reported below rely on the unfiltered mRNA dataset.

## Gene fusions with protein expression

Somatic SV breakpoints falling within genes and associated with their increased protein expression may represent gene fusions. As done previously in other patient cohorts[6,19], we integrated predicted fusions using RNA-seq-based chimeric reads with WGS-based SV breakpoints. As anticipated, only a fraction of the chimeric-based fusions predictions involved an SV breakpoint falling within the boundary of one or both genes. Out of 9459 candidate fusion events by RNA-seq, 3419 involved SV breakpoints (Fig. 3a). Most of these 3419 events involved the over-expression of one or both genes at the mRNA level in the impacted sample, with a fraction of these events also showing protein over-expression (Fig. 3b). Protein data were available for 2844 candidate fusion events out of the 3419 with WGS support, and 1098 of these involved both protein and mRNA over-expression, while 886 had mRNA but not protein expression. Of all chimeric-based fusion events, those with supporting evidence by both SV and gene over-expression were highly enriched for previously identified gene fusions in cancer[43,44] (Fig. 3c). The set of 1098 fusion calls with combined WGS and both protein and mRNA expression support involved 1055 distinct gene fusions, 350 tumors, and 346 patients (Fig. 3d and Supplementary Dataset 4). Another 386 fusions had mRNA support where protein data were not available. These fusions involved those common to pediatric brain tumors[19], including *KIAA1549-BRAF* ($n = 42$ tumors) and *C11orf95-RELA* ($n = 7$). Other fusions included *EML4-ALK* ($n = 5$ tumors), *FGFR3-TACC3* ($n = 8$)[45], *ESR1-CCDC170* ($n = 4$)[46], *EGFR-SEPTIN14* ($n = 4$)[47], *RPS6KB1-VMP1* ($n = 3$)[48], and *EGFR-SEC61G* ($n = 3$)[49]. While in theory, gene fusions contributing to tumorigenesis would not necessarily need to involve altered gene expression, we find here that many

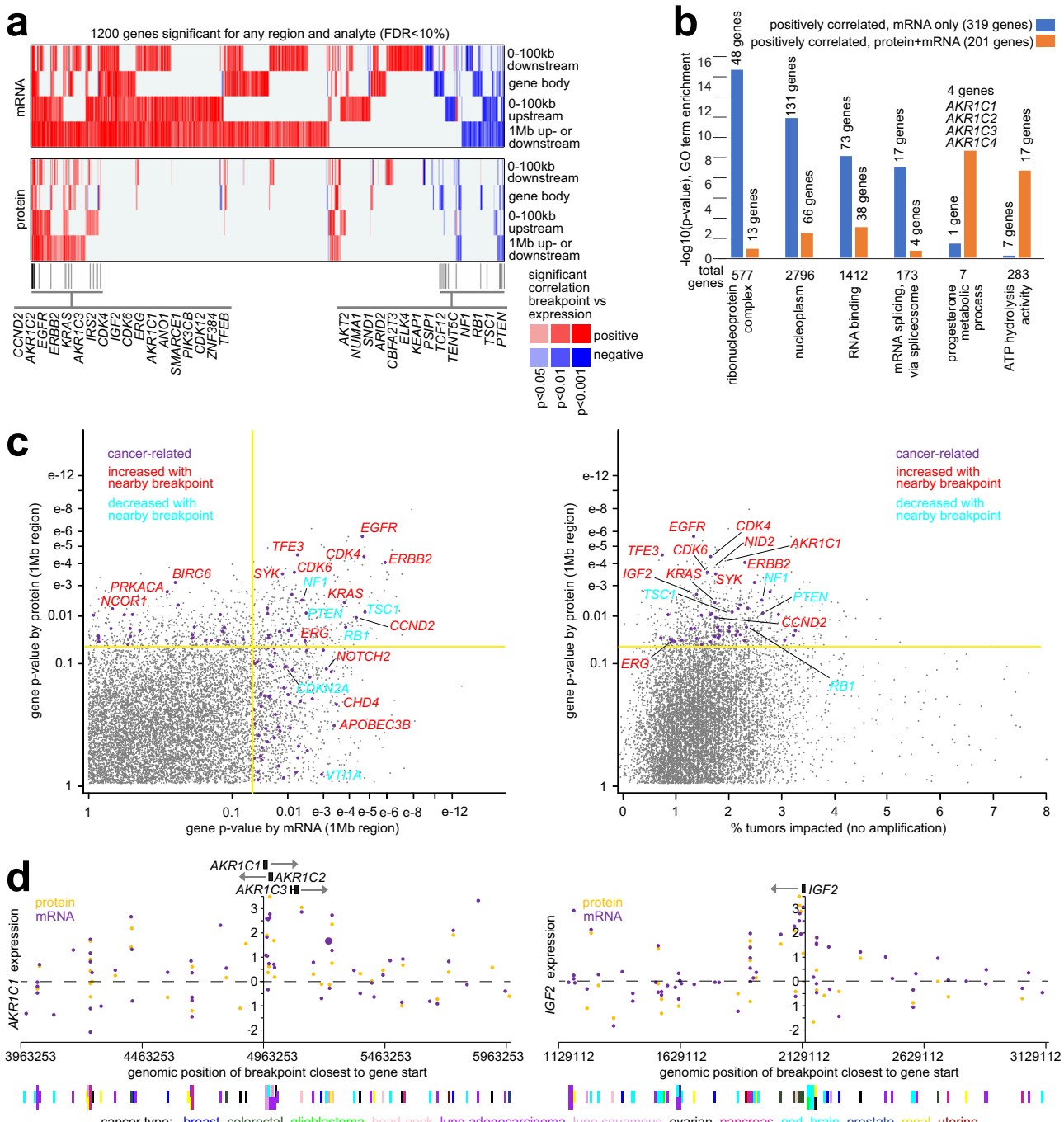

**Fig. 2 | Genes with altered protein or mRNA expression associated with nearby somatic SV breakpoints. a** Heatmap of significance patterns for 1200 genes associated with altered expression in conjunction with nearby SV breakpoints, at either the mRNA or protein levels (FDR < 10%), for any genomic region window examined (involving SV breakpoints 100 kb upstream of the gene, 100 kb downstream of the gene, within the gene body, or 1 MB upstream or downstream of the gene, as indicated). Red denotes a significant positive correlation; blue, significant negative correlation. Gene-level significance results at the protein level are represented alongside results at the mRNA level. **b** For the top genes with SV-associated over-expression (for the 1 MB region window) at both protein and mRNA levels (blue, *p* < 0.05 for both, *p* < 0.01 for either) or at the mRNA but not protein level (orange, mRNA *p* < 0.01, protein *p* > 0.05), represented categories by GO were assessed, with selected enriched categories represented here. *P*-values by one-sided Fisher's exact test. **c** Significance of SV-impacted genes at the protein level (involving breakpoints within 1 Mb of the gene), as plotted (y-axis) versus the

significance at the mRNA level (x-axis, left) and versus the number of tumors impacted (x-axis, right, expression >0.4 SD from sample median, not including tumors with gene amplification for the positive SV-associations). "Cancer-related," by COSMIC[43]. **d** Protein (orange) and mRNA (purple) expression levels of *AKR1C1* (left) and *IGF2* (right), corresponding to SV breakpoints located in the genomic region 1 Mb downstream to 1 Mb upstream of the respective gene. Each point represents a single tumor (the closest SV breakpoint being represented for each tumor). Breakpoints near the respective gene tend to be associated with its higher expression (e.g., above the sample median, indicated by the horizontal dashed line). Only one tumor represented involved gene amplification (represented by the larger data point for *AKR1C1*). In parts **a**–**c**, SV-expression association *p*-values correct for cancer type and CNA by linear modeling. For parts **b** and **c**, mRNA results are based on the mRNA dataset filtered for any expression data values not represented in the protein dataset.

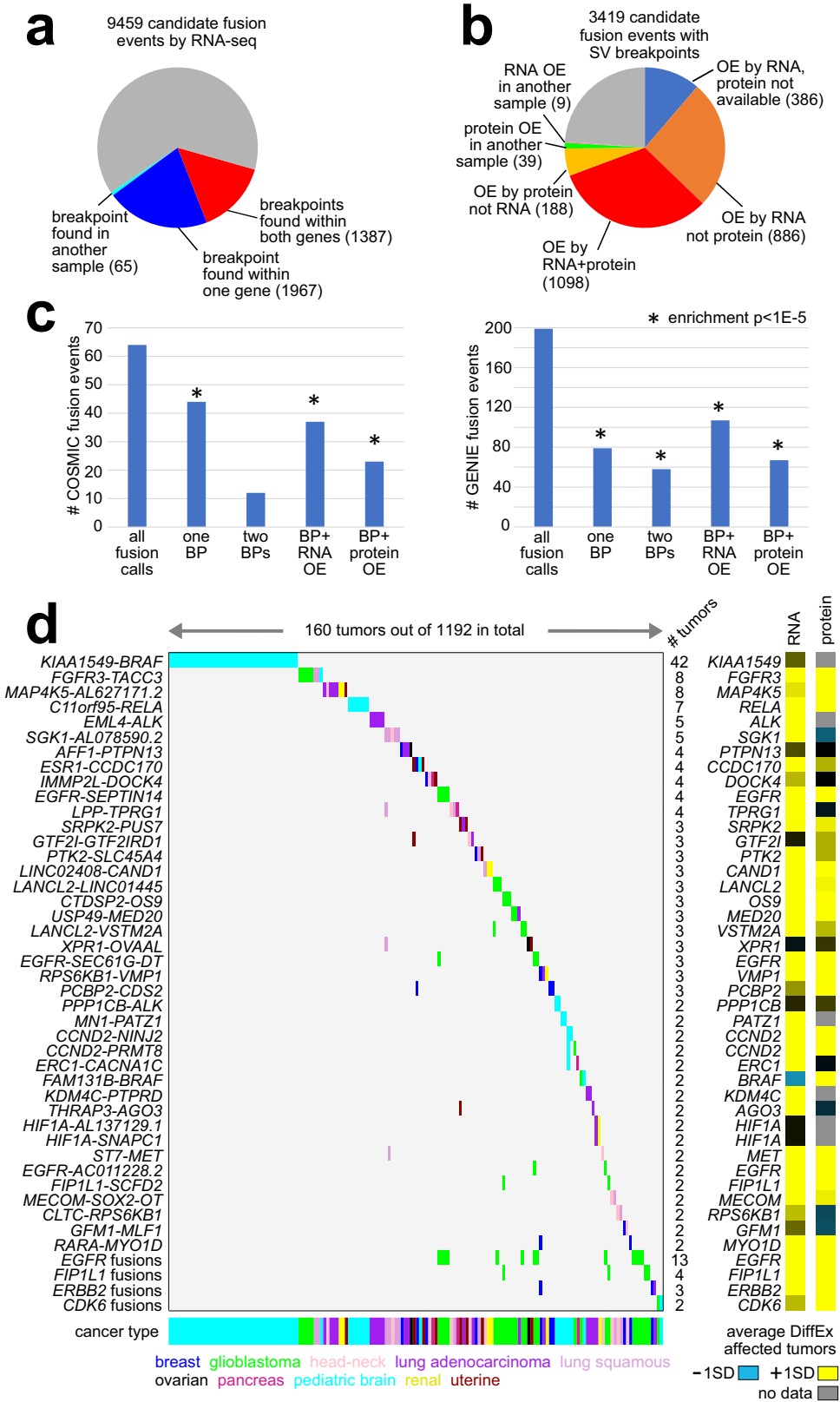

**d**  160 tumors out of 1192 in total

cancer type

breast  glioblastoma  head-neck  lung adenocarcinoma  lung squamous
ovarian  pancreas  pediatric brain  renal  uterine

average DiffEx
affected tumors

−1SD    +1SD

no data

predicted fusions involving protein over-expression include fusions observed elsewhere using RNA data.

## SVs involved with pathway alterations
We found somatic SVs and associated altered protein levels to considerably extend the numbers of patient tumors somatically altered for critical pathways. Taking a set of cancer-associated pathways and related genes previously annotated based on domain knowledge[19,35], we examined the tumors in our WGS-expression compendium cohort ($n = 1426$ tumors) for alteration in these pathways. Alterations considered were gene fusion, SV-associated altered up-regulation or gene disruption (by protein or alternatively by mRNA if protein data not

**Fig. 3 | Identification of gene fusion events by combined RNA-seq, WGS, and proteomic analyses. a** Out of 9459 candidate fusion events identified by RNA-seq chimeric reads (across the 1192 tumors with proteomic data+RNA-seq-based fusion calls), the numbers of events with support from SV breakpoint analysis by WGS. As indicated, for 3354 candidate fusion events, SV breakpoints were found within one or both genes, with or without a high expression association (see Methods). An additional 65 events involved a fusion with both RNA-seq and WGS support in another tumor. **b** Of the 3419 gene body SV breakpoint events involving predicted fusions from part **a**, the fractions of events associated with gene over-expression ("OE") by both mRNA and protein, by mRNA but not protein, by mRNA with no protein data available, by protein but not mRNA, or by mRNA or protein in another tumor (see Methods). **c** Out of 9459 candidate fusion events identified by RNA-seq chimeric reads, the numbers of events involving a COSMIC[43] fusion (left) or a fusion identified in the GENIE dataset (v12) by DNA sequencing[44]. Of the candidate fusion

events involving COSMIC or GENIE, the numbers of events involving a within-gene SV breakpoint ("BP") by WGS (for one or both genes) and the number of events involving SV breakpoint with mRNA or protein over-expression ("OE") are indicated. Enrichment *p*-values by one-sided Fisher's exact test. **d** Gene fusion events with chimeric RNA-seq reads, SV support, and corresponding high protein or high mRNA levels, involving either greater than two tumors or two tumors plus a COSMIC gene, are represented as a data matrix, each colored entry representing a fusion event meeting the criteria from part **b**. Cancer type is indicated along the bottom and in the coloring of the fusion event. To the right is indicated the average differential expression ("DiffEx," at mRNA and protein levels, yellow denoting higher expression) for fusion-involved genes in the affected tumors. See Supplementary Dataset 4 for the entire set of the 9459 candidate fusion events from part **a**, with the corresponding filtering criteria used to derive the fusion events with SV and expression support.

available), SNV or indel, and deep deletion or high-level amplification (Supplementary Dataset 1). Across different cancer types, SV-associated alterations (Fig. 4a and Supplementary Dataset 5) involved Receptor Tyrosine Kinase (RTK) pathway-related genes (*EGFR*, *ERBB2*, *FGFR2*, *FGFR3*, *KRAS*, *NF1*), p53/Rb-related genes (*CCND1*, *CCNE1*, *CDK4*, *CDKN2A*, *E2F3*, *RB1*), *TERT*, MYC family genes (*MYC*, *MYB*, *MYCN*), and mTOR pathway-related genes (*AKT1*, *PIK3CA*, *PTEN*, *STK11*). A pathway-level assessment for key genes (Fig. 4b, c and Supplementary Fig. 4) found a high number of SV-associated fusion or altered regulation events involving RTKs (138 tumors uniquely altered by SVs out of 710 tumors altered for the pathway by any gene or somatic alteration), p53 or Rb (62 out of 788 altered tumors), mTOR pathway (49 out of 614 altered tumors), TERT (48 out of 178 altered tumors), MYC family (65 out of 155 altered tumors), and NRF2 (17 out of 151 altered tumors). In the above pathway-level annotations, SV-associated alterations were only tabulated after other somatic alterations (SNV/indel or CNA) were not found for any genes. This aspect means that looking only within coding regions and not considering non-coding regions for somatic alterations would result in missing pathway alteration events for many patients.

### Impact of SVs on DNA methylation

Somatic SVs have been associated with recurrent alterations in DNA methylation of CpG Islands (CGIs)[5]. We can expect some of these DNA methylation changes to impact gene expression at the protein level. CPTAC tumors were profiled for DNA methylation in addition to WGS and gene expression (*n* = 988 tumors). At the DNA methylation level, a set of 2151 CGI probes associated with recurrently SV-altered DNA methylation were significant at FDR < 10% for any one of four gene region windows examined: 100 kb upstream, 100 kb downstream, within the gene, and 1 Mb upstream or downstream (Fig. 5a and Supplementary Dataset 6). In our CPTAC cohort, CGI probes with SV-associated increased methylation were predominantly promoter-associated, while CGI probes with SV-associated decreased methylation were enriched for gene body CGIs (Fig. 5b), entirely consistent with previous observations in TCGA cohort[5]. At the DNA methylation level, top SV-associated altered CGIs in our CPTAC cohort overlapped highly with the top CGIs previously identified using TCGA cohort[5] (Supplementary Fig. 5a). We could also define SV-associated DNA methylation alterations according to cancer type as defined by tissue of origin, including CGIs and associated genes that were not statistically significant in the pan-cancer analyses (Supplementary Fig. 5b, c and Supplementary Dataset 7).

For the genes associated with the SV-altered CGIs, only a fraction showed concordant associations with nearby SV breakpoints at the mRNA or protein levels, with an even smaller fraction of concordant proteins as compared to concordant mRNAs (Fig. 5a, c). We examined the overlap between CGI probes with SV-associated altered methylation and the related genes with corresponding SV-associated altered expression (in the inverse direction) at either

mRNA or protein levels (Fig. 5c). Using a *p*-value cutoff of <0.01 (1 Mb region, with cancer type and CNA corrections), 2602 CGI methylation probes were positively correlated with nearby SV breakpoints, of which 52 probes involved genes negatively correlated between mRNA expression and SV breakpoints (*p* < 0.01), a significant overlap (*p* < 1E−8, chi-squared test), with just 14 of the 52 probes also showing inverse correlation (*p* < 0.05) between protein and SV breakpoints across the 1307 tumors. The 14 CGI probes significant by protein included four for *PTEN*[5]. Out of 517 CGI methylation probes negatively correlated with nearby SV breakpoints, 50 involved genes positively correlated between mRNA and SV breakpoints (overlap *p* < 1E−4, chi-squared test), of which 21 CGI probes involved positive correlations at the protein level. Genes involving SV-associated lower CGI methylation with concordant protein and mRNA changes included *NID2*[50] and *ANO1*[51] (Fig. 5d).

### Mechanisms of SV-associated altered protein expression

Previously explored mechanisms at work in SV-altered gene deregulation[5,6,19,35] were examined here and found to involve altered protein expression patterns. In our protein-WGS cohort, TAD-disrupting SVs (with breakpoints spanning two different TADs) were significantly enriched among the SVs associated with gene up-regulation at the mRNA level (Fig. 6a, *p* < 1E−10, chi-squared test). Of the TAD-disrupting SVs associated with mRNA over-expression (with available protein data), roughly 66% involved protein over-expression (>0.4 SD from sample median). In addition, SV breakpoints involving mRNA over-expression were enriched (*p* < 1E−47) for potential enhancer hijacking events, where the rearrangement positioned an enhancer within 0.5 Mb of the gene (Fig. 6b, Supplementary Dataset 8), with roughly 64% of these events involving protein as well as mRNA over-expression. SV-mRNA over-expression associations were also enriched (*p* < 1E−13) for retrotransposon hijacking events, including long interspersed elements (LINEs) and short interspersed elements (SINEs), with most of these events involving protein over-expression (Fig. 6c, Supplementary Dataset 8). A phenomenon of the somatic rearrangement of regions with higher or lower methylation from other parts of the genome being linked with SV-associated DNA methylation alterations, previously observed in TCGA cohort[5], was confirmed here using the CPTAC cohort (Fig. 6d). Enhancer hijacking events involving protein over-expression (>0.4 SD from sample median) involved 199 tumors and 171 genes, 59 of these genes involving two or more tumors (Fig. 6e), including *EGFR* (5 tumors) and *CDK4* (3 tumors). Rearrangement of a region of low methylation near a gene, with corresponding decrease in methylation and increase in protein expression being observed, involved 151 genes and 151 tumors (Fig. 6f and Supplementary Dataset 8).

### SVs and associated genes involving patient survival

A subset of genes with SV-protein associations in our protein-WGS compendium cohort also had associations with patient overall survival

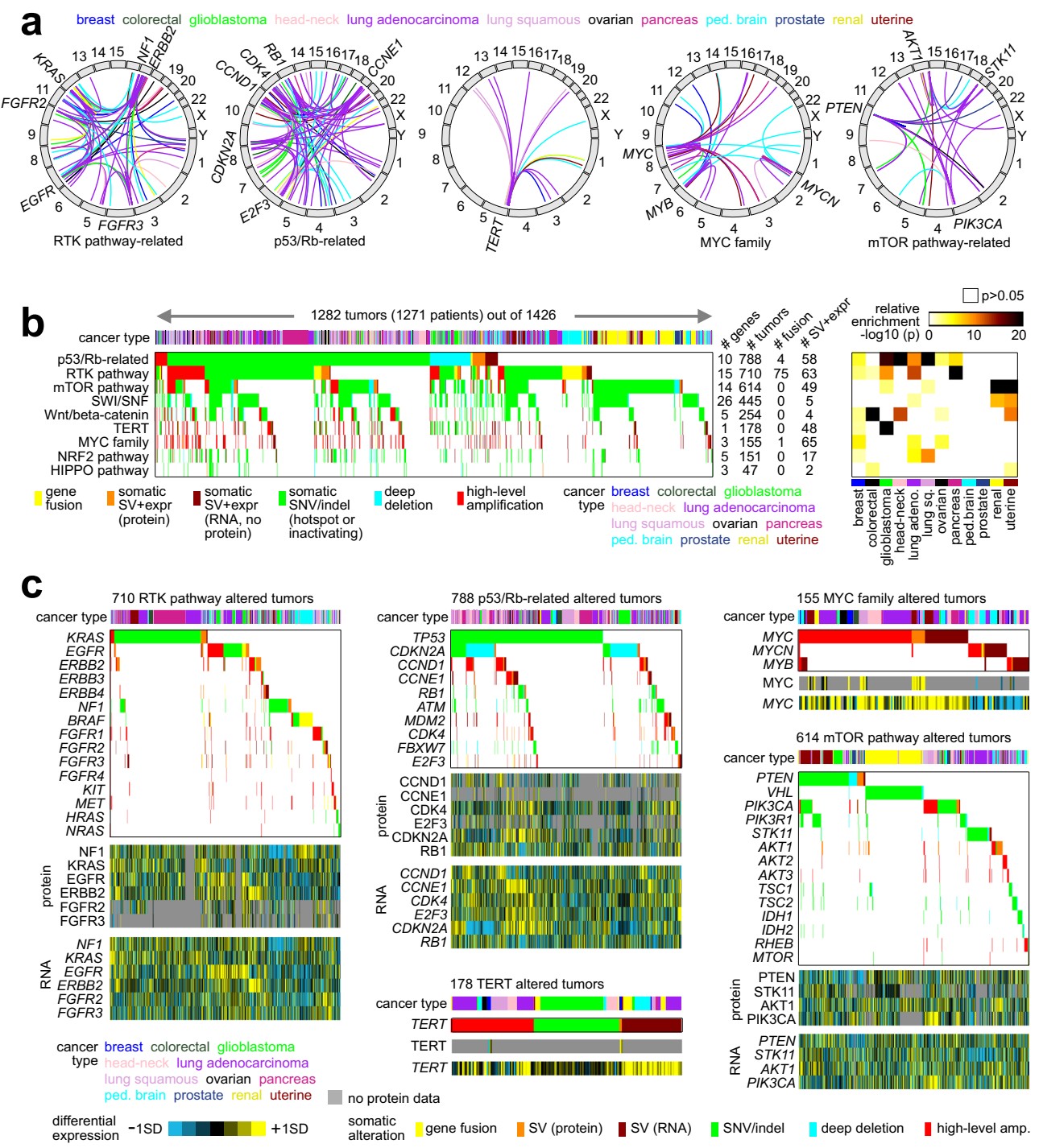

across an extended mRNA-WGS compendium cohort of over 3000 patients (Fig. 7a, Supplementary Dataset 1). Across the extended cohort, 3156 genes had somatic SV breakpoint patterns associated with worse survival (one-sided $p < 0.05$ stratified Cox analysis, Supplementary Dataset 9, see Methods), while 3476 genes had expression associated with worse survival (one-sided $p < 0.05$); the intersection between both gene sets was 679. When considering a set of 516 genes with positive SV-protein expression correlations ($p < 0.05$, correcting for cancer type and CNA), 34 were part of the 679 poor prognosis genes, representing a statistically significant overlap ($p = 0.01$, one-sided Fisher's exact test). Interestingly, only one gene, *PCBD2*, had the opposite patterns: a negative SV-expression association in our proteomics compendium dataset and both breakpoints and expression

associated with better overall survival in the extended cohort (Supplementary Dataset 9).

The above 34 genes, with both protein over-expression and worse patient survival associated with SVs, could collectively be associated with worse patient outcomes across multiple additional transcriptomic datasets involving various cancer types. We hypothesized that the 34-gene set would represent a broadly applicable, pan-cancer signature of worse patient outcomes. To test this, we first scored 10,224 tumors in TCGA pan-cancer cohort based on the entire 34-gene signature (taking the average of the normalized expression values in each tumor profile). In TCGA, the signature was associated with worse overall survival (Fig. 7b). Similarly, the 34 gene signature was associated with worse prognosis across additional transcriptomic datasets

**Fig. 4 | Somatic SVs associated with key oncogenic or tumor-suppressive pathways. a** Genomic rearrangements (represented in circos plots) involving altered expression Receptor Tyrosine Kinase (RTK) pathway-related genes (*KRAS, EGFR, ERBB2, FGFR2, FGFR3, NF1*), p53/Rb-related genes (*CCND1, CCNE1, CDK4, CDKN2A, E2F3, RB1*), *TERT*, MYC family genes (*MYC, MYB, MYCN*), and mTOR pathway-related genes (*AKT1, PIK3CA, PTEN, STK11*). SV events are colored according to cancer type. **b** Pathway-centric view of somatic alterations across 1426 human tumors, involving key pathways and genes previously annotated across multiple cancer types based on domain knowledge. Of the 1426 tumors, 1282 (representing 1271 patients) had at least one somatic alteration in the indicated pathways. The panel on the right represents the significance of enrichment (one-sided Fisher's exact test) of gene alteration events for each pathway within a given cancer type versus the rest of the tumors. **c** For the pathways from part **b** that also involve at least one SV event, somatic alteration events involving each gene included in the pathway are represented. For SV-impacted genes, the corresponding differential protein and mRNA expression patterns are shown. For parts **a**–**c**, SV events represent altered gene expression (by protein or alternatively by mRNA if protein data not available), defined for oncogenes as breakpoint falling with 1 Mb of gene and associated with expression >0.4 SD from the median for the given tumor, and defined for tumor suppressors as breakpoint falling within the gene and expression < −0.4 SD. For parts **a** and **b**, events are colored according to the type of somatic alteration: gene fusion, SV with altered expression (by protein or alternatively by mRNA if protein data not available), SNV or indel (for oncogenes, SNV within hotspot residue; for tumor suppressor genes, SNV within hotspot residue or inactivating mutation by indel/nonsense/nonstop), and deep deletion or high-level amplification (respectively approximating total copy loss and copy levels more than 2X greater than that of wild-type, based on thresholded values). All analyses are based on the 1426 tumors with combined WGS and expression data (by protein or, if protein data not available, by mRNA). Expr., expression; lung adeno., lung adenocarcinoma; lung sq., lung squamous; amp., amplication.

for lung adenocarcinoma, breast cancer, pediatric brain tumors, prostate cancer, and bladder cancer (Fig. 7c and Supplementary Fig. 6). Genes in the 34-gene signature included *G6PD* (glucose-6-phosphate dehydrogenase), which affects tumor development by regulating several metabolic pathways[52]. For *G6PD*, nearby SV breakpoints were associated with increased protein expression (Fig. 7d). *G6PD* also showed associations with worse patient survival in terms of both breakpoint patterns and expression (Fig. 7e and Supplementary Fig. 6 and Supplementary Dataset 9).

### SV-altered cell lines sensitive to gene knockout

We hypothesized that genes over-expressed in relation to nearby SV breakpoints represent dependencies in the cancer cells. Taking the set of 228 genes with SV-expression associations for both protein and mRNA ($p < 0.05$, 1 Mb region, correcting for cancer type and CNA), these genes were enriched within genes having high variability in gene dependencies across cancer cell lines, based on DepMap CRISPR assay[53] (Fig. 8a). For genes with high variability in gene effect scores, only a fraction of cell lines would presumably be sensitive to CRISPR knockout of those genes, as opposed to "essential" genes for which knockout could be deleterious for most or all cell types, including non-cancer cells. Similarly, when surveying combined protein expression and SV data across 328 cell lines[38,54,55], we observed a highly significant overlap ($p < 1E{-}100$ chi-square test) of events involving CRISPR knockout effect in genes with events involving SV-associated protein over-expression (Fig. 8b), involving 2682 genes, 178 with combined breakpoint and protein over-expression involving at least 5% of cell lines examined (Supplementary Dataset 10). When crossing the 2682 genes with genes having positive SV-expression association in human tumors (by both protein and mRNA) in our 1426-tumor cohort, a shorter list of 33 genes involved seven or more cell lines (Fig. 8b). These 33 genes include known cancer genes such as *KDM6A, CDK2*, and *CCND1*, for which CRISPR knockout show increased sensitivity in cell lines with combined SV breakpoint and gene over-expression (Fig. 8c).

## Discussion

Protein data, in addition to gene transcription data, can add another dimension in assessing the relevance of non-coding somatic alterations in cancer. Gene transcription data have previously helped demonstrate the functional impact of non-coding somatic alterations on nearby genes, including *TERT*, for which gene both recurrent point mutations and somatic SVs associate with increased expression[4,8,56,57]. Combined WGS and MS-based proteomic data on appreciable numbers of human tumors have only recently been available in the public domain. Here, we found that only about 25% of the genes associated with somatic SV-associated cis-regulatory alterations at the mRNA level were similarly associated at the protein level. Limitations to our findings would entail those involving the MS-based proteomics platform, including challenges in detecting less abundant proteins in particular[58]; however, even when accounting for this, the overall correspondence levels between mRNA- and protein-based associations are similar to the above. For an appreciable number of genes in our proteomics compendium dataset, protein expression was not detectable in most tumors, TERT being the more notable example. Nevertheless, for most genes, proteomics can be effectively leveraged to determine which gene expression alterations broadly observed at the mRNA level are carried over to the protein level. After assembling a catalog of combined SV-mRNA and SV-protein associations, our study could identify associations of particular interest when considering genes with well-established cancer roles, DNA methylation associations, gene-specific events involving fusions, enhancer hijacking or retrotransposon transposition, patient survival associations, and associations with cell line viability.

While protein levels broadly correlate with corresponding mRNA levels across human tumors[13], most of these statistically significant correlations would not be particularly strong in terms of predictability as assessed using r-values. In this present study, we focused on expression outliers associated with SV breakpoints. In the absence of strong protein-mRNA correlations, an expression outlier in one analyte may not show up similarly in the other analyte. Genes with weaker protein-mRNA correlations across tumors are reflected in the genes with SV-expression associations at the mRNA level but not at the protein level. Still, the overlapping gene-level associations by both mRNA and protein, while representing a fraction of the SV-mRNA associations, are highly statistically significant, drawing greater attention to these genes over genes significant by mRNA analysis alone. In comparing mRNA-based and protein-based results, we could use more relaxed *p*-value cutoffs for each individual analysis to limit false negatives. In this way, the integrative analysis approach offers an advantage in that, even when using a nominal *p*-value cutoff, the observation of consistent associations across multiple analytes helps identify additional genes of interest. Any observed discordances between protein and mRNA in our study can stem partly from biology or false positives or negatives, where the meaningful interpretation of a null *p*-value is inherently difficult.

Our present study involving a combined WGS and expression compendium dataset confirmed the overall observations on SV-altered expression patterns made previously using PCAWG[4], TCGA[5,6], CBTN[19], and POG570[35] WGS datasets. These observations included hundreds of genes with altered expression recurrently associated with nearby somatic SV breakpoints, DNA methylation changes associated with SV breakpoints, and mechanisms involving altered cis-regulation including enhancer hijacking and translocation of retrotransposons. In addition, here, we could identify SV breakpoint patterns and corresponding aberrantly expression genes associated with patient survival and genes with nearby SV breakpoints associated with increased cell dependency in cancer cell lines. This study's use of cell line data indicates that SV-associated over-expressed genes could represent unique

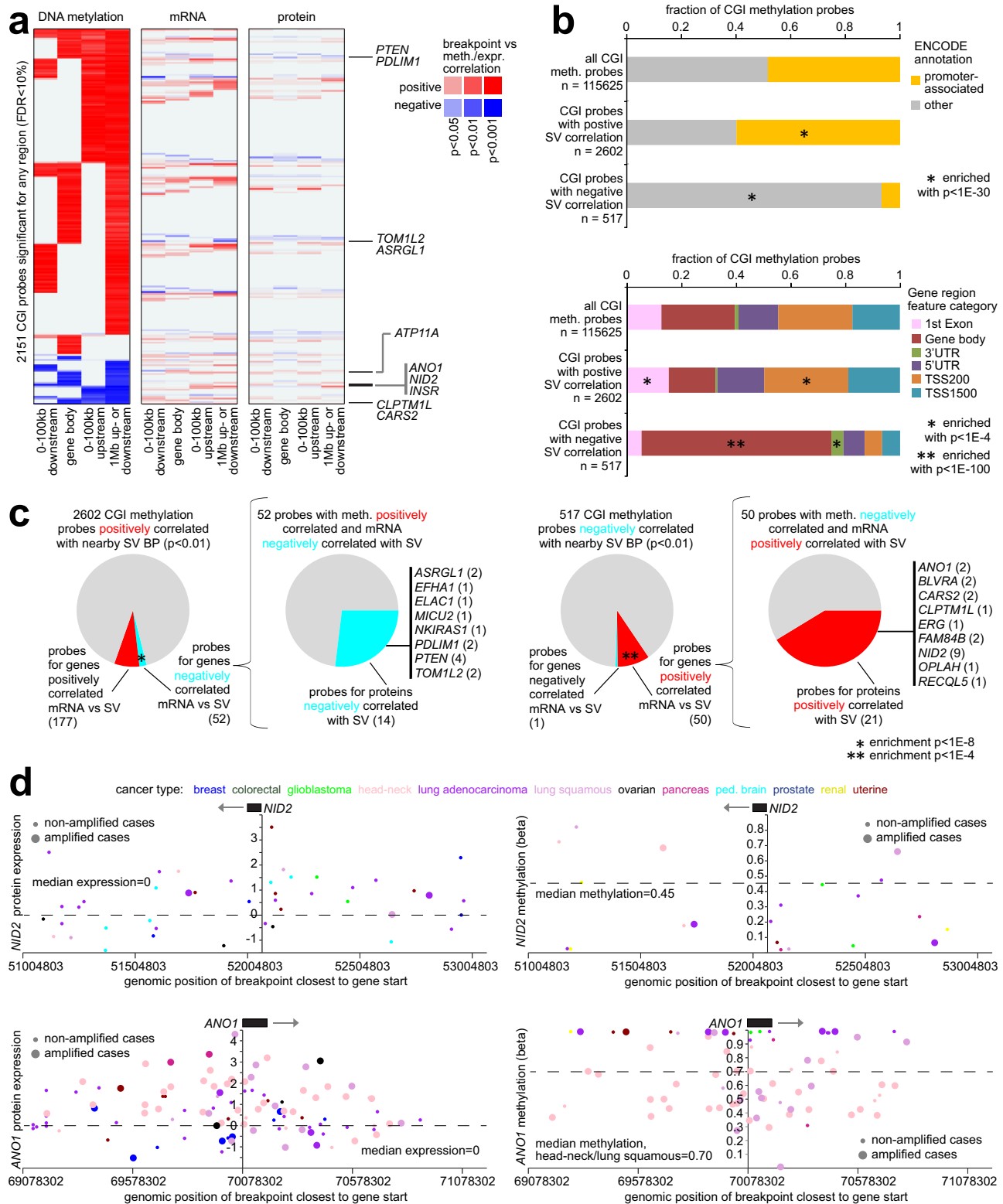

vulnerabilities in cancer cells. The top SV-altered genes uncovered here would overlap highly with the top genes from previous studies. At the same time, SV breakpoint patterns in relation to genes represent sparse events in cancer genomic data, whereby for a given gene typically <4% of tumors may have nearby SV breakpoints uninvolved with amplification. Therefore, future tumor and cell line datasets involving greater sample numbers would considerably aid in better establishing the catalog of recurrently altered genes, with proteomic data representing an important component of this aim.

Somatic SVs and associated altered protein levels would considerably extend the numbers of patient tumors altered for critical cancer-associated pathways, which would have implications for personalized or precision medicine approaches. There are also non-coding somatic point mutations associated with gene up-regulation, most notably for the TERT gene. The PCAWG consortium surveyed recurrent non-coding driver SNVs[3], but they found very few genes besides *TERT* impacted in this way that would represent drivers. In contrast, SV-associated cis-regulatory alterations do not need to be as

**Fig. 5 | Genes with altered DNA methylation and concordant expression associated with nearby somatic SV breakpoints. a** Heatmap of significance patterns for 2151 CGI probes associated with SV-altered DNA methylation (FDR < 10%, with at least one tumor with SV breakpoint involving methylation beta value difference>0.2 from sample median), for any genomic region window examined (involving SV breakpoints 100 kb upstream of the gene, 100 kb downstream of the gene, within the gene body, or 1 MB upstream or downstream of the gene, as indicated). Red denotes significant positive correlation; blue, negative correlation. The corresponding significance results for the CGI-associated genes are represented at both mRNA and protein levels. Gene listed off to the right were significant for methylation, protein, and mRNA (p < 0.05 for each) in concordant directions (high methylation and low expression or vice versa). **b** Top: Fraction of promoter-associated CGIs for the CGIs respectively associated with increased or decreased methylation (FDR < 10%, any region from part **a**). P-values by chi-square test. Bottom: Breakdown by probe position relative to the gene for the CGIs associated with increased or decreased methylation, respectively. P-values by chi-square test. TSS,

transcription start site; UTR, untranslated region. **c** Overlap between CGI probes with SV-associated altered methylation and nearby genes with corresponding SV-associated altered expression. For mRNA- and DNA methylation-SV breakpoint associations inverse to each other for the same genes (e.g., mRNA positively correlated and DNA methylation negatively correlated with nearby breakpoint, using p < 0.01, with at least one tumor with SV breakpoint involving methylation beta value difference>0.2 from sample median), the subset of CGI methylation probes for which the associated genes have protein-SV associations (p < 0.05) are indicated. P-values for significance of overlap by chi-squared test. In parts **a**–**c**, SV-expression and SV-methylation association p-values correct for cancer type and CNA. **d** As examples of significant genes, gene expression levels (left) and DNA methylation levels (right) of NID2 (top) and ANO1 (bottom), corresponding to somatic SV breakpoints located in the genomic region 1 Mb surrounding the gene. Each point represents a single tumor (closest SV breakpoint represented for each tumor). Breakpoints near the respective genes tend to associate with higher expression and lower methylation.

precise as for SNV hotspots, where SV breakpoints involving over-expression may fall at different locations with respect to the gene across the impacted tumors, with different mechanisms of SV-mediated cis-regulator alterations potentially being involved[5,6]. For the individual genes with SV-associated altered expression across tumors, the associations could not be accounted for by CNA patterns alone. When considering tumors with SV-altered protein expression but no amplification for a given gene, 1 to 3% of tumors conservatively show altered cis-regulation for key oncogenes. The patients involved would be considerable if such numbers apply to all human cancers. When grouping genes according to annotated pathways, SV-altered gene expression greatly extended the numbers of tumors altered for pathways beyond small mutation or CNA, involving an additional ~5% of tumors altered for several pathways. These additional altered tumors would presumably have not been identified using exome-centric approaches[12]. In light of this, precision medicine approaches utilizing focused DNA sequencing of specific genes might be improved by incorporating gene expression data[59], which could capture cis-altered gene regulation not involving CNA. The catalog of SV-altered genes, provided by studies such as ours, could greatly inform on better identifying patients with tumors altered for targetable pathways in the clinical setting.

## Methods

### Patient cohorts

The results here are based upon data generated by the Clinical Proteomic Tumor Analysis Consortium (CPTAC), the Applied Proteogenomic OrganizationaL Learning and Outcomes (APOLLO) research network, the Children's Brain Tumor Network (CBTN), the Cancer Genome Atlas (TCGA) Research Network, and the International Cancer Research Consortium (ICGC). We assembled a compendium dataset of combined WGS and gene expression data (by protein or RNA) on 1426 human tumors (representing 1409 patients). Combined WGS and mass spectrometry-based proteomic profiling was compiled for 1307 tumors in total, representing 1290 patients (Supplementary Dataset 1). The cancer types represented in the proteomics compendium dataset were the following: Breast Invasive Carcinoma (n = 18 tumors with proteomics data)[15], Colorectal Adenocarcinoma (n = 46)[16], Glioblastoma (n = 98)[22], Head and Neck Squamous Cell Carcinoma (n = 108)[24], Lung Adenocarcinoma (n = 197)[21,27], Lung Squamous Cell Carcinoma (n = 108)[23], Ovarian Serous Cystadenocarcinoma (n = 15)[14], Pancreatic Ductal Adenocarcinoma (n = 136)[25], Pediatric Brain Tumors (n = 219)[26,29], Prostate Adenocarcinoma (n = 47)[28], Renal Cell Carcinoma (n = 219)[18,20], and Uterine Corpus Endometrial Carcinoma (n = 99)[30]. Combined WGS and RNA-seq profiling was compiled for 1413 tumors in total, of which 1294 had proteomic data and of which 988 had DNA methylation data (Illumina MethylationEPIC platform) not analyzed by our group previously[5]. For 118 lung adenocarcinomas

in CPTAC Confirmatory cohort with combined WGS, RNA-seq, and DNA methylation data, no proteomics data were made publicly available at the time of our study. A subset of CBTN tumors with proteomic data represented multiple tumors taken from the same patient, involving 34 tumors from 17 patients (two tumors/patient). Different tumors from the same patient could demonstrate extensive molecular heterogeneity with respect to each other[19,60]. Therefore, each tumor sample was analyzed independently in the integrative analyses. Sex was not considered in the study design, but all tumors with available data were incorporated into the study.

### Somatic structural variant (SV) datasets

Somatic SV calls were compiled from the following sources, as noted for each tumor sample in Supplementary Dataset 1: from the Genome Data Commons (GDC), using the BRASS software package[61]; from the PCAWG consortium of calls made by two or more algorithms[11]; the CBTN Cavatica portal, using the Manta algorithm[19]; SV calls on TCGA tumors based on high pass WGS (~30–60x coverage) from Zhang et al. using Meerkat algorithm[5]; SV calls on TCGA tumors based on low pass WGS (~6–8x coverage) from Zhang et al.[6] using Meerkat algorithm; and the Manta SV calls from the GDC publication page for the APOLLO lung study[21]. All but 24 of the 1426 tumors involved in the study had SV calls by high-pass WGS. Our previous studies have collectively found that the phenomenon of SV-altered gene regulation may be observed independently of the SV calling algorithm used[4–6,19,35,38].

### Proteomic datasets

We assembled a compendium dataset of mass spectrometry-based proteomics data of primary tumors from previous studies (Supplementary Dataset 1)[14–16,20–30]. Of the 1307 tumors with proteomic data, 1155 were from CPTAC-led studies, 87 were from the APOLLO-led adenocarcinoma study[21], 47 were from a prostate cancer study[28], and 18 were from a medulloblastoma study[29]. The above studies analyzed the tumors using global proteomic and phosphoproteomic profiling by liquid chromatography-tandem mass spectrometry (LC-MS/MS). We obtained processed protein expression data from the CPTAC Data Portal[62], the Protein Data Commons [https://pdc.cancer.gov/pdc/], or the associated publications' supplementary datasets. Proteomic data, as collected from the public domain, were processed at the gene level rather than at the protein isoform level; as a simplification, we did not consider different isoforms for the same protein in the present study. We used non-imputed versions of the proteomics datasets, as our integrative SV-expression analyses aimed to identify expression outliers across tumors, which, by definition, could not be imputed.

For each study, taking the logged expression values provided in the associated data table, we normalized proteomic data for

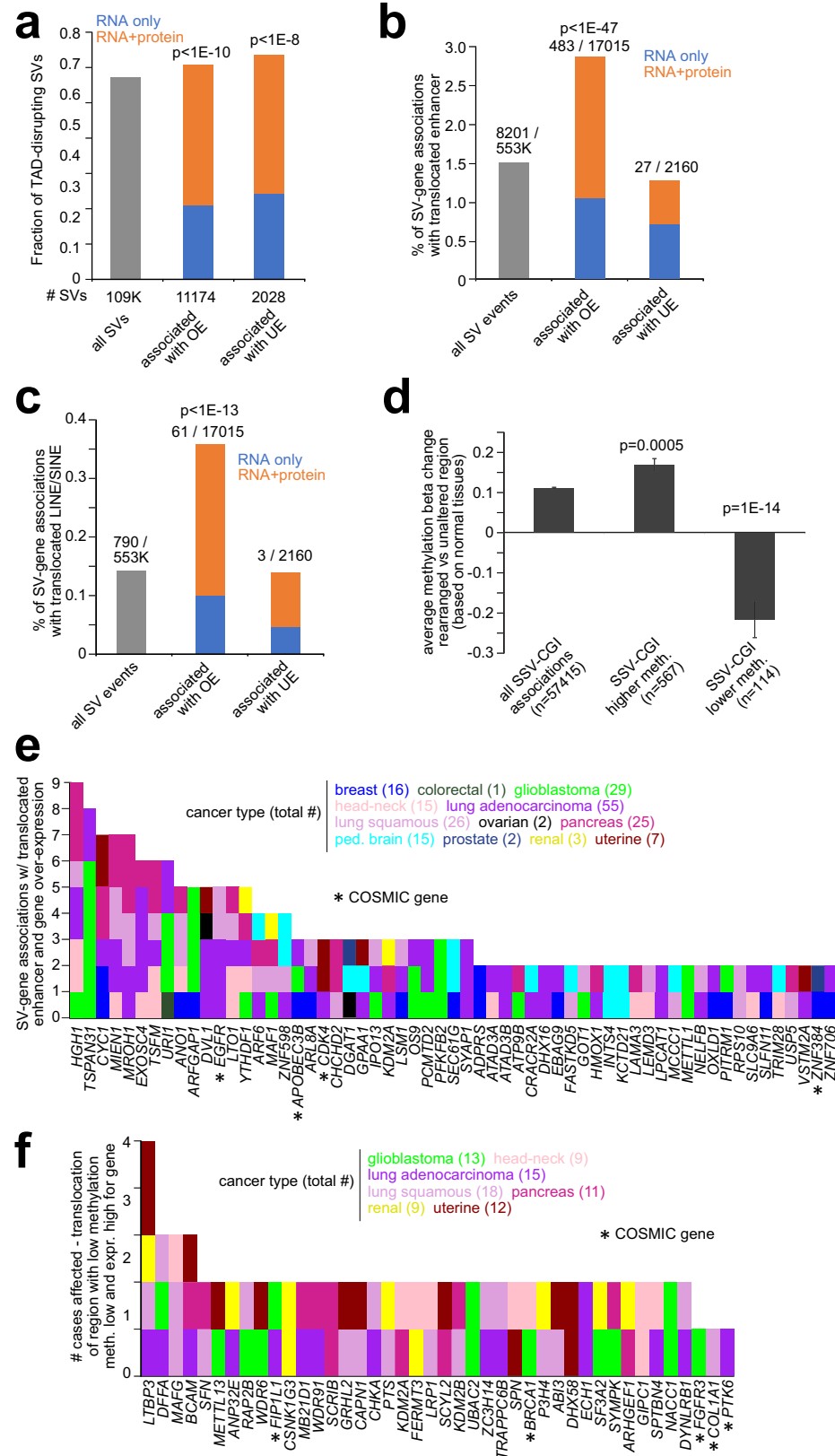

downstream analyses in the following manner and as described previously[13,34,40,42]. For the profiles taken from the CPTAC data portal, we first normalized expression values to standard deviations from the median within each proteomic profile. We then normalized expression values across samples to standard deviations from the median for all profiles. Similarly, we separately normalized both total protein and

phospho-protein datasets for a given cancer type and dataset. For datasets where two different data centers generated values on the same tumors, we averaged normalized values from the respective data centers in instances of duplicate profiles for the same tumor sample. As intended, by normalizing expression within each cancer type and each proteomic dataset, neither tissue-dominant differences nor inter-

**Fig. 6 | Mechanisms of SV-associated altered protein expression. a** As compared to all somatic SVs, fractions of SVs involving TAD disruption and altered gene expression. **b** Percentages of SV breakpoint associations involving an enhancer within 0.5 Mb of the SV breakpoint near the gene (and closer than any enhancer within 1 Mb of the unaltered gene), as tabulated for the entire set of SV breakpoint associations with breakpoint mate on the distal side from the gene, as well as for the subsets of SV breakpoint associations involving altered gene expression. **c** Percentages of SV breakpoint associations involving the translocation of a LINE or SINE retrotransposon within 20 kb of the SV breakpoint near the gene, for all SV breakpoint associations and the subsets of SV breakpoint associations involving altered gene expression. For parts **a–c**, SV-associated altered expression is defined as $p < 0.01$ by mRNA analysis (1 Mb region) and expression >0.4 SD or <−4 SD from the median for the tumor harboring the breakpoint. The SVs or SV-gene associations that would involve altered gene expression and that are represented in the figures have both mRNA and protein data available. $P$-values by chi-square test. SV-expression association $p$-values correct for cancer type and CNA. **d** Average DNA methylation different represented by the rearranged region compared with that of the CGI nearby the gene, with the average difference in methylation beta values computed for all SV-CGI associations and the subset of SV-CGI associations involving higher or lower DNA methylation ($p < 0.01$). $P$-values by Mann-Whitney $U$ test. Error bars represent standard error. **e** By gene and by cancer type, the number of SV breakpoint associations involving the translocation of an enhancer with altered protein expression for at least two tumors for each gene (from part **b**), which involved 59 genes and 141 tumors. **f** By gene and by cancer type, the number of tumors involving the rearrangement of a region of low methylation (average methylation beta difference < −0.1), with corresponding decrease in methylation and increase in protein expression observed (<− 4 SD and >0.4 SD from the median, respectively), involving 42 genes and 75 tumors (genes cancer-associated[43] or affected in >2 tumors). OE, over-expression; UE, under-expression.

laboratory batch effects would drive the downstream analysis results. Our SV-expression analytical approach was intended to identify expression outliers involving a small fraction of sample profiles, which outliers would remain in the data after normalization. The compendium dataset of total proteins included the 15,439 unique genes by Entrez Identifier that were represented in our previous proteomics compendium of 2002 tumors[13]. For the compendium dataset of phospho-proteins, a total of 199,284 phospho-proteins, involving 11,671 unique genes, were represented in at least one of the individual datasets. Of these phospho-proteins, 8835 had available data for at least 400 tumors, which features were analyzed for SV breakpoint versus phospho-protein associations.

### Transcriptomic datasets
For CPTAC projects utilizing tumors from TCGA, we obtained TCGA data RNA-seq data from the Broad Institute's Firehose data portal [https://gdac.broadinstitute.org], and we obtained RNA-seq data for the other CPTAC projects from the GDC [https://gdc.cancer.gov/]. We obtained RNA-seq data on CBTN pediatric brain tumors through the public project on the Kids First Data Resource Portal and Cavatica [https://cbtn.org]. We obtained data for the non-CPTAC projects from links or accession numbers provided with the associated publications. We normalized expression values across samples to standard deviations from the median within each cancer type and dataset, as we carried out above for proteomic data.

### Copy number alteration (CNA) datasets
All but one tumor in our study had corresponding gene-level CNA data (the one exception being a TCGA sample with SV by low pass WGS but no SNP array data). For CPTAC projects utilizing tumors from TCGA, we obtained SNP array-based CNA "thresholded" values (-2, -1, 0, 1, 2) from the Broad Institute's Firehose data portal [https://gdac.broadinstitute.org]. We obtained gene-level absolute copy data (0, 1, 2, 3, 4, 5) for the other CPTAC projects and the APOLLO dataset from the Genome Data Commons [https://gdc.cancer.gov/]. We first normalized the absolute copy data according to ploidy (dividing gene copy value by average copy value for all genes), then thresholded to values approximating homozygous deletion (-2), heterozygous deletion (-1), wild-type (0), gain of 1-2 copies (+1), and amplification with at least 5 copies (2). We obtained gene-level copy data from CBTN pediatric brain tumors from Cavatica [https://cbtn.org] and thresholded similarly to the CPTAC copy data. For the prostate cancer and medulloblastoma datasets, gene-level copy data based on WGS data were previously generated by the Pan-cancer Analysis of Whole Genomes (PCAWG) consortium from a consensus of multiple CNA callers[63]. For the APOLLO cohort, gene-level log(tumor-normal) values were obtained from the GDC publication page and then thresholded to amplification, gain, wild-type, loss, or deletion.

### DNA methylation datasets
For 988 tumors, DNA methylation profiles had been generated by CPTAC using the Illumina Infinium MethylationEPIC BeadChip array platform (Illumina, San Diego, CA), with level3 beta values being made available via the Genome Data Commons. Our study focused on the 115,625 array probes falling within CGIs that did not involves X or Y chromosomes (these chromosomes not being included as these would be present or not present or differentially methylated according to patient gender).

### Small mutation datasets
All but five tumors in our study had small somatic mutation calls (SNVs and indels) by whole-exome or whole-genome sequencing. We used whole-exome sequencing (WES) data over WGS for small mutation calls when available for a given tumor. For CPTAC projects utilizing tumors from TCGA, we obtained somatic mutation calls by WES from the publicly-available "MC3" TCGA MAF file [https://www.synapse.org/#!Synapse:syn7214402]; variants called by two or more algorithms were used in this study. CBTN used both Strelka2 and Mutect2 to call small somatic mutation, based on WGS[19]. We assessed the somatic variant MAFs through the public project on the Kids First Data Resource Portal and Cavatica [https://cbtn.org/]. We used only variant calls that passed quality filters in the analyses. Variant calls made by either Strelka2 or Mutect2 were considered, with allowances made for the lower sequencing coverage of WGS compared to that of WES. For the other CPTAC projects, we obtained WES somatic mutation calls from the Genome Data Commons [https://gdc.cancer.gov/]; variants called by two or more algorithms were used in this study. For the prostate cancer and medulloblastoma datasets, we obtained whole-genome somatic mutation calls from the supplemental of ref. 64. Somatic small mutation calls for the APOLLO tumors were taken from the GDC publication page.

### Integrative analyses between SVs and expression
Using SVExpress[38], we defined genes with altered expression (by protein or mRNA) associated with nearby somatic SV breakpoints. No germline SVs were used in any analyses. Relative to each gene, genomic region windows considered included the within-gene regions and within 100 kb upstream or 100 kb downstream of the gene. For the above regions, SVExpress constructed a gene-to-sample matrix with entries as 1, if a breakpoint occurs in the specified region for the given gene in the given sample, and 0 if otherwise. We also used SVExpress to examine a 1 Mb region surrounding each gene, using the "relative distance metric" option[5], whereby breakpoints close to the gene will have more numeric weight in identifying SV-expression associations, while breakpoints further away but within 1 Mb can have some influence. Gene-level SV-expression association analyses included 15,439 unique named genes, 10,087 of which had protein values for at least 400 tumors. The 10,087 genes were the focus of the recurrent SV-

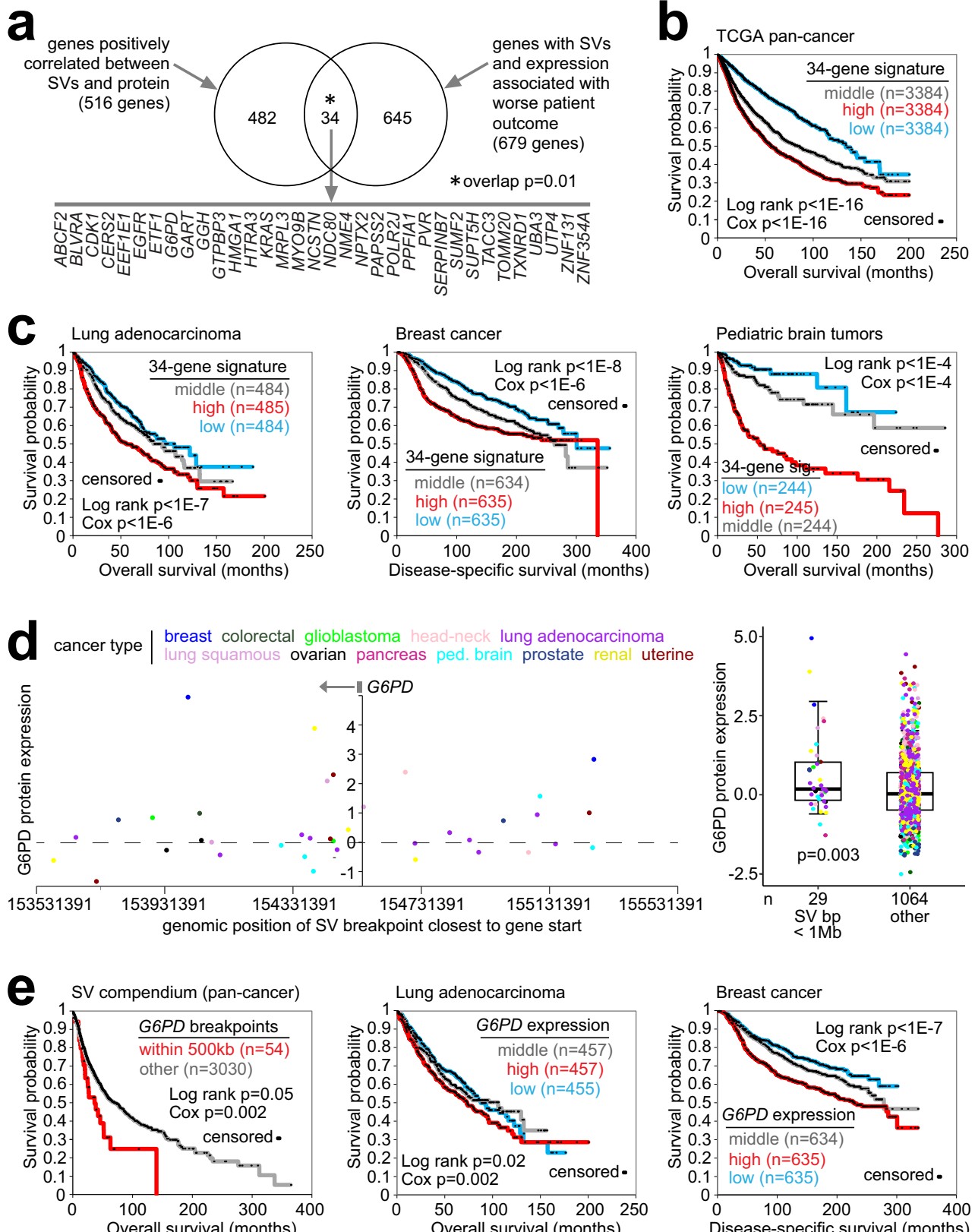

protein expression association analyses, where previously we could identify similar SV-mRNA associations using cancer datasets of 327 or 570 cancers[35,38]. The 10,087 proteins were under-represented for G-protein coupled receptors, which class was associated previously with SV-associated altered regulation[4,35]. Gene-level SV-protein association analyses were limited to the above 10,087 genes. Using the

geneXsample SV breakpoint matrix, SVExpress assessed the correlation between expression of the gene and the presence of an SV breakpoint using a linear regression model (with log-transformed expression values), incorporating sample cancer type and gene-level CNA. For the analyses involving within-gene, 100 kb upstream, and 100 kb downstream gene regions, we only considered genes with at

**Fig. 7 | Somatic SV-altered proteins involving patient survival. a** Gene-level associations with patient survival, at the levels of both mRNA and nearby SV breakpoints, were examined in a cohort of 3084 patients with combined SV-survival data from publicly available datasets (Supplementary Dataset 1). Of 516 genes with positive SV-protein expression associations ($p < 0.05$, using 1 Mb region, with corrections for tumor type and gene-level CNA), 34 genes had both nearby SV breakpoints and expression associated with worse patient outcome in 3084-patient cohort (one-sided $p < 0.05$ for each variable, stratified Cox correcting for cancer type; SV breakpoints also correcting for gene-level CNA). *P*-value for significance of overlap by one-sided Fisher's exact test. **b** Association of the 34-gene expression signature from part **a** with patient survival in TCGA pan-cancer dataset ($n = 10,224$)[31]. *P*-values by log-rank test (patients binned by tertiles) and by univariate Cox, as indicated, corrected for cancer type. **c** Association of the 34-gene expression signature from part **a** with patient survival across multiple cancer types and three separate datasets: lung adenocarcinoma ($n = 1453$)[70], breast cancer

($n = 1904$)[71], and pediatric brain tumors ($n = 893$)[19]. *P*-values by log-rank test (patients binned by tertiles) and by univariate Cox, as indicated. For the pediatric brain tumor dataset, *p*-values corrected for histologic type. **d** Protein expression levels of *G6PD* corresponding to SVs located in the genomic region 1 Mb downstream to 1 Mb upstream of the gene (left). Each point represents a single patient (closest SV breakpoint represented for each patient). Tumors with gene amplification are indicated. Boxplot (right, representing 5%, 25%, 50%, 75%, and 95%) shows *G6PD* expression by tumors with SV breakpoint within 1 Mb of gene start ($n = 29$) versus other tumors ($n = 1064$). *P*-value by *t*-test. **e** Association of *G6PD* SV breakpoint and expression patterns with worse patient outcome ($n = 54$ within 500 kb of gene, using the extended cohort of 3084 patients), and association of *G6PD* expression with worse patient outcome in lung adenocarcinoma ($n = 910$) and breast cancer ($n = 1904$) cohorts. *P*-values by log-rank test and by univariate Cox. For the SV breakpoints dataset, tests correct for cancer type.

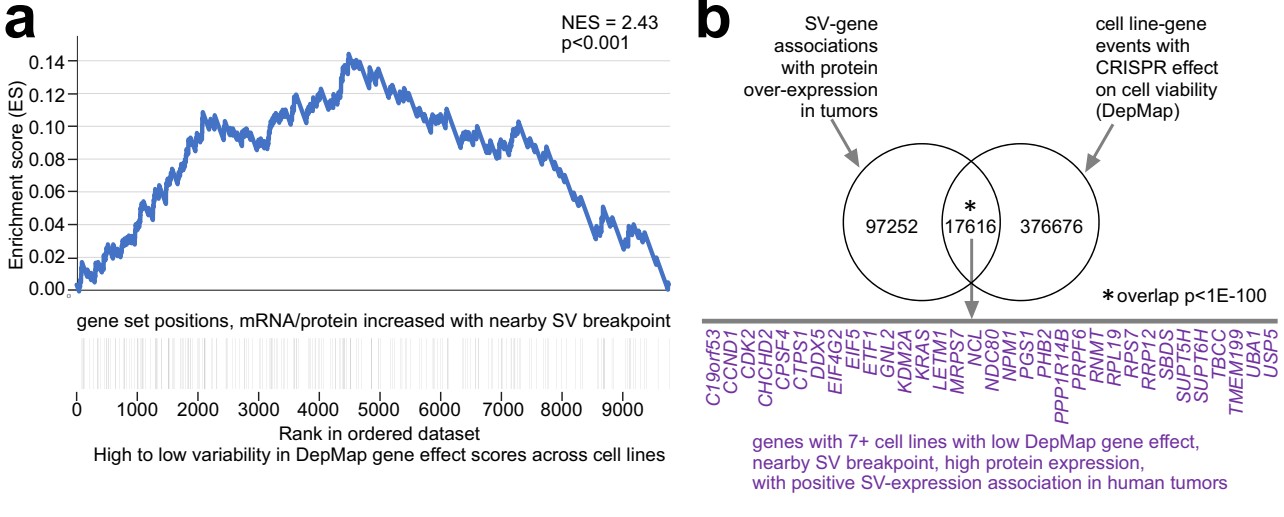

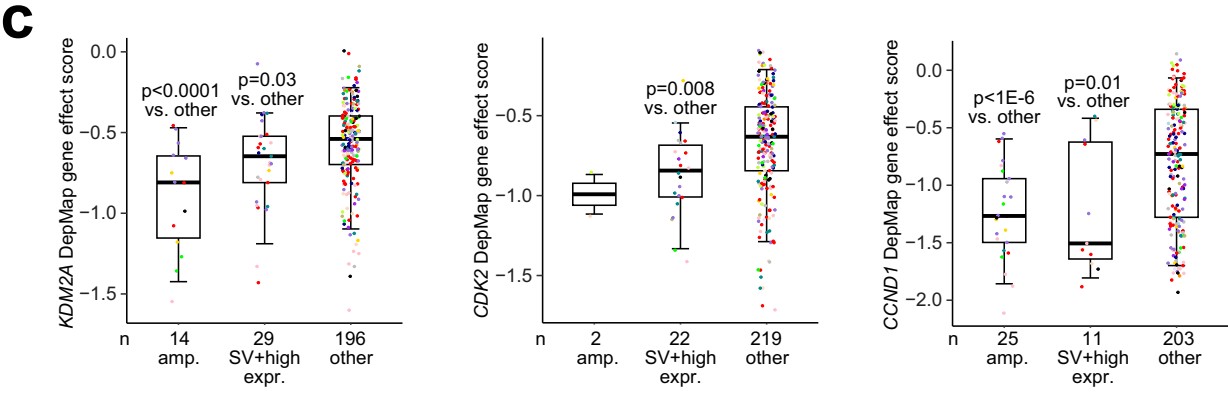

bladder, blood, breast, brain/cns, colorectal, kidney, liver-biliary, lung, ovary, pancreas, prostate, skin, stomach/esophagus, thyroid, uterus, other

**Fig. 8 | SVs associated with high protein expression and cell viability in cancer cell lines. a** By GSEA method[75], enrichment of genes having both mRNA and protein expression ($p < 0.05$) increased with nearby SV breakpoint across human tumors, within genes with high variability in DepMap gene effect scores[53] across cancer cell lines. SV-expression association *p*-values in tumors correct for cancer type and CNA by linear modeling, using 1 Mb region. For genes with high variability in gene effect scores, a fraction of cell lines, but not all, would presumably be sensitive to CRISPR knockout of those genes. **b** In cell lines, significant overlap of events involving CRISPR knockout effect in genes with events involving SV-associated over-expression. Based on the set of all SV-gene associations involving an SV breakpoint falling within 1 Mb of the genes combined with protein over-expression (taken from all gene X cell line sample pairings), the Venn diagram represents significant enrichment of these SV-gene associations with CRISPR knockout for these same

genes in the same cell lines. Genes listed represent those that had both a positive SV-expression association in human tumors ($p < 0.05$, for protein; one-sided $p < 0.05$ for mRNA, 1 Mb region) and—for seven or more cell lines—combined low DepMap gene effect ($< -0.5$, denoting sensitivity to knockout), nearby SV breakpoint, and high protein expression (SD > 0.4 from the median). **c** Box plots of DepMap gene effect scores (lower score, greater effect of knockout on cell viability) for *KDM2A*, *CDK2*, and *CCND1* by alteration class ("amp." or high-level gene amplification, approximating copy levels more than 2× greater than that of wild-type, involving 14, 2, and 25 cell lines, respectively; SV breakpoint within 1 MB combined with high protein or mRNA over-expression, involving 29, 22, and 11 cell lines, respectively; or none of the above, i.e., "unaligned," involving 196, 219, and 203 cell lines, respectively). Box plots represent 5% (lower whisker), 25% (lower box), 50% (median), 75% (upper box), and 95% (upper whisker). *P* values by *t*-test.

least three tumors associated with an SV within the given region when estimating False Discovery Rate[39].

By SVExpress, a gene shows significant SV-expression associations if the expression and SV breakpoint patterns line up non-randomly with respect to each other across all samples analyzed, after correction for covariates. By design[5], our integrative analytical approach does not assume the specific mechanism of altered expression and treats SV breakpoints representing different classes (tandem duplications, insertions, deletions, inversions, and translocations) and insert sizes the same[35]. If multiple breakpoints occur near the gene, the breakpoint closest to the gene start is used in the breakpoint matrix. In addition to analyzing all tumors from the various cancer types in our combined cohort, we carried out analyses within tumor subgroups by cancer type, focusing on the cancer types we previously examined for SV associations[5], with additional tumors represented in our present study. For some tumor subgroups, genes with significant SV-associated alterations may involve breakpoints in just one or two samples, combined with very high or very low expression changes relative to the other tumors in the subgroup.

Our analytical approach factored in the effects of CNA and cancer type, and the SV-expression associations that could be explainable by CNA alone were not represented in our main findings. Because genomic rearrangements are often involved in CNA[6,65], we used our assembled gene-level CNA table corresponding to the geneXsample proteomic and SV breakpoint tables (see above). For each gene, we assessed the correlation between expression of the gene (by protein or mRNA) and the presence of an SV breakpoint using a linear regression model, incorporating sample cancer type and gene-level CNA. By linear regression modeling, any genes selected as having significant correlations between SV breakpoints and expression must arise above any associations that would be better explained based on either CNA or cancer type alone. In other words, CNA alone cannot account for all observed cases of altered expression in conjunction with SV breakpoint events near the gene. In this study, cancer type largely reflected the batches involved in the expression compendium datasets, as each cancer type (by tissue of origin) was contained within one or two batches (as defined by separate studies), and no batches involved multiple cancer types. Alternative linear models incorporating the 15 batches as a covariate instead of the 12 cancer types (Supplementary Dataset 1) yielded the same overall results as those used in our study.

Using SVExpress, we also assessed the impact of SV breakpoints on phosphoprotein levels and on DNA methylation levels. Logit-transformed methylation beta values were used in the linear regression modeling for DNA methylation data. For the SV-mRNA integrative analyses, we generated mRNA results for both the full mRNA dataset, including expression values that may not be represented in the protein dataset (e.g., protein expression not detected for a particular gene in a particular tumor), and for a filtered mRNA dataset, with any expression data values not represented in the protein dataset removed. The results for the filtered mRNA dataset would help assess how missing protein values could contribute to disparate results between the protein and complete mRNA datasets. In the Results section, the filtered mRNA dataset results are utilized for the direct protein-mRNA concordance analyses surrounding the analyses presented in Fig. 2, as noted, with all other mRNA-related results reported relying on the unfiltered mRNA dataset.

## Gene fusion analysis

Fusion calls by RNA chimeric reads were available for 1192 of the 1307 tumors in our combined cohort with SV and protein expression data (Supplementary Dataset 1). Fusion calls were not available for all tumor samples with protein expression, as some samples did not have RNA data or had RNA data by expression arrays instead of sequencing (and hence, would not have chimeric read results available) or did not have fusion algorithms applied in the public data portals. For CPTAC, TCGA,

and APOLLO tumors, we obtained from the GDC candidate fusion calls based on RNA-seq data from STAR-Fusion and Arriba algorithms. For CBTN tumors, we previously obtained STAR-Fusion and Arriba calls through the public project on the Kids First Data Resource Portal and Cavatica [https://cbtn.org/]. No RNA chimeric fusion calls were available for ICGC prostate and medulloblastoma cohorts. For Arriba calls, we removed low-confidence fusion calls and did not consider any further candidate fusions for which 50% or more of the calls were designated as low confidence. Fusion candidates with breakpoints in intergenic regions were not considered. For each fusion call based on chimeric RNA-seq reads detected in a tumor, we determined whether SV breakpoints by WGS were found within one or both genes. For a subset of candidate fusion calls in tumors without SV breakpoints detected, we noted WGS support if at least 20% of tumors with that fusion candidate had SV breakpoints detected. Of the chimeric-based fusion calls with SV breakpoint support, we cataloged the fusion calls associated with gene over-expression by either mRNA or protein. Here, we defined over-expression in a tumor sample as >0.4 SD from the sample median within the given cancer type, for either gene. We noted RNA or protein expression support for a subset of candidate fusion calls in tumors without over-expression if at least 20% of tumors with that fusion candidate had over-expression by RNA or protein, respectively.

## Pathway-level somatic alteration categories

For the pathway-centric view of somatic alterations (Fig. 4b), key pathways and genes previously annotated across multiple cancer types based on domain knowledge[6,31,33,45] were included: Receptor Tyrosine Kinase (RTK) pathway, HIPPO pathway, chromatin modification, SWI/SNF complex, mTOR pathway, MYC family, *TERT*, Wnt/beta-catenin, and p53/Rb-related. Oncogenes and tumor suppressor genes falling within each pathway are listed in Figs. 4c and S4 and in Supplementary Dataset 1. For known oncogenes with known "hotspot" residues[41], if an SNV occurred in a "hotspot," the SNV was considered in the analysis. We also considered *TERT* activating promoter mutations[56]. At both the gene and pathway levels, we tabulated somatic alterations in the following order: SNV or indel, gene fusion, deep deletion (estimated zero gene copies), high-level amplification (estimated five or more copies), and somatic SV (for oncogenes, breakpoint falling with 1 Mb of gene and associated with expression>0.4 SD from the median across samples for the given tumor; for tumor suppressors, breakpoint falling within the gene body and expression < −0.4 SD).

## Integration of TADs, enhancers, and retrotransposons

To identify SV breakpoints associated with TAD disruption, we used SVExpress[38] and published TAD data from the IMR90 cell line[66], using the UCSC Genome Browser LiftOver tool to convert TAD coordinates from hg18 to the coordinate system used for the given SV call dataset (hg19 or hg38). We defined TAD-disrupting SVs as those for which the two breakpoints did not fall within the same TAD.

To identify potential enhancer hijacking events involving gene-level SV-expression associations, we used SVExpress with the enhancer annotations as provided by Kumar et al.[67]. For each SV breakpoint association within 1 Mb upstream of a gene (each association involving unique breakpoint and gene pairing, with only the SV breakpoint closest to the start of each gene being considered for each tumor in the instance of multiple breakpoints being detected), SVExpress determined the potential for translocation of an enhancer near the gene that would be represented by the rearrangement (based on the orientation of the SV breakpoint mate). SVExpress considered only SVs with breakpoints on the distal side from the gene in this analysis. Hg19 SV call sets were analyzed separately from hg38 SV call sets, using the corresponding genome annotation for the enhancer positions.

To identify the translocation of LINE and SINE retrotransponsons involving SVs associated with altered gene expression, we again used

the SVExpress enhancer associations module but using the annotations from RepeatMasker open v4.0.5 (hg19) and v4.0.5 (hg38)[68]. For each SV breakpoint association within 20 kb upstream of a gene, SVExpress determined the potential for translocation of a retrotransposon near the gene that would be represented by the rearrangement (based on the orientation of the SV breakpoint mate). SVExpress considered only SVs with breakpoints on the distal side from the gene in this analysis. The phenomenon of translocated retrotransposons, as studied here, is distinct from that of somatic retrotransposition[69].

### Survival analyses

We identified gene-level molecular correlates of patient survival, at the levels of both mRNA and nearby SV breakpoints, in a cohort of 3084 patients with combined SV-survival data from publicly available datasets (Supplementary Dataset 1). Of the 3084 patients, 1194 were represented in our proteomic compendium dataset. In addition, the 3084 patients included 100 lung adenocarcinomas from CPTAC confirmatory cohort and 1790 patients from the PCAWG cohort[11]. Of the 3084 patients, 2373 had mRNA data. For increased statistical power, we used a larger patient cohort for survival analyses, e.g., given shorter median patient follow-up times.

For associating nearby SV breakpoints with patient outcome, we utilized the [gene X tumor] relative distance breakpoint matrix, generated by SVExpress. For each gene, we used a stratified Cox (correcting for cancer type) to associate patient overall survival with the log2-transformed relative distance to the nearest breakpoint for that gene. We also associated mRNA expression of the gene with overall survival using stratified Cox (corrected for cancer type, using as.factor in R). Genes significant for both the relative breakpoint analysis and the expression analysis were compared with the set of genes with SV-protein associations. When overlapping different results sets, we used a more relaxed *p*-value cutoff to limit false negatives, as the degree of gene set overlap itself was significant and yielded significant results in multiple external datasets. The numbers of significant genes associated with worse survival by either expression or breakpoint pattern far exceeded the chance expected (see Results and Supplementary Dataset 9), and in taking the 679 genes that overlap between both expression and breakpoint survival results sets (Fig. 7a), multiple testing becomes even less of a concern.

We also examined genes and gene sets of interest in public cancer transcriptomic datasets for associations between expression and patient outcome. To analyze lung adenocarcinoma patient survival, we examined a compendium dataset of 11 published mRNA expression profiling datasets for human lung adenocarcinomas[40,70]. To analyze breast cancer patient survival, we used the Pereira et al. expression dataset[71] (as downloaded from CBioPortal). For analysis of pediatric brain tumor patient survival, we used RNA-seq data from CBTN[19,40]. The TCGA pan-cancer RNA-seq dataset, representing 32 major cancer types and 10,224 tumors, was assembled from the Broad Institute's Firehose data portal [https://gdac.broadinstitute.org][31]. Given a gene signature (e.g., the 34-gene signature of Fig. 7a), we scored patient profiles in the external expression dataset by taking the average of the normalized expression values (standard deviations from the median across samples) for the entire set of genes. We assessed the association of the expression of individual genes or a gene signature score with patient outcome using univariate Cox and log-rank (dividing the patients according to low, high, or intermediate signature scoring). In addition, for analyses utilizing the TCGA pan-cancer or CBTN datasets, stratified Cox models or stratified log-rank tests were used to evaluate survival association when correcting for tumor type. For analyses involving the lung adenocarcinoma compendium or TCGA pan-cancer datasets, patient survival was capped at 200 months. For the CBTN dataset, only one tumor per patient was included in the survival analyses, and patient survival was capped at 285 months.

### Cell line datasets

We assessed the mass spectrometry-based proteomics data on 949 cell lines in total from Gonçalves et al.[54] and on 375 cell lines in total from Nusinow et al.[55]. For any proteomic values not represented in the Gonçalves dataset (e.g., missing values or cell lines not represented), we used the values from Nusinow. We then z-normalized protein expression values to standard deviations across cell lines in the combined dataset. We assessed gene-level associations between protein expression and nearby somatic SV breakpoints across 328 cancer cell lines in the Cancer Cell Line Encyclopedia (CCLE) with WGS data. The CCLE datasets, including WGS and RNA profiling, were from the 2019 release[72], with somatic SV calls previously made in these cell lines by SVABA algorithm[72]. We cataloged all gene-to-nearby SV breakpoint associations (within 1 Mb), for which the cell line had over-expression of the associated protein (>0.4 SD from the median across cell lines). Gene effect scores (with low scores denoting essential genes), based on Cancer Dependency Map (DepMap) CRISPR assays, were also examined using the dataset as analyzed using the Chronos algorithm from Dempster et al.[53]

### Statistical analyses

All *p*-values were two-sided unless otherwise specified. Nominal *p*-values do not involve multiple comparison adjustments, while FDRs involve *p*-values adjusted for multiple gene feature comparisons. We relied on a stricter FDR cutoff for defining top genes when carrying out gene-level global molecular analyses for a single analyte (e.g., gene-level SV-protein associations or SV-mRNA associations). When overlapping different top-gene results sets across multiple independent analyses (e.g., gene-level SV-expression associations as observed in both protein and mRNA data), we used a more relaxed *p*-value cutoff for each individual analysis to limit false negatives better identify significant overlap patterns. As reported, the degree of gene set overlap across independent analyses was often highly statistically significant. This practice would be consistent with our previous studies identifying genes with demonstrated functional roles as originally identified using integrative analyses[73,74], and with standard analytical methods like GSEA that can identify enrichment patterns that would be missed using overly strict FDR cutoffs[75,76]. Enrichment of GO annotation terms[77] within sets of differentially expressed genes was evaluated using SigTerms software[78] and one-sided Fisher's exact tests. Visualization using heat maps was performed using both Java-Treeview (version 1.1.6r4)[79] and matrix2png (version 1.2.1)[80]. Figures indicate exact value of *n* (number of tumors or cell lines), and the statistical tests used are noted in the Figure legends and next to reported *p*-values in the Results section. Boxplots represent 5%, 25%, 50%, 75%, and 95%. Figures represent biological and not technical replicates.

### Reporting summary

Further information on research design is available in the Nature Portfolio Reporting Summary linked to this article.

## Data availability

All data used in this study are publicly available. Proteomics data are available for CPTAC and ICPC studies at the Proteomic Data Commons [https://pdc.cancer.gov/]. For CPTAC and APOLLO studies, structural variant data, transcriptome data, copy number data, and small somatic mutation data are available at the Genomic Data Commons [https://gdc.cancer.gov/]. CBTN structural variant, transcriptome data, copy number data, and small somatic mutation data are available through the public project on the Kids First Data Resource Portal and Cavatica [https://cbtn.org/] and through the PedCBioPortal [https://pedcbioportal.org/]. Raw genomic and transcriptomic CPTAC data can be accessed via dbGap Study Accession: phs001287.v13.p5. Raw genomic and transcriptomic APOLLO data can be accessed via dbGap

Study Accession: phs003011.v1.p1. For ICGC studies, structural variant data are available via the ICGC data porta [https://dcc.icgc.org/pcawg/]. Raw data for the above may be obtained once authorized access is granted via a Data Use Certification (DUC) agreement. Genomic and transcriptomic prostate cancer data are available at found on European Genome-Phenome Archive (EGA), under accession EGA: EGAS00001000900. TCGA RNA-seq data are also available from the Broad Institute's Firehose data portal [https://gdac.broadinstitute.org]. Cancer Cell Line Encyclopedia (CCLE) datasets are available from the CCLE website [http://www.broadinstitute.org/ccle]. For other published studies, molecular data availability information is provided in the associated publication. The compendium datasets of molecular profiles for gene-level total protein, mRNA, CNA, and SV breakpoint patterns (for genomic region windows 100 kb upstream, 100 kb downstream, within gene, and 1 Mb upstream or downstream)—compiled as part of our study—are available through figshare [https://doi.org/10.6084/m9.figshare.22669888]. Each molecular dataset has a common gene and sample set, allowing one to derive correlations between SV breakpoint patterns and gene expression, e.g., using the SVExpress software[38]. Any remaining data are available within the Article or Supplementary Information. Source data are provided with this paper.

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

## Acknowledgements

This work was supported by National Institutes of Health (NIH) grant P30CA125123 (C.J.C.) and NCI U54CA118948 (S.V.).

## Author contributions

Conceptualization: C.J.C.; Methodology: C.J.C., Y.Z., F.C.; Formal Analysis: C.J.C., Y.Z., F.C.; Data Curation: Y.Z., C.J.C., D.S.C., F.C.; Visualization; C.J.C.; Writing: C.J.C.; Manuscript Review: Y.Z., F.C., D.S.C., S.V.; Supervision: S.V., C.J.C.

## Competing interests

The authors declare no competing interests.
