## [Peer Review File · Nature Communications]

Global impact of somatic structural variation on the cancer proteomeReviewers' Comments:

Reviewer #1:

Remarks to the Author:

In the study the authors present the analysis of the impact of SVs on both mRNA expression and protein expression in cancer genomes. The focus is made on mRNA+protein expression alternations caused by altered cis-regulation, gene fusions, or gene disruption. Effect of copy number alterations (CNA) are not taken into account and supposed to be excluded (however the CNA correction is not clear to me, see the comments below). What authors noted is that mRNA expression level is not always correlated with protein expression level, in fact they state that it is true only for 25% of the genes. While this sounds as expected that mRNA level should not always correlate with protein level, a year ago in Nat Comm 2022 (<https://doi.org/10.1038/s41467-022-30342-3>) the same group stated the opposite: "Protein expression of genes broadly correlates with corresponding mRNA levels or copy number alterations (CNAs) across tumors". This fact is not discussed in this paper.

Though the big amount of work has been done, the presentation of the results is far from perfect. There are many discrepancies in presenting the results, especially in numbers that are not consistent throughout the text (see the comments below). Also, in many places the authors refer to results as (a fraction, a substantial fraction, hundreds of) that when one looking at figures does not find that description even qualitatively correct. Exact numbers and percentages required for understanding.

If the focus on the paper on the joint effect of SVs on mRNA and protein (let us denote it as SV-mRNA-protein), then the different sections refers to different numbers and fractions, and the entire picture is not assembled from this mosaic. I would suggest to start at the beginning of total set - how many genes have both mRNA+protein expressions available, and then which part is associated as significant for SV-mRNA-protein. Then describe the subgroups. In the first section for SV-mRNA-protein the reader got the number of 201, but then in the next section with gene fusion SV-mRNA-protein the number is 1098. Should we assume that SVs from the first sections are not SVs that contributed to gene fusions? But then the total number of SV-mRNA-proteins is 1098+210, but this number is never discussed. I would expect the authors first provide a big general classifications that would be logically divided into smaller one. And importantly all the numbers from subsection should sum up in the total so that a reader is content with arithmetic.

Also I got an impression that authors included all intermediary figures they generated without taking care of highlighting the most important and present them for the final readers outside the group. Some figure section are even described in the text and they are not self-explanatory.

The discussion is also poor and does not summarize the found effects of SVs quantitatively expect for 25% of proteins correlated to mRNA levels. The conclusions like "a fraction" or "hundreds" or "many" do not provide a knowledge. I think that once a general line of the presentation is restricted from general to subgroups (SVs, fusions, transposons, regulatory elements, etc) and the overlap between subgroups is fixed then the conclusion would follow logically and be more precise and understandable.

Major comments:

- 1 The contradictions in the statement published a year earlier in Nat Comm 2022 that "protein expression of genes broadly correlates with corresponding mRNA levels" should be explained. It looks like the same mRNA+protein expression data was used for the last year Nat Comm and the presented study. If it is not the same data then it should be also clarified.
2. The numbers are not consistent. Last paragraph of the section "Impact of SVs on protein expression" – numbers do not sum up – "657 had a significant SV association for the 1Mb region ($p < 0.01$ with cancer 147 type and CNA correction) at the mRNA level, of which just 170 (26%) had a corresponding 148 association at the protein level ($p < 0.05$)."

Next page the number increases up to 201 "Taking a set of 201 genes with significant SV-associations for both protein and mRNA (using filtered mRNA dataset)," unexplainable number 319 ("the 319 genes with 157 significant SV-association for mRNA") – one expects $657-170= 487$

Lines 52-53 Contradictions in the statement in NatComm 2022 and this submission

Line 77-78 - "We found that a substantial percentage of SV-altered mRNAs were not similarly altered at the protein level." – specify what are the exact numbers of "a substantial percentage"

Line 92-93 - SVs also involved DNA methylation alterations in a fraction of associated protein expression changes. - specify what are the exact numbers "in a fraction"

Line 122 – "By integrating proteomic and transcriptomic data with WGS-derived SV data³⁸, hundreds of genes showed altered gene expression—at either the protein or mRNA levels—in relation to nearby somatic SV breakpoints, that would not be explainable due to any associated CNA" – How do we know that they are not explainable due to CNA?

Line 127 - Significant genes after CNA correction (see Methods) – I did not find description of CNA correction in the methods. The only mention of CNA in the methods is this: "Using the geneXsample SV breakpoint matrix, SVExpress assessed the correlation between expression of the gene and the presence of an SV breakpoint using a linear regression model (with log-transformed expression values), incorporating sample cancer type and gene-level CNA." Please explain.

Line 132 – First, the author refer to some other results on SV-mRNA-protein with some numbers and only further down present their own results. I would suggest to do the reverse. First present their own and then compare them from what the others saw. Why the numbers are different?

Line 140 – "a set of 1200 recurrently altered genes" – please provide definition of recurrently altered genes

Line 150-152 "Most of these 3419 events involved over-expression of one or both genes at the mRNA level, with a fraction of these events also showing protein over-expression (Figure 3b)." One looks at the Figure 3B and sees that 1098 out of 3419 $\Rightarrow \sim 30\%$. One third is not "a fraction"

Line 165-184 Section Gene fusions with protein expression. Is that correct that the number of gene-fusions-mRNA-protein is 5 times larger than recurrent-genes-mRNA-proteins described in the previous section ($\sim 1098 : 201$)? Also, the authors describe fusions with effect for protein+mRNA, then describe fusions for mRNA-protein non-available. Why other categories are excluded (mRNA-noprotein or no-mRNA-protein) then?

Figure 3D is not intuitive and I did not find the explanation in the text. For Figure 3d and many others sorting by mRNA and then by proteins would help to see genes with both effects on mRNA+protein.

Line 262-264 - A significant fraction of the genes with SV-protein associations in our protein-WGS compendium cohort also correlated with patient survival across an extended mRNA-WGS compendium cohort of over 3000 patients (Figure 7a, Table S1)." – how many genes? Previously the authors were talking about 201 genes SV-mRNA-protein in the first section and 1098 fusion(SV?)-mRNA-protein, and now they talk about "3156 and 3476 genes were associated with worse overall survival by somatic SV breakpoint pattern and expression, respectively". The figure 7 gives the number 512 for SV-mRNA-protein – I am entirely confused.

Line 333 "At the same time, SV breakpoint patterns in relation to genes represent sparse events in

cancer genomic data.” – I did not understand what the authors wanted to say.

Methods

Line 646 – “All but 24 of the 1426 tumors.” – Again, confusion with numbers - in the previous section “1413 tumors in 636 total, of which 1294 had proteomic data and of which 988 had DNA methylation data” – where from the 1426 came from?

Line 658 “Of the 1307 tumors with proteomic data” contradicts with line 646 “of which 1294 had proteomic data”

Line 780 Fusion calls by RNA chimeric reads were available for 1192 of the 1307 tumors in our combined cohort with SV and protein expression data (Table S1).

Minor corrections

phospho-proteomic -> phosphoproteomic
altered tumor – means SV-altered tumor?

Line 186 – “We found somatic SVs and associated altered protein levels” – means both mRNA+protein?

Line 216 SSV-associated -> SV-associated

Line 223-225 – only a fraction – how much?

Line 230 SSV breakpoints -> SV breakpoints

Figure 2 - bad quality

Figure 3d – the tumor type labels are unreadable

Legend Figure 4 – “Genomic rearrangements (represented in circos plots) involving altered expression” – altered expression both mRNA+protein?

In all figures - color-coded cancer types are almost impossible to read

Reviewer #2:

Remarks to the Author:

Chen et al present a study describing the impact of somatic structural variation at both transcription and proteome level. Uniquely this paper includes not only WGS and RNA-seq data but also mass spectrometry data. This multi-omics approach to interpreting the effects of somatic structural variation has the potential to increasing our understanding of SVs affecting both coding and non-coding regions of the genome, however the lack of correlation between values makes interpretation challenging. After reading the manuscript I have the following comments.

Major remarks

1. The paper would be strengthened if authors could highlight what the added value is of the proteomics and what new knowledge is generated in comparison to the WGS & RNA results.
2. One of the main findings is that only ~25% of genes with mRNA expression changes associated with SVs also have protein expression changes. This observation raises many questions. Is this expected given relatively poor overall correlation between protein and mRNA expression (page 5, lines

113-116)? Were there differences observed across the spectrum of SV types, since they can vary a lot in size and complexity? And did they see a better correlation between mRNA and protein expression for known driver alterations?

3. Following on from point 2, How do SVs compare to other somatic mutation types in this regard? Do the authors expect a difference in predictability of effect on expression between SVs and other mutations 'in cis'.

4. In general, the authors found more associations between SVs and overexpression or upregulation of genes compared to downregulation of genes. However, SV breakpoints do not necessarily need to be accompanied by lower expression to be disruptive to a (tumor-suppressor) gene, especially if they fall within the gene body.

Therefore what is now implemented in the methods might be overly strict towards tumor suppressor genes.

"for oncogenes, breakpoint falling with 1Mb of gene and associated with expression $>0.4SD$ from the median across samples for the given tumor; for tumor suppressors, breakpoint falling within the gene body and expression $<-0.4SD$ " (page 34, lines 809-812)

Can you repeat the analysis requiring either a breakpoint inside the gene or a breakpoint within 1Mb together with reduced expression?

Minor remarks

1. At present the study seems to contradict itself, as it states "Proteogenomics identifies targetable non-coding alterations." (page 2, line 50), Yet the paper focusses on gene-level analysis, and only considers SVs affecting genes.

2. In the introduction there are several mentions of "percentage" or "fraction" without giving the numbers (page 4, lines 90-97). Mentions of "only a fraction" implies that the fraction is small, but the numbers elsewhere indicate sizeable fractions such as $\sim 30\%$. Please consider rephrasing and use more specific values.

3. See above. Also at other locations in the text the phrase "only a fraction" is used without percentages and refers to a non-small fraction.

"Only a fraction of the genes significant for a given region at the mRNA level were significant at the protein level .." (page 6, line 144)

"As anticipated, only a fraction of the chimeric-based fusions predictions ..." (page 7, line 169)

4. It is not always clear what cohort is used for what analysis. Different names for cohorts currently used throughout the text are: protein-WGS cohort, CPTAC cohort, protein WGS compendium cohort, extended mRNA-WGS compendium cohort, 1426 tumor cohort. Can the authors add a supplementary figure describing the overlap between the different cohorts?

5. Reference is missing for: "Consistent with previous observations, many more genes showed positive correlations with SV breakpoints (i.e., expression was higher when a nearby SV breakpoint was present) than negative correlations, the former including known oncogenes and the latter including tumor suppressor genes." (page 6, lines 129-132)

6. The authors show enrichment of certain terms in the gene set with both protein and mRNA SV-associations, and also in the gene set with only mRNA and no protein SV-associations (page 7, lines 153-159). How do these compare to the background correlation between protein/mRNA expression levels overall? Could it be the case that the observed enrichments are due to genes that inherently

show more or less correlation between their protein/mRNA expression levels?

7. Its currently unclear what is the main message and novelty is of the gene fusion section? Please rephrase, and note gene fusions don't need over- or underexpression to contribute to tumorigenesis.

8. SSV instead of SV is used on on page 9, line 215 and page 10, line 230.

9. More details are required of the definition of overexpression or upregulation used, as it is currently unclear. At the start of the manuscript "overexpression" seems to be defined as significant by SVExpress and in the section "Mechanisms of SV-mediated altered protein expression" it seems to be defined in another way: "protein over-expression ($>0.4SD$ from sample median)" (page 10, line 245).

10. The abstract requires some rewriting as it does not seem to represent the contents of the manuscript as it focuses very heavily on non-coding variants and does not mention seemingly important analyses such as the gene sets identified in section.

Reviewer #3:

Remarks to the Author:

This paper integrates proteogenomics data from multiple cohorts (e.g. CPTAC, APOLLO, TCGA, ICGC, CBTN) to study the impact of somatic structural variation (SV) on cancer proteome. It evaluates SV from various aspects, such as their impact on protein levels, pathways, DNA methylations, and survival. It also has a dedicated evaluation for gene fusion, a critical type of SV, and explores the mechanisms of SV-mediated alteration of protein levels. Finally, using data from cell line experiments, it highlights 33 genes whose knockout is sensitive to a fraction of cell lines.

1. This study involves many combined analysis datasets (e.g. WGS - proteome, WGS - translation, WGS-methylation, survival data) with different sample sizes. It's very confusing to figure out what analysis is done in which data set on how many genes using what significant calling. An overview figure for sample availability, data types and analyses is helpful.

2. Proteomic studies uses LC-MS/MS profiling technology, which quantifies relative abundance of a protein in one sample relative to the same protein in other samples in the same experiment/study. It's questionable that your integrated data (with median/standard deviation based normalization) has no batch effect. There is no evaluation of the batch effect in your results. One potential way to evaluate batch effect is to compare data of the same cancer type from different studies. For example, CPTAC and APOLLO both have lung adenocarcinoma samples. Can you plot their principle components (PCs) or UMAP in the combined data to demonstrate these two studies do not form clusters (batch effects)?

3. Similarly, can you evaluate if transcriptomics and other data types have no batch effects after your normalization? For example, both CPTAC and CBTN have pediatric brain tumor. Do you see samples from these two studies cluster together (batch effect) in PC/UMAP plots or simple clustering analysis?

4. A number of SV-RNA associations are not observed at SV-protein level, probably because SV-RNA analysis has bigger sample size and thus greater power to identify signals. How do you differentiate the biological difference from power difference in your RNA vs protein comparison?

5. It seems you did not report any SV-protein unique association that's not reflected at the RNA level. Then, what unique contribution does this proteomic-focused study make? Can you demonstrate what additional biological discoveries do we have by adding proteomics data into existing multi-omics data?

6. SV-RNA association analysis controlled for the effect of CNA, while somatic alternations also include mutation. Why mutation is not controlled in the analysis?

7. Line 146-148: Why raw p-value is used for claiming significance, without considering of the multiple testing? Why different p-value cutoffs (0.01 vs 0.05) are used for mRNA and protein levels?

8. Gene fusion is of high translational relevance, as they can be important candidates for drug target. Your study identifies 1055 and 386 gene fusions with mRNA/protein and mRNA associations. How does one make use of this long list? For example, who are casual, who are important for survival or other clinical outcomes, who should be potential drug targets for future studies?

9. What does "SV-mediated" alterations means? Who is the predictor, who is the outcome, and who is the mediator? What analysis guarantees SV is a mediator, instead of a predictor or effect modifier, in the association?

10. Line 261-272. Why no multiple testing is considered for this large screening of survival associated genes? At 10%FDR, are you able to see anything associated with survival outcome?

11. Figure 7B-C: Where is 34-gene signature coming from?

12. This study has a great width on SV analysis, but not much in depth. It identifies a number of long list of significant genes from different analyses, but it's hard to understand who are important and what can we do with these genes?

Minor

13. Line 298. Please right out exact p-value in scientific format, instead of using $p \sim 0$.

Our responses and comments are marked in red italics.

Reviewer #1, expertise in structural variants, cancer genomics and bioinformatics:

In the study the authors present the analysis of the impact of SVs on both mRNA expression and protein expression in cancer genomes. The focus is made on mRNA+protein expression alternations caused by altered cis-regulation, gene fusions, or gene disruption. Effect of copy number alterations (CNA) are not taken into account and supposed to be excluded (however the CNA correction is not clear to me, see the comments below).

We thank the reviewer for evaluating our manuscript. Our study represents a first look at just how many mRNA-level SV associations show up at the protein level. We have published our analytical approach to identify SV-expression associations at the mRNA level in several previous studies¹⁻⁶. We have examined SV-mRNA associations starting with the TCGA⁶ and PCAWG⁴ cohorts of mostly primary tumors, as well as examining a cohort of advanced and metastatic tumors¹ and another cohort of pediatric brain tumors², and we examined SV-methylation associations in the context of SV-mRNA associations⁵. It is just very recently, within the last year, that data on a sizeable number of tumors with combined protein, mRNA, and WGS (for SV detection) have become available in the public domain.

As we had noted in our manuscript, we corrected for gene-level CNA in the SV-expression associations using a linear regression model (with log-transformed expression values), incorporating sample cancer type and gene-level CNA. Below and in the revised manuscript, we provide more details as to how specifically we took the effects of CNA into account in all our results, and how SV-expression associations that could be explainable by CNA are not represented in our reported findings.

What authors noted is that mRNA expression level is not always correlated with protein expression level, in fact they state that it is true only for 25% of the genes. While this sounds as expected that mRNA level should not always correlate with protein level, a year ago in Nat Comm 2022 (<https://doi.org/10.1038/s41467-022-30342-3>) the same group stated the opposite: “Protein expression of genes broadly correlates with corresponding mRNA levels or copy number alterations (CNAs) across tumors”. This fact is not discussed in this paper.

Below (e.g., in our response to Major comment #1), we elaborate on the discrepancies between mRNA-based and protein-based results. In brief, there would be no contradiction between this present study and our recent Nature Communications study⁷ that examined protein expression profiles of 2002 human tumors. These comparison results between mRNA and protein can be viewed optimistically or pessimistically, as not everything will agree, though a significant number of genes or sample-specific events will agree.

Though the big amount of work has been done, the presentation of the results is far from perfect. There are many discrepancies in presenting the results, especially in numbers that are not consistent throughout the text (see the comments below). Also, in many places the authors refer to results as (a fraction, a substantial fraction, hundreds of) that when one looking at figures does not find that description even qualitatively correct. Exact numbers and percentages required for understanding.

In the manuscript revision, we have provided more details on exact numbers where necessary. Also, one can get at the exact numbers and associated molecular features using the Excel tables that have been provided as supplemental. Using the Excel data filters, one should be able to use the selection criteria laid out in the manuscript to get at the gene lists reported. All numbers we report would have a basis in the data and can be pulled up using the Supplemental data. See below regarding the perceived discrepancies or inconsistent numbers.

If the focus on the paper on the joint effect of SVs on mRNA and protein (let us denote it as SV-mRNA-protein), then the different sections refers to different numbers and fractions, and the entire picture is not assembled from this mosaic. I would suggest to start at the beginning of total set - how many genes have both mRNA+protein expressions available, and then which part is associated as significant for SV-mRNA-protein. Then describe the subgroups. In the first section for SV-mRNA-protein the reader got the number of 201, but then in the next section with gene fusion SV-mRNA-protein the number is 1098. Should we assume that SVs from the first sections are not SVs that contributed to gene fusions? But then the total number of SV-mRNA-proteins is 1098+210, but this number is never discussed. I would expect the authors first provide a big general classifications that would be logically divided into smaller one. And importantly all the numbers from subsection should sum up in the total so that a reader is content with arithmetic.

There would be some major points of misunderstanding with the above and with related Reviewer comments below. The above 201 number refers to genes with SV-expression associations for both protein and mRNA, using $p < 0.05$ for both mRNA and protein, with $p < 0.01$ for either mRNA or protein and the filtered mRNA dataset (see response to point #2). Gene-level SV-expression associations involve genes that showed significantly altered gene expression—at either the protein or mRNA levels—in relation to nearby somatic SV breakpoints (relative to tumors without breakpoints). The 1098 number refers to fusion events, not genes. As explicitly noted in the manuscript: “The set of 1098 fusion calls with combined WGS and both protein and mRNA expression support involved 1055 distinct gene fusions, 350 tumors, and 346 patients.” There would be no way for us to reconcile the 1098 number with the 201 number, because they respectively refer to different things entirely.

Broadly speaking, it would not be possible for us to start with a single analysis and a single gene set and have all subsequent results based off this initial analysis. Figures 2,5,7,8 build from the SV-expression association analyses first presented in Figure 2, but with different statistical cutoffs as explicitly noted. Figure 3 (fusions) focuses on fusion events as originally by detection of chimeric RNA-seq reads. Figure 4 (pathways) focuses on sample-level annotation focusing on specific genes (not all of which involve SVs). Figure 6 (enhancers, etc.) focuses on genes and samples, selecting samples with higher or lower expression for the genes with SV-expression associations. The 201 genes are not explicitly used elsewhere in the manuscript other than for analyses surrounding Figure 2b. When overlapping different top-gene results sets across multiple independent analyses (e.g., gene-level SV-expression associations as observed in both protein and mRNA data), we used a more relaxed p -value cutoff for each individual analysis to limit false negatives. Statistical selection criteria are a balance between false positives and false negatives, and in places it behooves us to use a more relaxed criteria, such as when directly comparing SV-mRNA associations with SV-protein associations, where being overly strict in the SV-protein gene set cutoff would make the mRNA-protein concordance seem worse than it actually is. For all gene sets reported in our manuscript, the explicit selection used to get that gene set should have been provided, e.g., in a parenthetical statement specifying the p -value cutoff.

Also I got an impression that authors included all intermediary figures they generated without taking care of highlighting the most important and present them for the final readers outside the group. Some figure section are even described in the text and they are not self-explanatory.

In the manuscript revision, we have expanded on the Figure legends to make the interpretation of the Figures clearer. The main figures representing SV-expression associations would largely follow the templates we have used in our previous studies on SV-mRNA associations¹⁻⁶. Here we strike a balance in showing the underlying data supporting a claim, while more ancillary results can be represented in the supplementary figures. Our studies seek to provide a catalog of SV-mRNA and SV-protein expression associations, where many genes we would recognize as of known relevance to cancer but where other

genes may be less known to some but better studied by others. Therefore, we do not wish to hide genes of potential interest to other readers in presenting results. Also, readers, in general, need to see for themselves the underlying patterns behind any stated findings. For example, when we put forth specific genes arising from our SV-expression analyses as being of particular interest, we can show visually (such as by scatterplots or boxplots) how SV breakpoints near the gene more often involve over-expression. If we fail to show the underlying data underscoring a result, some readers might feel we are hiding something.

The discussion is also poor and does not summarize the found effects of SVs quantitatively expect for 25% of proteins correlated to mRNA levels. The conclusions like “a fraction” or “hundreds” or “many” do not provide a knowledge. I think that once a general line of the presentation is restricted from general to subgroups (SVs, fusions, transposons, regulatory elements, etc) and the overlap between subgroups is fixed then the conclusion would follow logically and be more precise and understandable.

We have expanded upon the Discussion section in light of the comments from the Reviewers. The overall structure of the expanded Discussion section is as follows: (1) the importance of incorporating proteomic data to refine the catalog of SV-mediated cis-regulatory alterations observed in cancer; (2) the overall correlation or lack of correlation between protein and mRNA across tumors, and how this has a bearing on our study; (3) our observations of phenomena from previous studies integrating SVs with mRNA data, with similar observations being made at the protein level in the present study; (4) the potential for combined SV and protein data to inform on personalized or precision medicine approaches. In terms of specific numbers, we prefer to keep these more general for both the Introduction and Discussion sections, as the numbers themselves are subject to the specific cutoffs used. And future studies using other patient cohorts or expanded patient cohorts will find slightly different sets of significantly associated genes. But the overall phenomenon that we have previously reported using mRNA data and now report here using protein data ought to be reflected in future studies in terms of “hundreds” or “many” or “a fraction,” even though the specific numbers will differ.

Major comments:

1 The contradictions in the statement published a year earlier in Nat Comm 2022 that “protein expression of genes broadly correlates with corresponding mRNA levels” should be explained. It looks like the same mRNA+protein expression data was used for the last year Nat Comm and the presented study. If it is not the same data then it should be also clarified... Lines 52-53 Contradictions in the statement in NatComm 2022 and this submission

The manuscript states that of the 1307 tumors in the combined SV and proteomics compendium dataset, 1093 were represented in the previous 2002-tumor study.

There would be no contradiction between this present study and our recent Nature Communications study⁷ that examined protein expression profiles of 2002 human tumors. When we previously stated that protein expression broadly correlates with the corresponding mRNA, we would put some emphasis on the word broadly. As shown in main Figure 1d of our manuscript (also shown below), even a Pearson’s r-value of 0.2 would be considered statistically significant. Still, from a practical standpoint this would not be a strong correlation (e.g., something along the lines of an r-value of 0.8 or better).

In this present study, we focus on expression outliers associated with SV breakpoints. In the absence of strong protein-mRNA correlations, an expression outlier in one analyte may not show up similarly in the other analyte. We have long known that protein and mRNA expression patterns often show weak or discordant correlations patterns⁸, as confirmed by recent studies by CPTAC and others^{9,10}. It is just very recently, within the last year, that data on a sizeable number of tumors with combined protein, mRNA, and WGS (for SV detection) have become available in the public domain. With this present study, we have a unique opportunity to observe how SV-mediated expression alterations are reflected at both the mRNA and protein levels. Proteomics would be a key factor in helping determine which somatic SVs associated with genes are important. Proteomic data, combined with transcriptomic data, are shown in our present study to greatly help refine the catalog of SV-mediated cis-regulatory alterations in cancer. The revised manuscript, including the Discussion section, further addresses this general question of overall concordance versus discordance between proteome and transcriptome.

2. The numbers are not consistent. Last paragraph of the section “Impact of SVs on protein expression” – numbers do not sum up – “657 had a significant SV association for the 1Mb region ($p < 0.01$ with cancer 147 type and CNA correction) at the mRNA level, of which just 170 (26%) had a corresponding 148 association at the protein level ($p < 0.05$).” ...Next page the number increases up to 201 “Taking a set of 201 genes with significant SV-associations for both protein and mRNA (using filtered mRNA dataset),” unexplainable number 319 (“the 319 genes with 157 significant SV-association for mRNA”) – one expects $657 - 170 = 487$

Some of the perceived discrepancies involve our use of both filtered and unfiltered mRNA datasets to generate the SV-expression associations. When comparing gene-level SV associations common between mRNA and protein, we generated two sets of gene-level SV-mRNA associations, using both the complete mRNA compendium with all available expression data and a “filtered” mRNA dataset with any values not represented in the protein dataset filtered out. With the filtered mRNA dataset, any disparate results observed between protein and mRNA should have less to do with the diminished power for some genes due to lack of detection by proteomics. This puts protein and mRNA on a more level playing field for the purposes of direct comparisons of the respective results.

The numbers in our manuscript are consistent, as reported. The 657 SV-mRNA-associated genes are based on the unfiltered mRNA dataset, using $p < 0.01$. The 201 genes are based on the filtered mRNA dataset are obtained using $p < 0.05$ for both mRNA and protein, with $p < 0.01$ for either mRNA or protein, because here we want to be more inclusive when doing the GO term enrichment analyses, as false negatives would not help us here. Similarly, the 319 genes are based on the filtered mRNA dataset, using mRNA $p < 0.01$ and protein $p > 0.05$. If we had used the unfiltered mRNA dataset, we would get a number close to the Reviewer's 487 number (except we also require protein and mRNA to correlate in the same direction for the 170 number, which drops out maybe three genes). However, with the direct comparisons between protein and mRNA, we want to account for the fact that the proteomic data has missing values due to platform limitations, hence our use of the unfiltered mRNA dataset in this section. In the manuscript revision, we have rewritten the section to make clearer where we got the numbers and what dataset is being used where.

The reported numbers of gene sets above would also be obtainable using the Excel data tables provided as supplemental. Using the Excel data filters, one should be able to use the selection criteria laid out in the manuscript to get at the gene lists reported. In addition, the revised Table S2 has columns for which of the genes in the compendium datasets fall under the above 170 genes, the 201 genes, and the 319 genes.

Line 77-78 - “We found that a substantial percentage of SV-altered mRNAs were not similarly altered at the protein level.” – specify what are the exact numbers of “a substantial percentage”... Line 92-93 - SVs also involved DNA methylation alterations in a fraction of associated protein expression changes. - specify what are the exact numbers “in a fraction”

As these statements are in the Introduction section, we save the specifics for the Results section. The above statements are broad by intention, as, for example, there are the recurrent gene-level SV-expression associations of Figure 2 versus the gene fusion analysis of Figure 3, which represent two different sets of analysis results. Also, the specific numbers depend on the statistical cutoff considered, where it would be better to expand on this in the Results section.

Line 122 – “By integrating proteomic and transcriptomic data with WGS-derived SV data³⁸, hundreds genes showed altered gene expression—at either the protein or mRNA levels—in relation to nearby somatic SV breakpoints, that would not be explainable due to any associated CAN” – How do we know that they are not explainable due to CNA?...Line 127 - Significant genes after CNA correction (see Methods) – I did not find description of CNA correction in the methods. The only mention of CNA in the methods is this: “Using the geneXsample SV breakpoint matrix, SVExpress assessed the correlation between expression of the gene and the presence of an SV breakpoint using a linear regression model (with log-transformed expression values), incorporating sample cancer type and gene-level CNA.” Please explain.

Our analysis approach, established in our previous studies examining SV-mRNA associations¹⁻⁶, took the effects of CNA into account in all our results, and the SV+expression associations that could be explainable by CNA are not represented in our reported findings and main figures. As noted above and in our previous manuscript version, we factored CNA into our analyses using linear regression models. Because genomic rearrangements are often involved in CNA^{6,11}, as described in Methods, we assembled a gene-level CNA table corresponding to the geneXsample proteomic and SV breakpoint tables. For each gene, we assessed the correlation between the expression of the gene (by protein or mRNA) and the presence of an SV breakpoint using a linear regression model, incorporating sample cancer type and gene-level CNA. Using the linear modeling approach, any genes selected as having significant correlations between SV breakpoints and expression must arise above any associations that would be better explained based on either CNA or cancer type alone. In other words, CNA alone cannot account for all observed cases of altered expression in conjunction with SV breakpoint events near the gene. The

figure represented below is taken from our previous study published as part of the PCAWG consortium⁴ (Figure 3b of that paper), illustrating specific instances of genes with significant SV-mRNA associations in the PCAWG cohort (n=1220 tumors) not explainable by CNA alone. Genes such as *TERT*, *MDM2*, *ERBB2*, and *CDK4* would each have a sizeable number of cases with amplification of the corresponding genes but also would have an additional number of cases that do not have amplification but do have an SV breakpoint and are associated with elevated expression. Such genes are not identified by eye but empirically using linear modeling.

For individual genes that we highlight in the paper using scatterplots (*AKRIC1*, *IGF2*, *NID2*, *ANO1*, *G6PD*), we illustrate which tumors have corresponding amplification event along with SV breakpoint. For most of the above genes, just a few tumors at most have amplification. The one exception is *ANO1*, which has a lot of amplification but also SV-associated cases without amplification but with elevated expression. In addition, *ANO1* shows a trend towards higher expression the closer the breakpoint is to the gene, which pattern is picked up for the 1Mb region analysis that weights the gene-level SV breakpoint patterns by their relative distance to the gene (for more information, see ref⁵). For the pathway-level annotations represented in Figure 4, we tabulated somatic alterations in the following order: SNV or indel, gene fusion, deep deletion (estimated zero gene copies), high-level amplification (estimated five or more copies), and somatic SV last; therefore, SVs associated with amplification are counted as amplification rather than as SVs, this being a more conservative approach taking into account that SVs are often associated with CNA.

In our previous studies examining SV-mRNA associations¹⁻⁶, we included a figure, like the one below, to illustrate how our linear modeling approach explicitly corrects for CNA. In the manuscript revision, we include such a figure for the 1426 tumor compendium. Supplementary Figures S2a and S2b (shown below) graph the numbers of significant genes (FDR < 0.1) showing a correlation between mRNA expression and associated SV events. Numbers above and below the zero point of the y-axis denote positively and negatively correlated genes, respectively. Where the SV-expression associations presented in the main paper all involve a correction for CNA (as well as cancer type), below we also show results from linear regression models that have no correction (blue) or that correct for cancer type but not for CNA (red). The model represented by the green bars below, which corrects for both cancer type and gene-level CNA, is what we used for all our downstream analyses. For the SV-mRNA integrative analyses, we generated mRNA results for both the full mRNA dataset, including expression values that may not be represented in the protein dataset (e.g., protein expression not detected for a particular gene in a particular tumor), and for a filtered mRNA dataset, with any expression data values not represented in the protein dataset removed. (The results for the filtered mRNA dataset would help assess how missing

protein values could contribute to disparate results between the protein and complete mRNA datasets. See our response to comment #2 above.)

From the above, we see a big drop in the numbers of significant genes when moving from a model that does not include CNA correction (red) to a model that does (green). For most of the genes in red, the SV breakpoints involve CNA, and the CNA patterns alone can explain the observed differential expression. For the model with CNA correction, some SV breakpoints may still involve CNA, but the SV breakpoint patterns involving altered expression would include additional information not captured by CNA. Factoring in cancer type does not change the results much, with slightly fewer significant genes though this may be partly due to a more complex model. The above results are entirely consistent with those of our previous studies, where we observed the same patterns in other cohorts.

In the revised manuscript, we include additional explanation in Methods as to how we correct for CNA in identifying SV-expression associations, and we refer to the new Supplementary Figures S2a and S2b in the Results section.

Line 132 – First, the author refer to some other results on SV-mRNA-protein with some numbers and only further down present their own results. I would suggest to do the reverse. First present their own and then compare them from what the others saw. Why the numbers are different?

The previous statement in line 132 does refer to our present cohort (1426 tumors, 1413 with mRNA, 1307 with protein). We have added further clarification on this point in the revision.

We did present our mRNA results from the WGS-expression compendium and then compared these results with those from our previous study of mRNA patterns in the PCAWG-TCGA cohort. In Figure S2c (also shown below), the Venn diagrams represent the overlaps between the genes with mRNAs correlated ($p < 0.05$) with SV breakpoint in our combined protein and mRNA compendium cohort ($n = 1413$ tumors) and the genes correlated ($p < 0.05$) with SV breakpoint in TCGA-ICGC cohort⁵ ($n = 2334$ tumors, 131 of which are represented in the 1413-tumor cohort). The left diagram is for positively correlated mRNAs; the right diagram is for negatively correlated mRNAs. P values are by one-sided Fisher's exact test. As we can see, the respective overlaps are highly statistically significant.

Genes significant for one cohort but not the other may be attributable to several factors involving differences in tumor type representation, false positives, and false negatives. Gene-associated SV breakpoint events may be sparse for any gene; in other words, for a given gene, on the order of just 1-3% of tumors in a cohort may have SV breakpoints within 1Mb of that gene after amplification events are excluded (see Figure 2c). Given this sparseness in amplification-independent SV breakpoint patterns, it would not be surprising for a gene to have significant SV-expression associations in some but not all cohorts examined. In other recent studies comparing SV-expression results among different patient cohorts, we found similar patterns of high overlapping gene-level associations^{1,3}. In the manuscript revision, we have added the above thoughts to the legend of Supplementary Figure S2c.

Line 140 – “a set of 1200 recurrently altered genes” – please provide definition of recurrently altered genes

In the manuscript revision, we add the following qualifier to the above-referenced statement: “For either mRNA or protein levels, a set of 1200 recurrently altered genes by SV breakpoints combined with expression differences...” The previous version did not explicitly state SVs as associated with these 1200 genes in that particular statement, though the Figure legend and the preceding Results paragraph would link the 1200 genes to SVs (e.g., “...hundreds of genes showed altered gene expression—at either the protein or mRNA levels—in relation to nearby somatic SV breakpoints...”).

Line 150-152 “Most of these 3419 events involved over-expression of one or both genes at the mRNA level, with a fraction of these events also showing protein over-expression (Figure 3b).” One looks at the Figure 3B and sees that 1098 out of 3419 => ~30%. One third is not “a fraction”

The sentence after the above statement in question gives the 1098 number: “Protein data were available for 2844 candidate fusion events out of the 3419 with WGS support, and 1098 of these involved both protein and mRNA over-expression, while 886 had mRNA but not protein expression.” We would consider ~30% to represent a fraction. Anything less than 50% in this context would be worthwhile to note as a fraction, but we also provide the underlying numbers for the reader’s benefit.

Line 165-184 Section Gene fusions with protein expression. Is that correct that the number of gene-fusions-mRNA-protein is 5 times larger than recurrent-genes-mRNA-proteins described in the previous section (~ 1098 : 201)? Also, the authors describe fusions with effect for protein+mRNA, then describe fusions for mRNA-protein non-available. Why other categories are excluded (mRNA-noprotein or no-mRNA-protein) then?

The above comment would represent a point of misunderstanding. The above 201 number refers to genes with SV-expression associations for both protein and mRNA, using $p < 0.05$ for both mRNA and protein,

with $p < 0.01$ for either mRNA or protein, using the filtered mRNA dataset (see our response to point #2). The 1098 number refers to fusion events, not genes. As explicitly noted in the manuscript: “The set of 1098 fusion calls with combined WGS and both protein and mRNA expression support involved 1055 distinct gene fusions, 350 tumors, and 346 patients.” There would be no way for us to reconcile the 1098 number with the 201 number, because they respectively refer to different things entirely. The 201 genes are not explicitly used elsewhere in the manuscript other than for the analyses surrounding Figure 2b. The other categories (involving fewer events) noted above would not be excluded, as they are highlighted in the associated figure and can be pulled up in the associated Excel supplementary data file.

Figure 3D is not intuitive and I did not find the explanation in the text. For Figure 3d and many others sorting by mRNA and then by proteins would help to see genes with both effects on mRNA+protein.

The data matrix of Figure 3d represents gene fusion events with chimeric RNA-seq reads, SV support, and corresponding high protein or high mRNA levels, involving either greater than two tumors or two tumors plus a COSMIC gene, each colored entry representing a fusion event meeting the criteria from Figure 3b. The average differential expression (at both mRNA and protein levels, yellow denoting higher expression) for fusion-involved genes in the affected tumors is indicated to the right of the data matrix. The average differential expression bars show that most of the fusion events are reflected at both protein and RNA levels, though we do not have protein data (as indicated in grey) for some genes and samples. And in a couple of instances, differential expression may be high in one sample with the fusion but not in the other. In the revision, we have expanded on the Figure 3d legend describing the figure. Also, Table S4 includes all the underlying data behind Figure 3d; one can use the Excel data filters and column AA to filter for the top fusion events represented in the figure and examine the corresponding protein and RNA expression data in columns L-O. We have used the basic format of Figure 3d in our previous studies to represent top fusion events, including Figure 4c of our study of SVs in pediatric brain tumors (2021²) and Figure 3c of our study of TCGA data (2018⁶).

Line 262-264 - A significant fraction of the genes with SV-protein associations in our protein-WGS compendium cohort also correlated with patient survival across an extended mRNA-WGS compendium cohort of over 3000 patients (Figure 7a, Table S1).” – how many genes? Previously the authors were talking about 201 genes SV-mRNA-protein in the first section and 1098 fusion(SV?)-mRNA-protein, and now they talk about “3156 and 3476 genes were associated with worse overall survival by somatic SV breakpoint pattern and expression, respectively”. The figure 7 gives the number 512 for SV-mRNA-protein – I am entirely confused.

The above 201 number refers to genes with SV-expression associations for both protein and mRNA, using $p < 0.05$ for both mRNA and protein, with $p < 0.01$ for either mRNA or protein, using the filtered mRNA dataset (see our response to point #2). The 1098 number refers to fusion events, not genes. As explicitly noted in the manuscript: “The set of 1098 fusion calls with combined WGS and both protein and mRNA expression support involved 1055 distinct gene fusions, 350 tumors, and 346 patients.” The 201 genes are not explicitly used elsewhere in the manuscript other than for the analyses surrounding Figure 2b.

The selection criteria for the 516 genes in Figure 7a is noted in the figure legend and the text: “Taking a set of 516 genes with positive SV-protein expression correlations ($p < 0.05$, correcting for cancer type and CNA), 34 of these intersected with the 679 poor prognosis genes, a significant overlap ($p = 0.01$, one-sided Fisher’s exact test).” Pertinent to the Figure 7-related analyses and other analyses, when overlapping different results sets, we used more relaxed p-value cutoffs to limit false negatives and to better identify enrichment patterns, as the degree of gene set overlap itself was significant and yielded significant results in multiple external datasets. We used a very similar approach in our recent study of SV-associated survival patterns across 570 advanced and metastatic cancers¹, where we also used relaxed statistical cutoffs to define a set of 80 genes with SV-expression association, SV survival association, and expression

association for the 570-patient cohort, which signature associated with patient survival across multiple independent cohorts representing diverse cancer types. Importantly, the 34-gene signature of Figure 7a was evaluated in datasets not used to generate the 34 genes, where, as a group, the 34 genes associate with worse prognosis in these other datasets.

Line 333 “At the same time, SV breakpoint patterns in relation to genes represent sparse events in cancer genomic data.” – I did not understand what the authors wanted to say.

We have expanded on the statement in question to clarify the meaning of sparse events here: “At the same time, SV breakpoint patterns in relation to genes represent sparse events in cancer genomic data, whereby for a given gene typically <4% of tumors may have nearby SV breakpoints uninvolved with amplification (Figure 2c).”

Methods: Line 646 – “All but 24 of the 1426 tumors.” – Again, confusion with numbers - in the previous section “1413 tumors in 636 total, of which 1294 had proteomic data and of which 988 had DNA methylation data” – where from the 1426 came from?...Line 658 “Of the 1307 tumors with proteomic data” contradicts with line 646 “of which 1294 had proteomic data”...Line 780 Fusion calls by RNA chimeric reads were available for 1192 of the 1307 tumors in our combined cohort with SV and protein expression data (Table S1).

The above statements from the manuscript are correct as written. The 1426 tumors are noted throughout the manuscript and are introduced in the first statement in the Results section: “We assembled a compendium dataset of combined WGS and gene expression data on 1426 human tumors (representing 1409 patients).” In the revised manuscript, we also note the 1426 number in Methods to go with the above statements.

The complete 1294 statement is as follows: “Combined WGS and RNA-seq profiling was compiled for 1413 tumors in total, of which 1294 had proteomic data and of which 988 had DNA methylation data (Illumina MethylationEPIC platform) not analyzed by our group previously.” So, of the 1413 tumors with BOTH mRNA and WGS data (from the above-noted 1426 tumors), 1294 tumors have protein+mRNA+WGS. The 1307 tumors noted throughout the manuscript have protein+WGS, though 13 of these tumors do not have mRNA data. One can use the Excel data filters in Data File S1 to get the same data availability numbers reported in the manuscript. Also, we have added a new supplementary figure (Figure S1a) to graphically illustrate data platform availability across the 1426 tumors (also shown below).

There were 1192 tumors with fusion data out of the 1307 with protein data. Fusion calls were not available for all tumor samples with protein expression, as some samples did not have RNA data or had

RNA data by expression arrays instead of sequencing (and hence, would not have chimeric read results available) or did not have fusion algorithms applied in the public data portals. In the revision, we have added a note to this effect under the “Gene Fusions” section of Methods. Also, we have added a new supplementary figure (Figure S1b) to illustrate the overlap in samples represented between any two data platforms. From this figure (also shown below), one can get at the 1307 tumors with proteomic data, the 1413 tumors with RNA or transcriptomic data, the 988 tumors with DNA methylation data, the 1294 tumors with both proteomic and transcriptomic data, the 1192 number of tumors with both proteomic and gene fusion data

	total protein	phospho-protein	RNA	fusions by RNA-seq	small mutation	methylation (EPIC)
total protein	1307	1207	1294	1192	1302	869
phospho-protein	1207	1207	1203	1184	1207	869
RNA	1294	1203	1413	1311	1408	988
fusions by RNA-seq	1192	1184	1311	1311	1309	988
small mutation	1302	1207	1408	1309	1421	988
methylation (EPIC)	869	869	988	988	988	988

Minor corrections

phospho-proteomic -> phosphoproteomic

We are happy to go with whichever convention the editorial staff recommends. We have seen both used in the literature. Our previous Nature Communication paper⁷ did use “phospho-proteomic”.

altered tumor – means SV-altered tumor?

For Figure 4, we considered all alterations in addition to SV-associated alterations. For Figure 4, we first consider other types of alterations and then consider SV-associated alterations if no other alterations were found for the pathway-associated genes. In this way, we were conservative in describing how SVs would extend upon the numbers of tumors altered for a given pathway. Other alterations noted in the manuscript would focus on SV-associated alterations.

Line 186 – “We found somatic SVs and associated altered protein levels” – means both mRNA+protein?

As noted further below in the same paragraph, we considered SV-mediated altered up-regulation or gene disruption by protein data or alternatively by mRNA if protein data was not available. This approach would be particularly relevant for genes like TERT, which did not have much protein data outside of the breast cancer samples. But most SV-related genes had adequate protein data, so the statement in question would be correct as written, given our emphasis on proteins.

Line 216 SSV-associated -> SV-associated... Line 230 SSV breakpoints -> SV breakpoints

We have fixed this. Thank you.

Line 223-225 – only a fraction – how much?

The opening statement of the paragraph in question is expanded upon in that same paragraph with specific numbers, as also presented in Figure 5c: “Using a p-value cutoff of <0.01 (1 Mb region, with cancer type and CNA corrections), 2602 CGI methylation probes were positively correlated with nearby SSV breakpoints, of which 52 probes involved genes negatively correlated between mRNA expression and SV breakpoints ($p < 0.01$), a significant overlap ($p < 1E-8$, chi-squared test), with just 14 of the 52 probes also showing inverse correlation ($p < 0.05$) between protein and SV breakpoints across the 1307 tumors. The 14 CGI probes significant by protein included four for PTEN. Out of 517 CGI methylation probes negatively correlated with nearby SV breakpoints, 50 involved genes positively correlated between mRNA and SV breakpoints (overlap $p < 1E-4$, chi-squared test), of which 21 CGI probes involved positive correlations at the protein level.”

Figure 2 - bad quality

We don't know how this would be. Perhaps the version of the manuscript downloaded from the journal manuscript submission system had some issues. Our local version of the figure and the version of the manuscript we obtained from the submission system uses clean line art (using Adobe Illustrator) and would be of high resolution.

Figure 3d – the tumor type labels are unreadable

This should not be the case, based on our local version of the figure and the version of the manuscript we obtained from the journal manuscript submission system.

Legend Figure 4 – “Genomic rearrangements (represented in circos plots) involving altered expression” – altered expression both mRNA+protein?

As noted in the figure legend: “For parts a-c, SV events represent altered gene expression (by protein or alternatively by mRNA if protein data not available...” In addition, Table S5 includes the SVs associated with genes of interest, with associated expression for both protein and mRNA, for each of the genes represented in the circos plots of Figure 4a.

In all figures - color-coded cancer types are almost impossible to read

The color coding of cancer types is basically the same as that of our previous 2022 Nature Communications paper on the 2002 tumors.⁷ Except that we switched light green to cyan for pediatric brain tumors to better contrast it with glioblastoma. If there would be a better color scheme to use, we would be open to that. Representing so many cancer types can be challenging, particularly for color-challenged readers. But the cancer type information represented in the figures would be secondary. The main purpose of representing the different cancer types in the figures is to illustrate how the SV-related patterns would cut across different cancer types. If the reader needed to definitively know the specific cancer types relating to certain data points, all the underlying data would be represented in the Supplementary data files and Source Data, for one to look up this information.

Reviewer #2, expertise in structural variants, cancer genomics and bioinformatics, signalling pathway analysis:

Chen et al present a study describing the impact of somatic structural variation at both transcription and proteome level. Uniquely this paper includes not only WGS and RNA-seq data but also mass spectrometry data. This multi-omics approach to interpreting the effects of somatic structural variation has the potential to increasing our understanding of SVs affecting both coding and non-coding regions of the genome, however the lack of correlation between values makes interpretation challenging. After reading the manuscript I have the following comments.

We thank the reviewer for evaluating our manuscript. Our study represents a first look at just how many mRNA-level SV associations show up at the protein level. We have published our analytical approach to identify SV-expression associations at the mRNA level in several previous studies¹⁻⁶. We have examined SV-mRNA associations starting with the TCGA⁶ and PCAWG⁴ cohorts of mostly primary tumors, as well as examining a cohort of advanced and metastatic tumors¹ and another cohort of pediatric brain tumors², and we examined SV-methylation associations in the context of SV-mRNA associations⁵. It is just very recently, within the last year, that data on a sizeable number of tumors with combined protein, mRNA, and WGS (for SV detection) have become available in the public domain.

We would not see any discordance between protein-based and mRNA-based as a negative. In fact, proteomics is a key factor here in helping determine which somatic SVs associated with genes are important. Proteomic data, combined with transcriptomic data, are shown in our present study to greatly help refine the catalog of SV-mediated cis-regulatory alterations in cancer. Gene transcription data have previously helped demonstrate the functional impact of non-coding somatic alterations on nearby genes, including TERT, for which gene both recurrent point mutations and somatic SVs associate with increased expression^{4,12-14}. We can better prioritize the hundreds of genes with SV-associated altered mRNA expression if we know which also show up by protein analyses, which would be just a fraction—about 25-31%—of the mRNA-level associations. However, as we report in the manuscript revision, this overlap between the protein-level and mRNA-level associations is highly statistically significant.

After assembling a catalog of combined SV-mRNA and SV-protein associations, our study could identify important SV-expression associations by focusing on the following: well-established oncogenes and tumor suppressor genes (Figure 4), DNA methylation associations (Figure 5), gene-specific events involving fusions, enhancer hijacking or retrotransposon transposition (Figure 6), patient survival associations (Figure 7), and associations with cell line viability (Figure 8). We have expanded the Discussion section to cover the above in the revised manuscript.

Major remarks

1. The paper would be strengthened if authors could highlight what the added value is of the proteomics and what new knowledge is generated in comparison to the WGS & RNA results.

In the revised manuscript, we have expanded the Discussion section to highlight the added value of the proteomics compared to WGS-RNA results, e.g., as described below and in our preamble above.

2. One of the main findings is that only ~25% of genes with mRNA expression changes associated with SVs also have protein expression changes. This observation raises many questions. Is this expected given relatively poor overall correlation between protein and mRNA expression (page 5, lines 113-116)? Were there differences observed across the spectrum of SV types, since they can vary a lot in size and complexity? And did they see a better correlation between mRNA and protein expression for known driver alterations?

Our premise for this present study is that protein and mRNA levels often do not highly correlate (Figure 1c), which would contribute to observed disparities in results between mRNA and protein. In the manuscript revision, we further considered the set of 201 “concordant” genes with significant SV-associations for both protein and mRNA ($p < 0.05$ for both, $p < 0.01$ for either, using filtered mRNA dataset) and the 319 “discordant” genes with significant SV-association for mRNA but not protein (mRNA $p < 0.01$, protein $p > 0.05$). The 201 concordant genes had higher mRNA-protein correlations on average across the 1307 tumors than the 319 discordant genes (average Pearson’s r of 0.49 versus 0.36, $p < 1E-19$ by t -test). So, the discordant genes do tend to have worse (though still statistically significant) protein-mRNA correlations overall.

We also find that the 201 genes and the 319 genes did not significantly differ in terms of intrachromosomal SV size (6.4 versus 6.2 Mb, respectively, $p = 0.44$, t -test on logged values) or of SV class representation (translocation, deletion, amplification, inversion, insertion, see plot below). We have noted the above in the manuscript revision.

In terms of enrichment for driver alterations, we note in the manuscript revision that the 201 concordant genes, but not the 301 discordant genes, were statistically enriched ($p = 0.03$, one-sided Fisher’s exact test) for genes found elsewhere to harbor somatic SNV hotspots¹⁵, involving eight genes: CDK4, EGFR, ERBB2, KRAS, PLCB3, S100A3, TACC3, and COBL.

3. Following on from point 2, How do SVs compare to other somatic mutation types in this regard? Do the authors expect a difference in predictability of effect on expression between SVs and other mutations ‘in cis’.

For a few genes, non-coding somatic point mutations are associated with gene up-regulation, most notably for the TERT gene. The PCAWG consortium surveyed recurrent non-coding driver SNVs¹⁶, but they found very few genes besides TERT impacted in this way that would represent drivers. In examining the supplemental results from the PCAWG study, just 11 genes (TERT, HOXB5, HIST1H2AM, WDR74, PTGIR, HIST1H2AE, GOLGA5, NFKBIZ, MTG2, RPPH1, G031024) were annotated as drivers with non-coding SNVs that also showed significant associations ($p < 0.05$) with mRNA expression changes. In contrast, SV-associated cis-regulatory alterations do not need to be as precise as for SNV hotspots, where SV breakpoints involving over-expression may fall at different locations with respect to the gene across the impacted tumors, with different mechanisms of SV-mediated cis-regulator alterations potentially

being involved^{5,6}. Regarding *TERT*, *TERT*-associated promoter SNVs have consistently been observed to be mutually exclusive with SV-associated altered expression^{4,12}. We touch on the above in the expanded Discussion section in the manuscript revision.

4. In general, the authors found more associations between SVs and overexpression or upregulation of genes compared to downregulation of genes. However, SV breakpoints do not necessarily need to be accompanied by lower expression to be disruptive to a (tumor-suppressor) gene, especially if they fall within the gene body. Therefore what is now implemented in the methods might be overly strict towards tumor suppressor genes.

“for oncogenes, breakpoint falling with 1Mb of gene and associated with expression >0.4SD from the median across samples for the given tumor; for tumor suppressors, breakpoint falling within the gene body and expression <-0.4SD” (page 34, lines 809-812)

Can you repeat the analysis requiring either a breakpoint inside the gene or a breakpoint within 1Mb together with reduced expression?

Below, we explore the alternate analyses as requested by the reviewer. Our study's overall theme and basic approach is integrating SV breakpoints with gene and protein expression data. If there were a scenario where SVs may disrupt a gene without altering its expression pattern in some cases, such patterns would not be something our study approach would be optimized to identify. Consistent with our present study, in our previous studies, we have also found more associations with SVs and over-expression versus under-expression. We have found that phenomena such as enhancer hijacking or demethylation patterns are involved with the over-expressed genes. We have found that the under-expressed genes mostly involve SV breakpoints within the gene, disrupting it directly. Even though breakpoints outside the gene may also be associated with under-expression, we do not have a way to explain many of these broadly.

We first associated within-gene SV breakpoints as disrupting key tumor suppressor genes as part of our comprehensive study of the PI3K pathway across the TCGA datasets¹⁷. The figure below is taken from this previous study, where a limited number of cases (1093) had SV data by WGS. By protein data (using RPPA data, not MS data, for a very limited set of genes), we could see, for example, that PTEN had very low protein expression when examining cases with SV breakpoints falling within the gene.

We followed the PI3K pathway study with an SV-mRNA expression integrative study⁶, where we surveyed TCGA cases for alterations within specific pathways, similar to what we did in the present study. As highlighted in the figure below (Figure 4 from that previous study), when considering oncogene- or tumor suppressor-associated genes represented by canonical pathways, a high overlap was observed between tumor suppressor genes and genes with decreased expression associated with within-gene SV breakpoints:

In the pathway-level annotation of Figure 4, we required for SV-alteration of tumor suppressors that the SV breakpoint falls within the gene body with concordant expression $< -0.4SD$ from the sample median, consistent with our previous studies^{1,2,6}. The reviewer's concern is whether we are being too strict regarding the Figure 4 analyses specifically, which would not impact the other analyses presented in the manuscript. The pathway-level analyses involved nine tumor suppressor genes: *PTEN*, *STK11*, *RB1*, *CDKN2A*, *NF1*, *ARID2*, *BRD7*, *PHF10*, and *SMARCC2*. By definition, using less strict criteria for calling a gene as being SV-impacted will include additional tumors as being impacted. As requested, we explore some alternative criteria below, where we repeat the analysis requiring either a breakpoint inside the gene or a breakpoint within 1Mb combined with reduced expression. In the table below, the first two columns are the numbers of SV-impacted tumors for each of the nine tumor suppressor genes, as reported in our study. In the subsequent columns of the table, we tabulate how many additional tumors would be noted as SV impacted if either an SV breakpoint was within the gene without under-expression ("UE") or an SV breakpoint was within 1Mb with UE but with the breakpoint being not within the gene.

gene	# affected cases (out of 1426)							
	SV within gene, UE protein	SV within gene, UE RNA (no protein data)	SV within gene, no protein UE	SV 1MB + protein UE, outside gene	SV within gene, no UE, adds to pathway-altered cases	SV within gene, no UE, adds to pathway-altered cases, no low-level copy loss	SV 1MB+UE, outside gene, adds to pathway-altered cases	SV 1MB+UE, outside gene, adds to pathway-altered cases, no low-level copy loss
PTEN	18	5	12	15	7	2	4	4
STK11	2	6	7	24	5	4	15	13
RB1	16	1	7	12	3	3	4	3
CDKN2A	11	3	18	61	4	1	10	5
NF1	18	1	3	17	0	0	5	2
ARID2	3	0	8	22	5	5	10	6
BRD7	1	0	3	9	3	3	9	8
PHF10	2	0	0	16	0	0	13	12
SMARCC2	1	0	1	24	1	1	15	13

In doing the pathway-level annotation of SV-alteration versus other somatic alteration, we had taken a conservative approach of first counting all other types of somatic alteration in any other genes before noting pathway alteration by SV and not by other alteration type. This would fit within our overall conclusion that SVs would uniquely add to what typically utilized precision medicine approaches that just focus on gene mutation or gene copy would identify. As seen above, of the additional cases that could be identified using the more relaxed criteria, fewer of these would actually contribute to additional pathway-altered cases being identified. When we further consider that SVs associate with CNA, we find that many additional tumors, as called by the relaxed criteria, would show low-level copy loss (representing loss of one gene). If we remove tumors that would additionally contribute to pathway alteration by SVs but with the SV also associated with copy loss, there are even fewer additional tumors involved (SV within the gene, no UE, third column from the end; SV within 1Mb and UE, last column). In addition, some genes, such as STK11 and SMARCC2, had significant SV-expression associations for within-gene SV breakpoints but NOT for the 1Mb region.

In summary, using relaxed criteria would involve additional tumors being called as SV-impacted for specific pathways. However, the overall message of SVs uniquely contributing to pathway deregulation would remain the same. As stated in our manuscript, we used a conservative approach in calling SV-impacted tumors at the pathway level. For our study, we would prefer to use the same selection criteria for pathway annotation as our previous studies^{1,2,6}.

Minor remarks

1. At present the study seems to contradict itself, as it states “Proteogenomics identifies targetable non-coding alterations.” (page 2, line 50), Yet the paper focusses on gene-level analysis, and only considers SVs affecting genes.

We have edited the abstract to indicate that the non-coding alterations identified as targetable would be targetable by virtue of the associated deregulated genes. These targetable alterations would include those involved in the pathway-level annotations of Figure 4 and the genes whose knockout is sensitive to a fraction of cell lines (Figure 8).

2. In the introduction there are several mentions of “percentage” or “fraction” without giving the numbers (page 4, lines 90-97). Mentions of “only a fraction” implies that the fraction is small, but the numbers elsewhere indicate sizeable fractions such as ~30%. Please consider rephrasing and use more specific values.

As these statements are in the Introduction section, we save the specifics for the Results section. The above statements are broad by intention, referring to different sets of analysis results. In the context of our study, we consider fractions of ~25-30% as noteworthy. However, it may also depend on the perception of the reader, e.g., a glass half-empty versus half-full mindset, when considering the concordance of mRNA versus protein. On the one hand, there is a lot of agreement between mRNA and protein results, e.g., in terms of high significant overlap in the results. On the other hand, not everything observed at the mRNA level will necessarily be observed at the protein level, which is a point for the revised Discussion.

3. See above. Also at other locations in the text the phrase “only a fraction” is used without percentages and refers to a non-small fraction.

“Only a fraction of the genes significant for a given region at the mRNA level were significant at the protein level ..” (page 6, line 144)

“As anticipated, only a fraction of the chimeric-based fusions predictions ...” (page 7, line 169)

In the above statements and elsewhere in the manuscript, we start with a general conclusion, and then

within the same paragraph, we provide the specific numbers that underscore that statement. For example, “Only a fraction of the genes significant for a given region at the mRNA level were significant at the protein level... Out of 10087 genes with protein expression data for at least 400 tumors, 657 had a significant SV association for the 1Mb region ($p < 0.01$ with cancer type and CNA correction) at the mRNA level using the unfiltered dataset, of which just 170 (26%) had a corresponding association at the protein level ($p < 0.05$)...” And elsewhere: “As anticipated, only a fraction of the chimeric-based fusions predictions involved an SV breakpoint falling within the boundary of one or both genes. Out of 9459 candidate fusion events by RNA-seq, 3419 involved SV breakpoints (Figure 3a).”

4. It is not always clear what cohort is used for what analysis. Different names for cohorts currently used throughout the text are: protein-WGS cohort, CPTAC cohort, protein WGS compendium cohort, extended mRNA-WGS compendium cohort, 1426 tumor cohort. Can the authors add a supplementary figure describing the overlap between the different cohorts?

In the manuscript revision, we have added a Supplementary Figure S1a (also shown below) to illustrate the cancer types, data platforms, and cohorts represented in the 1426 tumor cohort. Of the 1426 tumors, 1294 were represented in the extended mRNA-WGS compendium cohort (this cohort involving tumors with survival data available, only one tumor per patient being represented).

Figure S1b (shown below) shows the overlap in samples represented between any two data platforms. The 1426-tumor cohort with SV data involves 1307 tumors with proteomic data, 1413 tumors with RNA or transcriptomic data, 988 tumors with DNA methylation data, 1294 tumors with both proteomic and transcriptomic data, and 1192 tumors with both proteomic and gene fusion data.

	total protein	phospho-protein	RNA	fusions by RNA-seq	small mutation	methylation (EPIC)
total protein	1307	1207	1294	1192	1302	869
phospho-protein	1207	1207	1203	1184	1207	869
RNA	1294	1203	1413	1311	1408	988
fusions by RNA-seq	1192	1184	1311	1311	1309	988
small mutation	1302	1207	1408	1309	1421	988
methylation (EPIC)	869	869	988	988	988	988

In addition, Table S1 provides a tumor-level sample annotation table as an Excel file, including the information represented below, as well as a list of datasets and samples used for the gene-level breakpoint pattern survival analysis.

5. Reference is missing for: “Consistent with previous observations, many more genes showed positive correlations with SV breakpoints (i.e., expression was higher when a nearby SV breakpoint was present) than negative correlations, the former including known oncogenes and the latter including tumor suppressor genes.” (page 6, lines 129-132)

For this statement, we now refer to our previous studies, which made the same broad observations at the mRNA level as seen in our present study¹⁻⁶.

6. The authors show enrichment of certain terms in the gene set with both protein and mRNA SV-associations, and also in the gene set with only mRNA and no protein SV-associations (page 7, lines 153-159). How do these compare to the background correlation between protein/mRNA expression levels overall? Could it be the case that the observed enrichments are due to genes that inherently show more or less correlation between their protein/mRNA expression levels?

Our premise for this present study is that protein and mRNA levels often do not highly correlate (Figure 1c), contributing to observed disparities in results between mRNA and protein. In the revised manuscript, we include the following regarding the 201- and 319-gene sets examined in Figure 2b: “On average, the 201 concordant genes had higher mRNA-protein correlations across the 1307 tumors than the 319 discordant genes (average Pearson’s r of 0.49 versus 0.36, $p < 1E-19$ by t -test).” So, the discordant genes tend to have worse (though still statistically significant) protein-mRNA correlations overall. However, these differences would only partially explain the GO term enrichment results. Elsewhere, we have found that certain gene categories like ribosomes tend to correlate poorly between protein and mRNA, though such categories did not show up in the Figure 2b GO term enrichment results.

7. Its currently unclear what is the main message and novelty is of the gene fusion section? Please rephrase, and note gene fusions don’t need over- or underexpression to contribute to tumorigenesis.

We have noted this in the manuscript revision. We note that, while in theory, gene fusions contributing to tumorigenesis would not necessarily need to involve altered gene expression, we find with the fusion analysis that many predicted fusions involving protein over-expression include fusions observed elsewhere by RNA data (e.g., fusions by COSMIC or GENIE, Figure 3c).

8. SSV instead of SV is used on on page 9, line 215 and page 10, line 230.

We have fixed this. Thank you.

9. More details are required of the definition of overexpression or upregulation used, as it is currently unclear. At the start of the manuscript “overexpression” seems to be defined as significant by SVExpress and in the section "Mechanisms of SV-mediated altered protein expression" it seems to be defined in another way: "protein over-expression ($>0.4SD$ from sample median" (page 10, line 245).

The gene-level associations by SVExpress are distinct from the sample-centric analyses (e.g., the fusion analyses and the enhancer hijacking analyses). In the revised manuscript, we note explicitly that for the gene-level analyses, altered expression is relative to tumors without breakpoints and that for the fusion analyses, over-expression involves the impacted sample.

10. The abstract requires some rewriting as it does not seem to represent the contents of the manuscript as

it focuses very heavily on non-coding variants and does not mention seemingly important analyses such as the gene sets identified in section.

We have edited the abstract to indicate that the non-coding alterations identified as targetable would be targetable by virtue of the associated deregulated genes. These targetable alterations would include those involved in the pathway-level annotations of Figure 4 and the genes whose knockout is sensitive to a fraction of cell lines (Figure 8). While our present study focuses on SVs, one larger implication here is that proteomic data should be factored into studies of other types of non-coding alterations, where possible.

Reviewer #3, expertise in proteogenomics and cancer (Remarks to the Author):

This paper integrates proteogenomics data from multiple cohorts (e.g. CPTAC, APOLLO, TCGA, ICGC, CBTN) to study the impact of somatic structural variation (SV) on cancer proteome. It evaluates SV from various aspects, such as their impact on protein levels, pathways, DNA methylations, and survival. It also has a dedicated evaluation for gene fusion, a critical type of SV, and explores the mechanisms of SV-mediated alteration of protein levels. Finally, using data from cell line experiments, it highlights 33 genes whose knockout is sensitive to a fraction of cell lines.

We thank the reviewer for evaluating our manuscript. Our study represents a first look at just how many mRNA-level SV associations show up at the protein level. We have published our analytical approach to identify SV-expression associations at the mRNA level in several previous studies¹⁻⁶. We have examined SV-mRNA associations starting with the TCGA⁶ and PCAWG⁴ cohorts of mostly primary tumors, as well as examining a cohort of advanced and metastatic tumors¹ and another cohort of pediatric brain tumors², and we examined SV-methylation associations in the context of SV-mRNA associations⁵. It is just very recently, within the last year, that data on a sizeable number of tumors with combined protein, mRNA, and WGS (for SV detection) have become available in the public domain.

1. This study involves many combined analysis datasets (e.g. WGS - proteome, WGS – translation, WGS-methylation, survival data) with different sample sizes. It's very confusing to figure out what analysis is done in which data set on how many genes using what significant calling. An overview figure for sample availability, data types and analyses is helpful.

In the manuscript revision, we have added a Supplementary Figure S1 to illustrate the cancer types, data platforms, and cohorts represented in the 1426 tumor cohort. Reviewer #2 suggested including such an overview figure in the supplemental. Figure S1a (shown below) represents data platforms, cohorts, and cancer types represented across the 1426-tumor cohort of combined SV and expression data (expression by protein or mRNA if protein not available).

Figure S1b (shown below) shows the overlap in samples represented between any two data platforms. The 1426-tumor cohort with SV data involves 1307 tumors with proteomic data, 1413 tumors with RNA or transcriptomic data, 988 tumors with DNA methylation data, 1294 tumors with both proteomic and transcriptomic data, and 1192 tumors with both proteomic and gene fusion data.

	total protein	phospho-protein	RNA	fusions by RNA-seq	small mutation	methylation (EPIC)
total protein	1307	1207	1294	1192	1302	869
phospho-protein	1207	1207	1203	1184	1207	869
RNA	1294	1203	1413	1311	1408	988
fusions by RNA-seq	1192	1184	1311	1311	1309	988
small mutation	1302	1207	1408	1309	1421	988
methylation (EPIC)	869	869	988	988	988	988

In addition, Table S1 provides a tumor-level sample annotation table as an Excel file, including the information represented below, as well as a list of datasets and samples used for the gene-level breakpoint pattern survival analysis.

2. Proteomic studies uses LC-MS/MS profiling technology, which quantifies relative abundance of a protein in one sample relative to the same protein in other samples in the same experiment/study. It's questionable that your integrated data (with median/standard deviation based normalization) has no batch effect. There is no evaluation of the batch effect in your results. One potential way to evaluate batch effect is to compare data of the same cancer type from different studies. For example, CPTAC and APOLLO both have lung adenocarcinoma samples. Can you plot their principle components (PCs) or UMAP in the combined data to demonstrate these two studies do not form clusters (batch effects)?

By design, our normalization of the expression datasets would not lead to any batch effects, as demonstrated below and in the revised manuscript. As noted in the manuscript, we assembled our expression compendiums from various published studies. Within each study's dataset, we normalized proteomic data for downstream analyses noted above (transforming log-transformed values to standard deviations from the median). We have utilized this approach successfully in our previous studies collating expression data from multiple sources^{7,18-20}. A gold standard for demonstrating the absence of batch effects would be to carry out unsupervised clustering of the sample profiles and show how they cluster by biology rather than by batch. In fact, we did show this in our recent study of 2002 tumor proteomic profiles⁷, where we identified 11 pan-cancer proteome-based subtypes cutting across cancer type and laboratory. In another recent study, we collated the transcriptomes of patient-derived xenografts and patient tumor metastases, and these data collectively represent 38 studies and over 3,000 patients and 4,000 tumors. With these data, we identified four expression-based subtypes of metastasis transcending tumor lineage and cutting across all the various individual studies involved.

The figure below is from our previous study of the various urologic cancer types in TCGA²¹, and it illustrates how our normalization works at the gene level. The figure represents the expression of PDCDI mRNA (PDI, expressed in lymphocytes), before and after the normalization. While absolute PDCDI expression, on average, varies considerably between cancer types (left), normalizing the expression within each cancer type would serve to subtract out such differences (right). As intended, by normalizing expression within each cancer type and each proteomic dataset, neither tissue-dominant differences nor inter-laboratory batch effects would drive the downstream analysis results. Our SV-expression analytical approach was intended to identify expression outliers involving a small fraction of sample profiles, which outliers would remain in the data after normalization. In practice, batch effects involve widespread significant expression differences according to batch. But with our normalization, there are no differences in expression between the studies, as the median of each gene within each study is zero.

In the revised manuscript, we clustered the 1307 proteomic profiles using ConsensusClusterPlus²², as done in our previous 2002-tumor proteomics study. As seen in the new Supplementary Figure S1c (also shown below), the $k=9$ subtypes, called by ConsensusClusterPlus, do not cluster by cohort or cancer type. Each subtype (except for k_9 , which represents our previously identified s_{11} subtype consisting of brain tumors) has lung adenocarcinoma profiles from both CPTAC and APOLLO cohorts. (These results use k -means clustering, not hierarchical clustering. So, we can sort by cancer type and cohort within each subtype, but the samples do not automatically segregate by study within each subtype.)

In addition, as requested, we carried out principal component analysis (PCA) for the combined set of lung adenocarcinomas. The new Supplemental Figure S1g (also shown below) provides PCA results using either the proteomic data (left) or the transcriptomic data (right). As expected, the sample profiles do not segregate by batch as defined by cohort, but rather the CPTAC and APOLLO samples are mixed together in the PCA plots.

3. Similarly, can you evaluate if transcriptomics and other data types have no batch effects after your normalization? For example, both CPTAC and CBTN have pediatric brain tumor. Do you see samples from these two studies cluster together (batch effect) in PC/UMAP plots or simple clustering analysis?

In fact, the CBTN proteomic profiling by done in conjunction with CPTAC²³, so there are no independent CPTAC and CBTN pediatric brain tumor cohorts. It's all one cohort. However, a handful of medulloblastomas in our compendium were generated by ICGC (n=18 tumors). If one examines the k9 subtype in the above heat map figure from Supplementary Figure S1c, which subtype represents our previous s11 subtype featuring brain tumors⁷, one can see mostly CBTN tumors (light green) with a sliver of ICGC medulloblastomas (grey).

The PCA plots under response to comment #2 include mRNA as well as proteomic results. We also clustered the mRNA profiles in the same way that we did the proteomic profiles. As seen in the new Supplementary Figure S1e (also shown below), the k=9 subtypes, as called by ConsensusClusterPlus on the mRNA data, do not cluster by cohort or cancer type. In contrast to the other data platforms, the DNA methylation data were all generated uniformly in the same laboratory by CPTAC, so batch effects would not be an issue there. In addition, all our linear regression models incorporated cancer type as a covariate, so any cancer type or cohort-specific differences would presumably be corrected for in the statistical modeling.

4. A number of SV-RNA associations are not observed at SV-protein level, probably because SV-RNA analysis has bigger sample size and thus greater power to identify signals. How do you differentiate the biological difference from power difference in your RNA vs protein comparison?

We did correct for the missing samples in the protein dataset when doing direct protein-mRNA comparisons. In the revised manuscript, we have rewritten the Results sections to describe better our use of an mRNA dataset filtered for expression values not represented in the protein dataset, as we also briefly describe below.

When comparing gene-level SV associations common between mRNA and protein, we generated two sets of gene-level SV-mRNA associations, using both the complete mRNA compendium with all available expression data and a “filtered” mRNA dataset with any values not represented in the protein dataset filtered out. With the filtered mRNA dataset, any disparate results observed between protein and mRNA should have less to do with the diminished power for some genes due to lack of detection by proteomics. This puts protein and mRNA on a more level playing field for the purposes of direct comparisons of the respective results. The manuscript revision includes a new Supplementary Figures S2a and S2b (shown below), which graph the numbers of significant genes (FDR < 0.1) showing a correlation between mRNA expression and associated SV event. S2a shows results for the full mRNA dataset, while S2b shows results for the filtered mRNA dataset. Numbers above and below the zero point of the y-axis denote positively and negatively correlated genes, respectively.

The results for the filtered mRNA dataset would help assess how missing protein values could contribute to disparate results between the protein and complete or unfiltered mRNA datasets. As observed, there are somewhat fewer significant genes at a cutoff of $FDR < 10\%$ when using the filtered dataset, as the filtered mRNA data by design would have less data and therefore less statistical power for some genes.

Out of the 10087 genes with protein expression data for at least 400 tumors, 657 had a significant SV association for the 1Mb region ($p < 0.01$ by linear model with cancer type and CNA correction) at the mRNA level using the unfiltered dataset, of which just 170 (26%) had a corresponding association at the protein level ($p < 0.05$), this overlap being highly statistically significant (chance expected overlap of 31 genes, one-sided Fisher's exact test $p < 1E-60$ for over-expressed genes, $p < 1E-13$ for under-expressed genes). Using the filtered mRNA dataset, 574 genes had SV associations at the mRNA level ($p < 0.01$), of which 180 (31%) were concordant by protein analysis ($p < 0.05$). So, while the filtered mRNA dataset is closer to being "fair" in comparing protein- versus mRNA-based associations, the overall findings of just a fraction of the SV-mRNA associations showing up at the protein level are basically the same.

Regarding the reviewer's question about how we might differentiate the biological difference from the power difference in the RNA versus protein comparison, we cannot differentiate the two from a statistics standpoint. As stated in the manuscript, any observed discordances between protein and mRNA can stem in part from biology (e.g., post-transcriptional regulation) as well as from statistical false negatives. There is no way for us to put a meaningful biological interpretation on a null p-value, as a $p > 0.05$, for example, does not necessarily mean "no change." It just means the null hypothesis was not refuted at that significance level. This is a point we also address in the Discussion section of the revision.

5. It seems you did not report any SV-protein unique association that's not reflected at the RNA level. Then, what unique contribution does this proteomic-focused study make? Can you demonstrate what additional biological discoveries do we have by adding proteomics data into existing multi-omics data?

From Figure 2a, we can see a lot of gene-level SV-mRNA associations do not show up in the SV-protein associations, but not many in the other direction (protein-level associations not showing up in mRNA results). Figure 2a uses $FDR < 10\%$ for EITHER mRNA or protein datasets, where even a $p < 0.05$ for the other analyte will show up as light red or blue.

In comparing SV-mRNA associations with SV-protein associations, our emphasis is on associations found for both mRNA and protein. In the manuscript revision (Results, 3rd paragraph), we explicitly consider protein associations that do not show up in the mRNA data. Using a more stringent statistical cutoff for protein, at $p < 0.01$, with a less stringent cutoff for mRNA, at $p < 0.05$ for the filtered dataset, a relatively higher percentage—57%—of the 245 statistically significant proteins were also significant by mRNA. This result contrasts with results using $p < 0.01$ for mRNA and $p < 0.05$ for protein, where just 31% of the top mRNA associations show up for protein. As noted in the manuscript, any observed discordances between protein and mRNA can stem in part from biology (e.g., post-transcriptional regulation) as well as from statistical false negatives, and we can't necessarily distinguish between the two. We can appreciate how mRNA does not always translate into protein levels, but we may not have a good explanation for associations found for protein but not mRNA, though these may largely represent false positives or false negatives.

Regarding the significance of adding protein into these SV expression analyses, the proteomic data would greatly help in refining the catalog of SV-mediated cis-regulatory alterations in cancer. Our present study would be consistent with other recent cancer proteogenomics studies in demonstrating the need for proteomics to be a component along with mRNA in multi-omic studies. As part of PCAWG and other studies, our group and others have systematically cataloged genes recurrently altered in expression by SVs, through gene fusion or altered cis-regulation, across cancers of diverse lineages, but using mRNA

data not protein data.^{4,5,16,24} This present study represents a first look at just how many of these mRNA-level SV associations show up at the protein level. We can better prioritize the hundreds of genes with SV-associated altered mRNA expression if we know which also show up by protein analyses, which would be a fraction—about 25-31%—of the mRNA-level associations. This situation is similar to the challenge of prioritizing other types of somatic mutations, such as CNA, where there would be many functionally relevant alterations along with more “passenger” events. Of course, future datasets with much greater numbers of tumors with proteomic data would considerably help further refine the catalog. Still, our present study would be pointing us in that direction.

6. SV-RNA association analysis controlled for the effect of CNA, while somatic alternations also include mutation. Why mutation is not controlled in the analysis?

Regarding small mutations (SNVs and indels), these would not typically be associated with altered expression to the same extent as CNA. For oncogenes (e.g., KRAS or EGFR family genes), SNV mutations within the gene itself may result in hyperactivity of the gene but would not lead to over-expression. For tumor suppressors, SVs within the gene can disrupt expression^{6,17}, and these might show up in exome analysis as indels but could be considered as stemming from the genomic rearrangement, so we would count these as SVs. For a few genes, non-coding somatic point mutations are associated with gene up-regulation, most notably for the TERT gene. The PCAWG consortium surveyed for recurrent non-coding driver SNVs¹⁶, but they found very few genes besides TERT impacted in this way that represent drivers, and such alterations are generally mutually exclusive from SV-associated altered expression^{4,12}. In contrast to SNVs, SV breakpoints are strongly associated with CNA, so we would be interested in SV-expression associations that gene-level CNA would not explain away.

7. Line 146-148: Why raw p-value is used for claiming significance, without considering of the multiple testing? Why different p-value cutoffs (0.01 vs 0.05) are used for mRNA and protein levels?

We relied on a stricter FDR cutoff for defining top genes when carrying out gene-level global molecular analyses for a single analyte (e.g., gene-level SV-protein associations or SV-mRNA associations). However, when overlapping different top-gene results sets across multiple independent analyses (e.g., gene-level SV-expression associations as observed in both protein and mRNA data), we used a more relaxed p-value cutoff for each individual analysis to limit false negatives. As reported, the degree of gene set overlap across independent analyses was often highly statistically significant. This practice would be consistent with our previous studies identifying genes with demonstrated functional roles as originally identified using integrative analyses^{25,26}, and with standard analytical methods like GSEA that can identify enrichment patterns that would be missed using overly strict FDR cutoffs^{27,28}.

Alternatively, if we had used strict FDR cutoffs of <10% in identifying the concordance between protein and mRNA SV-expression associations, then of the 336 genes with FDR<10% at the mRNA level (1Mb region, using the filtered mRNA dataset, see our response to point #4), only 22 (6.5%) are similarly significant for protein at FDR<10%. So, the implication here is that only 6.5% of the SV-mRNA associations are reflected at the protein level, the concordance being even worse than our reported 31%. But using the 6.5% number is not accurate because, as reported, many proteins are, in fact, statistically significant at a less stringent cutoff. Selection criteria is always a balance between false positives and false negatives. In comparing mRNA-based to protein-based results in this instance, we want to be generous with respect to protein, particularly as proteomic data do not have data for tumors for many genes. As we report in the manuscript revision, out of 10087 genes with protein data for at least 400 tumors, 657 had a significant SV association for the 1Mb region ($p<0.01$ with cancer type and CNA correction) at the mRNA level using the unfiltered dataset, of which just 170 (26%) had a corresponding association at the protein level ($p<0.05$), this overlap, however, being highly statistically significant (chance expected overlap of 31 genes, one-sided Fisher's exact test $p<1E-60$ for over-expressed genes,

p<1E-13 for under-expressed genes). With the filtered mRNA dataset, 574 genes had SV associations at the mRNA level (*p*<0.01), of which 180 (31%) were concordant by protein analysis (*p*<0.05, Figures 2b and 2c and Table S3). We expect most of the overlapping mRNA/protein SV-expression associations to be true positives, given that the overlap greatly exceeds chance expected.

*Our practice here would be consistent with our previous studies identifying genes with demonstrated functional roles as originally identified using integrative analyses^{25,26}. In our recent study involving tumor grade correlates at the protein versus mRNA levels¹⁸, we mined the proteomic grade correlations, and we identified protein kinases—including MAP3K2, MASTL, and TTK—that by experiment had a functional impact in vitro in uterine endometrial cancer cells. These proteins were selected using nominal *p*-values, rather than FDR, yet functional roles could be confirmed. In another series of studies^{26,29,30}, we carried out bioinformatics analysis of multiple molecular profiling datasets involving cross-species comparison of the genes overexpressed in autochthonous genetically engineered metastatic murine lung tumors and syngeneic lung cancer models, intersected with human copy number amplifications by TCGA, identified a set of 217 putative driver genes in lung cancer²⁶. This set of 217 genes was arrived at using nominal *p*-values, not strict FDR cutoffs. Otherwise, we would not have come up with anything. Nevertheless, subsequent experiments have demonstrated functional roles for several of these genes, including GATAD2B²⁶, TMEM106B³⁰, IMPAD1²⁹, and KDELR2²⁹. In a similar vein, one of the first studies to showcase the GSEA method²⁸ involved a gene expression profiling dataset where no genes were found significantly expressed using strict FDR cutoffs, while GSEA showed a robust pattern or coordinate expression involving OX-PHOS genes, their functional role then being experimentally confirmed. When carrying out integrative analyses using cutoffs based on nominal *p*-values, the FDR, as estimated for a single analysis, does not apply to the overlapping genes spanning multiple analyses. If one wanted to use strict FDR instead of nominal *p*-values for integrative analyses, then either the datasets involved should have very large numbers of samples (~10,000) for adequate power to get small *p*-values, or the authors should be very candid as to how the results and their interpretation would be severely impacted due to very high false negative rates.*

8. Gene fusion is of high translational relevance, as they can be important candidates for drug target. Your study identifies 1055 and 386 gene fusions with mRNA/protein and mRNA associations. How does one make use of this long list? For example, who are casual, who are important for survival or other clinical outcomes, who should be potential drug targets for future studies?

Our supplementary Table S4 provides, as an Excel file, the entire set of the 9459 candidate fusion events examined in Figure 4, with the corresponding filtering criteria used to derive the fusion events with SV and expression support. Most of the more recurrent fusions highlighted in Figure 3a have been observed or described elsewhere. Most of the above fusions occur in just one or two tumors, so there would be no way to derive associations of these events with survival or drug response.

9. What does “SV-mediated” alterations means? Who is the predictor, who is the outcome, and who is the mediator? What analysis guarantees SV is a mediator, instead of a predictor or effect modifier, in the association?

We have changed “SV-mediated” to “SV-associated” throughout the manuscript.

10. Line 261-272. Why no multiple testing is considered for this large screening of survival associated genes? At 10%FDR, are you able to see anything associated with survival outcome?

*For the 34 genes from Figure 7a, perhaps two have an FDR of <10% for the SV-protein expression analysis. But using such a stringent cutoff leads to type II error in this situation. As noted in the manuscript: “When overlapping different results sets, we used a more relaxed *p*-value cutoff to limit false*

negatives, as the degree of gene set overlap itself was significant and yielded significant results in multiple external datasets.” We used a very similar approach in our recent study of SV-associated survival patterns across 570 advanced and metastatic cancers¹, where we also used relaxed statistical cutoffs to define a set of 80 genes with SV-expression association, SV survival association, and expression association for the 570-patient cohort, which signature associated with patient survival across multiple independent cohorts representing diverse cancer types. In line with our responses to comment #7 above, False Discovery Rate is not a p-value but rather provides an estimate of what percent of the gene set at a given FDR cutoff might be false positives for that specific analysis. An FDR of 50% would indicate that 50% of the genes would be true positives. But we would need to bring in additional information to know which genes would likely represent true positives, as we do in the integrative analyses. The 34 genes must cross multiple hurdles, and the FDRs represented by each individual analysis cannot be correctly applied to the 34 genes.

11. Figure 7B-C: Where is 34-gene signature coming from?

As noted in parts b and c, the 34-gene signature is described in part a of Figure 7. We have expanded the Figure 7a legend to describe better the analysis leading to the 34 genes, as also described in the Results section. Briefly, gene-level association with patient survival, at the levels of both mRNA and nearby SV breakpoints, was examined in a cohort of 3084 patients with combined SV-survival data from publicly available datasets (Table S1). Of 516 genes with positive SV-protein expression associations ($p < 0.05$, using 1Mb region, with corrections for tumor type and gene-level CNA), 34 genes had both nearby SV breakpoints and expression associated with worse patient outcome in the 3084-patient cohort, this 34-gene overlap being statistically significant. Importantly, the 34-gene signature was evaluated in datasets not used to generate the 34 genes, where, as a group, the 34 genes associate with worse prognosis in these other datasets.

12. This study has a great width on SV analysis, but not much in depth. It identifies a number of long list of significant genes from different analyses, but it’s hard to understand who are important and what can we do with these genes?

Proteomics is a key factor here in determining which somatic SVs associated with genes would be important. Proteomic data, combined with transcriptomic data, are shown in our present study to greatly help refine the catalog of SV-mediated cis-regulatory alterations in cancer. Gene transcription data have previously helped demonstrate the functional impact of non-coding somatic alterations on nearby genes, including TERT, for which gene both recurrent point mutations and somatic SVs associate with increased expression^{4,12-14}. Combined WGS and MS-based proteomic data on appreciable numbers of human tumors have only recently been available in the public domain. Our study represents a first look at just how many of these mRNA-level SV associations show up at the protein level. We can better prioritize the hundreds of genes with SV-associated altered mRNA expression if we know which also show up by protein analyses, which would be just a fraction—about 25-31%—of the mRNA-level associations.

After assembling a catalog of combined SV-mRNA and SV-protein associations, our study could identify important SV-expression associations by focusing on the following: well-established oncogenes and tumor suppressor genes (Figure 4), DNA methylation associations (Figure 5), gene-specific events involving fusions, enhancer hijacking or retrotransposon transposition (Figure 6), patient survival associations (Figure 7), and associations with cell line viability (Figure 8). To draw these interesting associations between SV-protein relationships with outside datasets, we often use relaxed statistical cutoffs to lower our false negatives to identify patterns of enrichment by integrative analyses better (see our response to comment #7). We have expanded the Discussion section to cover the above in the revised manuscript.

Minor

13. Line 298. Please right out exact p-value in scientific format, instead of using $p \sim 0$.

Our statistical software gave the p-value as 0, likely because the true number would be below the numerical limits of the software. We can safely report the p-value as being $< 1E-100$, which we do in the manuscript revision.

References:

- 1 Zhang, Y. *et al.* Rearrangement-mediated cis-regulatory alterations in advanced patient tumors reveal interactions with therapy. *Cell Rep* **37**, 110023 (2021).
- 2 Zhang, Y., Chen, F., Donehower, L., Scheurer, M. & Creighton, C. A pediatric brain tumor atlas of genes deregulated by somatic genomic rearrangement. *Nat Commun* **12**, 937 (2021).
- 3 Zhang, Y., Chen, F. & Creighton, C. SVExpress: identifying gene features altered recurrently in expression with nearby structural variant breakpoints. *BMC Bioinformatics* **22**, 135 (2021).
- 4 Zhang, Y. *et al.* High-coverage whole-genome analysis of 1220 cancers reveals hundreds of genes deregulated by rearrangement-mediated cis-regulatory alterations. *Nat Commun* **11**, 736 (2020).
- 5 Zhang, Y. *et al.* Global impact of somatic structural variation on the DNA methylome of human cancers. *Genome biology* **20**, 209 (2019).
- 6 Zhang, Y. *et al.* A Pan-Cancer Compendium of Genes Deregulated by Somatic Genomic Rearrangement across More Than 1,400 Cases. *Cell Reports* **24**, 515-527 (2018).
- 7 Zhang, Y., Chen, F., Chandrashekar, D., Varambally, S. & Creighton, C. Proteogenomic characterization of 2002 human cancers reveals pan-cancer molecular subtypes and associated pathways. *Nat Commun* **13**, 2669, doi:10.1038/s41467-022-30342-3 (2022).
- 8 Chen, G. *et al.* Discordant protein and mRNA expression in lung adenocarcinomas. *Mol Cell Proteomics* **1**, 304-313, doi:10.1074/mcp.m200008-mcp200 (2002).
- 9 Vasaikar, S. *et al.* Proteogenomic Analysis of Human Colon Cancer Reveals New Therapeutic Opportunities. *Cell* **177**, 1035-1049.e1019 (2019).
- 10 Clark, D. *et al.* Integrated Proteogenomic Characterization of Clear Cell Renal Cell Carcinoma. *Cell* **179**, 964-983 (2019).
- 11 Yang, L. *et al.* Analyzing Somatic Genome Rearrangements in Human Cancers by Using Whole-Exome Sequencing. *Am J Hum Genet* **98**, 843-856 (2016).
- 12 Davis, C. *et al.* The somatic genomic landscape of chromophobe renal cell carcinoma. *Cancer Cell* **26**, 319-330 (2014).
- 13 Huang, F. W. *et al.* Highly recurrent TERT promoter mutations in human melanoma. *Science* **339**, 957-959, doi:10.1126/science.1229259 (2013).
- 14 Horn, S. *et al.* TERT promoter mutations in familial and sporadic melanoma. *Science* **339**, 959-961 (2013).
- 15 Chang, M. *et al.* Identifying recurrent mutations in cancer reveals widespread lineage diversity and mutational specificity. *Nat Biotechnol* **34**, 155-163 (2016).
- 16 Rheinbay, E. *et al.* Analyses of non-coding somatic drivers in 2,658 cancer whole genomes. *Nature* **578**, 102-111 (2020).
- 17 Zhang, Y. *et al.* A Pan-Cancer Proteogenomic Atlas of PI3K/AKT/mTOR Pathway Alterations. *Cancer Cell E-pub* **May 8** (2017).
- 18 Monsivais, D. *et al.* Mass-spectrometry-based proteomic correlates of grade and stage reveal pathways and kinases associated with aggressive human cancers. *Oncogene* **40**, 2081-2095, doi:10.1038/s41388-021-01681-0 (2021).
- 19 Chen, F., Chandrashekar, D., Varambally, S. & Creighton, C. Pan-cancer molecular subtypes revealed by mass-spectrometry-based proteomic characterization of more than 500 human cancers. *Nat Commun* **10**, 5679, doi:10.1038/s41467-019-13528-0 (2019).
- 20 Zhang, Y., Chen, F. & Creighton, C. Pan-cancer molecular subtypes of metastasis reveal distinct and evolving transcriptional programs. *Cell Rep Med* **4**, 100932, doi:10.1016/j.xcrm.2023.100932 (2023).

- 21 Chen, F. *et al.* Pan-urollogic cancer genomic subtypes that transcend tissue of origin. *Nat Commun* **8**, 199 (2017).
- 22 Wilkerson, M. & Hayes, D. ConsensusClusterPlus: a class discovery tool with confidence assessments and item tracking. *Bioinformatics* **26**, 1572-1573 (2010).
- 23 Petralia, F. *et al.* Integrated Proteogenomic Characterization across Major Histological Types of Pediatric Brain Cancer. *Cell* **183**, 1962-1985, doi:10.1016/j.cell.2020.10.044 (2020).
- 24 Pkawg_Transcriptome_Core_Group *et al.* Genomic basis for RNA alterations in cancer. *Nature* **578**, 129-136 (2020).
- 25 The_Cancer_Genome_Atlas_Research_Network. Comprehensive molecular characterization of clear cell renal cell carcinoma. *Nature* **499**, 43-49, doi:10.1038/nature12222 (2013).
- 26 Grzeskowiak, C. *et al.* In vivo screening identifies GATAD2B as a metastasis driver in KRAS-driven lung cancer. *Nat Commun* **9**, 273, doi:10.1038/s41467-018-04572-3 (2018).
- 27 Subramanian, A. *et al.* Gene set enrichment analysis: a knowledge-based approach for interpreting genome-wide expression profiles. *Proc Natl Acad Sci U S A* **102**, 15545-15550 (2005).
- 28 Mootha, V. K. *et al.* PGC-1alpha-responsive genes involved in oxidative phosphorylation are coordinately downregulated in human diabetes. *Nature genetics* **34**, 267-273 (2003).
- 29 Bajaj, R. *et al.* IMPAD1 and KDELR2 drive invasion and metastasis by enhancing Golgi-mediated secretion. *Oncogene* **39**, 5979-5994, doi:10.1038/s41388-020-01410-z (2020).
- 30 Kundu, S. *et al.* TMEM106B drives lung cancer metastasis by inducing TFEB-dependent lysosome synthesis and secretion of cathepsins. *Nat Commun* **9**, 2731, doi:10.1038/s41467-018-05013-x (2018).

Reviewers' Comments:

Reviewer #1:

Remarks to the Author:

The authors addresses all the issues raised in the first review.

Reviewer #2:

Remarks to the Author:

Overall the updates to the manuscript have satisfactory addressed points raised. It is much more explicit about what criteria used and where, plus more careful in the phrasing and conclusions. The most important points are 1) SVs are important and can dysregulate RNA+protein and 2) overlapping gene-level associations by both mRNA and protein gives more confidence / allows for using more relaxed statistical criteria. They end on "The catalog of SV-altered genes, provided by studies such as ours, could greatly inform on better identifying patients with tumors altered for targetable pathways in the clinical setting." In fact this is something that can be included more of in the title instead of "Global impact of somatic structural variation on the cancer proteome".

Reviewer #3:

Remarks to the Author:

The authors have adequately addressed most of my comments. The part I am still concerning is normalization of proteomics data (Comment 2).

Unlike RNAseq, which quantifies absolute counts, proteomics quantifies the relative abundance of a protein in a sample compared to the same protein in the reference sample. The abundances of two proteins within a sample is not comparable, so a rescale (median/sd) of the abundances do not necessarily provide comparable values across samples that come from different studies with no common reference. Therefore, methods in RNAseq normalization cannot be directly translated into proteomics, but should be careful evaluated. The proteomics PC plot provided in response letter does show that the lower right corner is mostly CPTAC samples while upper left corner is APOLLO samples, which is a sign of remaining batch effects. If authors associate proteins with batch, I suspect a number of proteins will be identified. Two things can be done to improve the analysis: (1) a better batch correction, and (2) an adjustment of batch in association analysis.

Our responses and comments are marked in red italics.

Reviewer #1, expertise in structural variants, cancer genomics and bioinformatics:

The authors addresses all the issues raised in the first review.

We thank the reviewer for evaluating our work.

Reviewer #2, expertise in structural variants, cancer genomics and bioinformatics, signalling pathway analysis:

Overall the updates to the manuscript have satisfactory addressed points raised. It is much more explicit about what criteria used and where, plus more careful in the phrasing and conclusions.

The most important points are 1) SVs are important and can dysregulate RNA+protein and 2) overlapping gene-level associations by both mRNA and protein gives more confidence / allows for using more relaxed statistical criteria. They end on “The catalog of SV-altered genes, provided by studies such as ours, could greatly inform on better identifying patients with tumors altered for targetable pathways in the clinical setting.”. In fact this is something that can be included more of in the title instead of “Global impact of somatic structural variation on the cancer proteome”.

We thank the reviewer for evaluating our work. We can confer with the editors regarding the title in light of the reviewer’s comments. In the manuscript, we put forth the prospect of better identifying patients for targetable pathways as a Discussion point for future studies, though we may not have achieved this in the present study for application towards the clinical setting in the near term. And we cover a range of topics in our study related to structural variation and the cancer proteome, hence the more broad title. We can think on this.

Reviewer #3, expertise in proteogenomics and cancer:

The authors have adequately addressed most of my comments. The part I am still concerning is normalization of proteomics data (Comment 2).

We appreciate the reviewer for being thoughtful and conscientious during the review process. Below, we further delve into the concerns regarding protein data normalization and any possible batch effects. In our study, we assembled a compendium dataset of mass spectrometry-based proteomics data of primary tumors from previous studies, with the data being generated by different laboratories. We, therefore, performed a normalization within each dataset that would effectively erase protein expression differences according to laboratory, analytical platform, or cancer type. We have used this approach in previous studies, including proteomics studies¹⁻⁵.

Unlike RNAseq, which quantifies absolute counts, proteomics quantifies the relative abundance of a protein in a sample compared to the same protein in the reference sample. The abundances of two proteins within a sample is not comparable, so a rescale (median/sd) of the abundances do not necessarily provide comparable values across samples that come from different studies with no common reference. Therefore, methods in RNAseq normalization cannot be directly translated into proteomics, but should be careful evaluated.

Our methodology does not assume that the datasets generated by different laboratories would provide comparable measures of expression, or that the datasets would even be generated using the same platform. In our recent study of pan-cancer molecular subtypes of metastases⁵, we made two compendium datasets of RNA expression, one of 2371 patient-derived xenografts drawn from 15 separate datasets and

another of 2405 patient metastases drawn from 24 separate datasets. The RNA datasets involved in that previous study were a mix of microarrays and RNA-seq, where the two platforms in no way would be comparable. However, after normalization, we could define robust molecular subtypes of metastasis that would not be confounded by laboratory-driven batch effects. There may be no perfect way to harmonize all these data, as in practice there are no perfect data. However, it is possible to derive a dataset where the inter-laboratory differences would not be dominant over the biology or would otherwise be able to explain away the associations observed. One caveat of our methodology is that any tissue-specific differences are not present, so comparisons between different cancer types (e.g., lung vs. breast, etc.) are not possible using our protein compendium dataset. However, our study is focused instead on identifying expression outliers involving a small fraction of sample profiles, which outliers would remain in the data after normalization.

The proteomics PC plot provided in response letter does show that the lower right corner is mostly CPTAC samples while upper left corner is APOLLO samples, which is a sign of remaining batch effects. If authors associate proteins with batch, I suspect a number of proteins will be identified.

Below, we have included the PCA plot, which was included in the manuscript revision as Figure S1g. Here, we have highlighted the cases in question on the left (proteomic data), whereby some cases in the upper left corner are from APOLLO, and other cases in the lower portion of the plot are from CPTAC.

While there is some separation, as noted, with PC2 representing only 8% of the variation in the dataset, we are not seeing any major batch effects from our perspective, as most samples are in the middle and mixed by batch. Furthermore, to the reviewer's point about associating proteins with batch, we compared proteomic profiles of APOLLO lung and CPTAC lung. Out of 8574 proteins shared between the two batches, just six were differential with $p < 0.01$ (t -test), where the chance expected at $p < 0.01$ by multiple testing would be ~ 86 . In the manuscript revision, we have noted the results of the above comparison in the legend of Supplementary Figure S1g.

Two things can be done to improve the analysis: (1) a better batch correction, and (2) an adjustment of batch in association analysis.

We further explore option #2 below. Option #1, a better batch correction method, would not be viable. Formal batch correction methods like Combat⁶ (which we have used in some of our previous studies, as appropriate, e.g., ref⁷) require that the biological or experimental groups are represented across different

batches. In our case, each cancer type (by tissue of origin) was contained within one or two batches (as defined by separate studies), and no batches involve multiple cancer types.

Regarding option #2, our protein-SV association analyses would already largely incorporate batch in the linear modeling, as the reviewer suggested. Our linear regression models evaluated significant associations when correcting for both cancer type and gene-level CNA. “Cancer type” represents most of the batches in our protein expression compendium. There are 12 tissue-based cancer types represented in our dataset: Breast, CRC, GBM, HNSC, LUAD, LUSC, OV, PAAD, PedBrain, PRAD, Renal, UCEC. The proteomic profiling of each cancer type was done in one or two batches, separate from the other cancer types. Even in the case of CPTAC, the data from the different cancer types were not done uniformly in the same laboratory within a comparable time frame. CPTAC proteomic data were generated across four or five laboratories over a period of several years, with each CPTAC-led study of a specific cancer type involving proteomic data being generated uniformly for that study.

In our study, SV-expression associations that would be better explained by cancer type versus SV breakpoint pattern would not be carried forward in our downstream analyses. Below is represented Figure S2a, which we had included in the previous revision. Results are shown for the mRNA dataset, but similar patterns are seen for the protein dataset. Figure S2a shows results from linear regression models that have no correction (blue) or that correct for cancer type but not for CNA (red). The model represented by the green bars below, which corrects for both cancer type and gene-level CNA, is what we used for all our downstream analyses. The plots below show a big drop in the numbers of significant genes when moving from a model that does not include CNA correction (red) to a model that does (green). For most of the genes in red, the SV breakpoints involve CNA, and the CNA patterns alone can explain the observed differential expression. For the model with CNA correction, the SV breakpoint patterns involving altered expression would include additional information not captured by CNA. Factoring in cancer type (which largely reflects batch) does not change the results much, with slightly fewer significant genes (going from blue to red), though this may be partly due to a more complex model.

If we explicitly consider batch instead of cancer type, we would have 15 batches from the 12 cancer types. As noted above, the lung cancer samples came from the respective CPTAC and APOLLO studies. The renal samples were done by CPTAC, but in separate Discovery⁸ versus Confirmatory⁹ studies, with a few years separating the two studies. Of the 216 Pedbrain tumors, 18 were carried out by ICGC and 198 by CPTAC in collaboration with CBTN. The 15 batches would be the following: Breast.CPTAC-Discovery, CRC.CPTAC-Discovery, GBM.CPTAC-Discovery, HNSC.CPTAC-Discovery, LUAD.APOLLO, LUAD.CPTAC-Discovery, LUSC.CPTAC-Discovery, OV.CPTAC-Discovery, PAAD.CPTAC-Discovery, PedBrain.CPTAC-CBTN, PedBrain.ICGC, PRAD.ICGC, Renal.CPTAC-Discovery, Renal.CPTAC-Confirmatory, and UCEC.CPTAC-Discovery.

How would the results change if we used the 15 batches instead of the 12 cancer types in our linear modeling of protein-SV associations? We explore this below, but the short answer is: very little. In addition to the linear models incorporating cancer type and CNA as covariates, used in the manuscript, we generated a separate set of linear models incorporating batch (instead of cancer type) and CNA as covariates. We generated results for the 1Mb, 100kb upstream, 100kb downstream, and within-gene genomic region windows. The plot below compares the significance of SV-impacted genes at the protein level (involving breakpoints within 1Mb of the gene), as plotted for the cancer type corrected results (y-axis) versus the batch corrected results (x-axis).

The above plot shows that the p-values generated by either model are comparable. Proteins that may miss a particular significance cutoff by one model would be just above the cut point and would be significant using a slightly relaxed cutoff. Below we show the numbers of significant proteins with either model for each genomic region examined, using cutoffs of FDR<10% and p<0.01.

# proteins with significant SV breakpoint associations			
region	cutoff	linear model (cancer type+CNA)	linear model (batch+CNA)
1Mb up- or downstream	FDR<10%	32	34
	p<0.01	245	241
gene body	FDR<10%	27	27
	p<0.01	203	201
0-100kb downstream	FDR<10%	39	41
	p<0.01	233	229
0-100kb upstream	FDR<10%	29	29
	p<0.01	227	233

The above chart shows that the numbers of significant proteins stay mostly the same between models. For some regions and cutoffs, the batch-corrected model might have a few more significant proteins, but for other regions and cutoffs, the cancer type-corrected model has a few more. Notably, we do not see any big drop off in the numbers of significant proteins when using batch over cancer type, e.g., like we see when incorporating CNA into the model (see Figure S2a above). The chart below features the proteins significant at $FDR < 10\%$ for the 1Mb region for one model but not the other. We can see that all these proteins could be considered significant by both models, e.g., the proteins would all meet a strict cutoff of $FDR \leq 11\%$, with robust p-values involved in every instance.

Symbol	cancer type+CNA model			batch+CNA model		
	1MB	1MB	1MB	1MB	1MB	1MB
	t	p	fdr	t	p	fdr
SEC62	3.592437	0.00034	0.097902	3.59573	0.000336	0.102553
B4GALT7	3.807425	0.000152	0.066647	3.547373	0.000414	0.106917
THG1L	3.599717	0.000333	0.098677	3.571417	0.000371	0.10375
STBD1	3.519595	0.00045	0.113248	3.593042	0.000341	0.098192
LRCH1	3.617106	0.000311	0.100951	3.640387	0.000284	0.092347

In our manuscript, we frequently highlight proteins of specific interest, including AKR1C1-4 and IGF2 (Figure 2); EGFR, KRAS, NF1, and PTEN (Figure 4); ANO1 and NID2 (Figure 5); G6PD (Figure 7); and CCND1, CDK2, and KDM2A (Figure 8). Below, we compare the protein-SV association p-values for these genes by the cancer type model versus the batch model. We can see that these genes are significant in both models. If we had applied the same statistical cutoffs to the batch-corrected model used for the cancer type-corrected model in our study, all these genes would be significant as before. (As detailed in our last rebuttal, these genes were not selected based on FDR but on integrating protein results with other results, e.g., mRNA or DNA methylation results, etc.)

Symbol	cancer type+CNA model			batch+CNA model		
	1MB	1MB	1MB	1MB	1MB	1MB
	t	p	fdr	t	p	fdr
AKR1C4	3.149873	0.001717	0.233765	3.155456	0.001685	0.22637
AKR1C1	3.972605	7.61E-05	0.05113	3.955637	8.16E-05	0.048394
AKR1C2	4.295595	1.88E-05	0.018962	4.2568	2.23E-05	0.022515
AKR1C3	3.522545	0.000443	0.114354	3.681848	0.000241	0.090052
IGF2	3.099749	0.001988	0.235692	3.072993	0.002174	0.238082
NID2	3.784141	0.000161	0.065027	3.838633	0.00013	0.062257
ANO1	3.411187	0.000684	0.146657	3.291605	0.001046	0.178708
G6PD	2.829684	0.004732	0.3037	2.810422	0.005023	0.31437
KDM2A	2.686961	0.007333	0.364	2.68364	0.007406	0.364028
CDK2	3.483032	0.000513	0.120122	3.468249	0.000541	0.121234
CCND1	2.317913	0.020778	0.516976	2.246615	0.025016	0.527371
EGFR	4.745773	2.31E-06	0.007757	4.730117	2.49E-06	0.006278
KRAS	2.867392	0.004206	0.292306	2.765788	0.00576	0.326075
NF1	-2.9243	0.003513	0.283193	-2.90828	0.003697	0.275951
PTEN	-2.62427	0.008793	0.395583	-2.67151	0.007653	0.367252

The differences in p-values we observe between the models do not point to any technical artifact involving one model over the other but rather reflect statistical noise. The differences are very slight and do not consistently go in one direction. Our study's overall findings would remain the same, whichever model is used. **In the manuscript revision, we note the following in Methods: "In this study, cancer type largely reflected the batches involved in the expression compendium datasets, as each cancer type (by tissue of origin) was contained within one or two batches (as defined by separate studies), and no batches involved multiple cancer types. Alternative linear models incorporating the 15 batches as a covariate instead of the 12 cancer types (Table S1) yielded the same overall results as those used in our study."**

In summary, the PCA plots of Figure S1g are not indicative of serious batch effects, as direct comparisons between the batches do not yield any significant proteins over chance expected. Our analyses used in the study would largely correct for any possible residual batch effects, through their inclusion of cancer type as a covariate. An alternative model that explicitly uses batch as a covariate does not lead to any improvement in the protein-SV associations.

References:

- 1 Zhang, Y., Chen, F., Chandrashekar, D., Varambally, S. & Creighton, C. Proteogenomic characterization of 2002 human cancers reveals pan-cancer molecular subtypes and associated pathways. *Nat Commun* **13**, 2669, doi:10.1038/s41467-022-30342-3 (2022).
- 2 Chen, F., Chandrashekar, D., Varambally, S. & Creighton, C. Pan-cancer molecular subtypes revealed by mass-spectrometry-based proteomic characterization of more than 500 human cancers. *Nat Commun* **10**, 5679, doi:10.1038/s41467-019-13528-0 (2019).
- 3 Chen, F. *et al.* Pan-cancer molecular classes transcending tumor lineage across 32 cancer types, multiple data platforms, and over 10,000 cases. *Clin Cancer Res.* **24**, 2182-2193 (2018).
- 4 Chen, F. *et al.* Pan-urolologic cancer genomic subtypes that transcend tissue of origin. *Nat Commun* **8**, 199 (2017).
- 5 Zhang, Y., Chen, F. & Creighton, C. Pan-cancer molecular subtypes of metastasis reveal distinct and evolving transcriptional programs. *Cell Rep Med* **4**, 100932, doi:10.1016/j.xcrm.2023.100932 (2023).
- 6 Johnson, W., Rabinovic, A. & Li, C. Adjusting batch effects in microarray expression data using Empirical Bayes methods. *Biostatistics* **8**, 118-127 (2007).
- 7 Zhang, Y. *et al.* Global impact of somatic structural variation on the DNA methylome of human cancers. *Genome biology* **20**, 209 (2019).
- 8 Clark, D. *et al.* Integrated Proteogenomic Characterization of Clear Cell Renal Cell Carcinoma. *Cell* **179**, 964-983 (2019).
- 9 Li, Y. *et al.* Histopathologic and proteogenomic heterogeneity reveals features of clear cell renal cell carcinoma aggressiveness. *Cancer Cell* **41**, 139-163 (2023).

Reviewers' Comments:

Reviewer #3:

Remarks to the Author:

My concerns are adequately addressed.

Our responses and comments are marked in red italics.

Reviewer #3:

My concerns are adequately addressed.

We thank the reviewer for evaluating our work.